# A modern automated patch-clamp approach for high throughput electrophysiology recordings in native cardiomyocytes

Fitzwilliam Seibertz [1,2,3], Markus Rapedius [4✉], Funsho E. Fakuade[1,2,3], Philipp Tomsits [5,6,7,8], Aiste Liutkute [1,2,3], Lukas Cyganek[2,3,9], Nadine Becker[4], Rupamanjari Majumder[2,10], Sebastian Clauß [5,6,7,8], Niels Fertig [4✉] & Niels Voigt [1,2,3✉]

Crucial conventional patch-clamp approaches to investigate cellular electrophysiology suffer from low-throughput and require considerable experimenter expertise. Automated patch-clamp (APC) approaches are more experimenter independent and offer high-throughput, but by design are predominantly limited to assays containing small, homogenous cells. In order to enable high-throughput APC assays on larger cells such as native cardiomyocytes isolated from mammalian hearts, we employed a fixed-well APC plate format. A broad range of detailed electrophysiological parameters including action potential, L-type calcium current and basal inward rectifier current were reliably acquired from isolated swine atrial and ventricular cardiomyocytes using APC. Effective pharmacological modulation also indicated that this technique is applicable for drug screening using native cardiomyocyte material. Furthermore, sequential acquisition of multiple parameters from a single cell was successful in a high throughput format, substantially increasing data richness and quantity per experimental run. When appropriately expanded, these protocols will provide a foundation for effective mechanistic and phenotyping studies of human cardiac electrophysiology. Utilizing scarce biopsy samples, regular high throughput characterization of primary cardiomyocytes using APC will facilitate drug development initiatives and personalized treatment strategies for a multitude of cardiac diseases.

[1] Institute of Pharmacology and Toxicology, University Medical Center Göttingen, Georg-August University Göttingen, Göttingen, Germany. [2] DZHK (German Center for Cardiovascular Research) partner site Göttingen, Göttingen, Germany. [3] Cluster of Excellence "Multiscale Bioimaging: from Molecular Machines to Networks of Excitable Cells" (MBExC), Georg-August University Göttingen, Göttingen, Germany. [4] Nanion Technologies GmbH, Munich, Germany. [5] Department of Medicine I, University Hospital, LMU Munich, Munich, Germany. [6] Institute of Surgical Research at the Walter-Brendel-Centre of Experimental Medicine, University Hospital, LMU Munich, Munich, Germany. [7] DZHK (German Centre for Cardiovascular Research) Partner Site Munich, Munich Heart Alliance (MHA), Munich, Germany. [8] Interfaculty Center for Endocrine and Cardiovascular Disease Network Modelling and Clinical Transfer (ICONLMU), LMU Munich, Munich, Germany. [9] Stem Cell Unit, Clinic for Cardiology and Pneumology, University Medical Center Göttingen, Georg-August University Göttingen, Gottingen, Germany. [10] Biomedical Physics Group, Max Planck Institute for Dynamics and Self Organisation, Gottingen, Germany. ✉email: markus.rapedius@nanion.de; niels.fertig@nanion.de; niels.voigt@med.uni-goettingen.de

Action potentials (AP) are classical electrical hallmarks of excitable cells such as cardiomyocytes and neurons. AP shape and duration are determined by the balanced sequential interaction of various ion channels and transporters residing in cellular membranes. Alterations in disease-relevant nanoscale functional units of the heart and brain often translate into the abnormal function of these channels, which can manifest as severe diseases with adverse clinical outcomes such as cardiac arrhythmias or neurological disorders[1–5]. Therefore, assessing the functional abnormalities of these channels and the resulting alterations in AP morphology on a cellular level is of the utmost importance to understand the impact of molecular abnormalities on whole organ function.

In recent years, high-throughput sequencing and proteomics techniques have become more available and accessible for molecular biology and protein biochemistry studies. These allow for detailed analysis of disease-relevant molecular abnormalities of ion channels and ion channel regulators. In contrast, methods that study cellular electrophysiology are largely based on manual approaches, such as the traditional patch-clamp, in which a skilled experimenter must slowly investigate individual cells one by one, resulting in very low throughput. This discrepancy illustrates a major limitation for the investigation of excitable cells and their regulatory units. Recently, tremendous progress has been made in the development of high-performance automated patch-clamp (APC) systems that allow for high throughput electrophysiological measurements[6–8] (Supplementary Fig. 1). Nevertheless, APC techniques are currently limited to the investigation of cultured cell lines and expression systems, which are characterized by relatively small and homogenous cell sizes. While useful, these constructs do not fully reflect cellular function and ion channel dynamics within a more complex, heterogeneous in vivo environment[9]. The restriction of APC systems to smaller cells is potentially related to the fact that many APC systems operate based on microfluidic 'flow-through design' approaches to handle cellular material and provide rapid and accurate solution exchange (Fig. 1a, b). This geometry appears to prevent larger non-symmetrical cells such as native cardiomyocytes from successfully reaching and attaching to the patch-clamp aperture. In addition, shear stress imposed on sensitive native cardiomyocytes is clearly suboptimal, possibly resulting in cell membrane damage and thus difficulties to establish stable high resistance seals for robust patch-clamp studies[10–12].

To circumvent these limitations, we have established an APC chip that is based on a fixed well format with 384 wells. Cells are automatically pipetted into each well and, utilizing gravity and suction pressure, settle on a borosilicate-glass base. This is equipped with the patch-clamp aperture, onto which cells attach (Fig. 1a, b).To the best of our knowledge, we report for the first time the successful application of a high throughput APC system for the recording of APs and ion currents in freshly isolated mammalian atrial and ventricular cardiomyocytes.

## Results

Native cardiomyocyte isolation produced $8790 \pm 1610$ viable atrial and $7200 \pm 3903$ viable ventricular cardiomyocytes per isolated heart ($n = 3$). Partial plates were utilized to conserve cellular numbers and allow for efficient tests of reproducibility (Fig. 2a). Optical assessment of cell attachment rate to the patch-clamp aperture showed fewer numbers of native cardiomyocytes were necessary in solution to attach to 50% of available patch-clamp apertures, compared with the much smaller diameter of induced pluripotent stem cell-derived cardiomyocytes (hiPSC-CM; Fig. 2b, g). The patching success rate, defined as effective seal formation (>100 MΩ) and whole-cell configuration, achieved through gentle negative pressure application, was $13.9 \pm 1.7\%$ in native cells (15 plates over 3 days) and

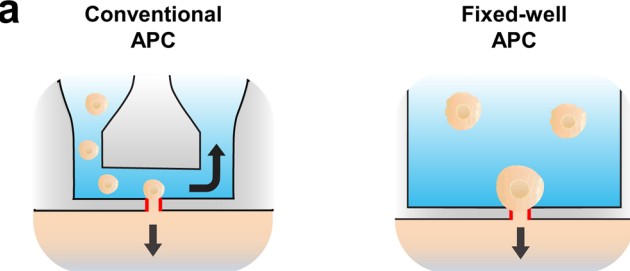

**a**
Conventional APC

Fixed-well APC

Human induced pluripotent stem cell-derived cardiomyocytes (hiPSC-CM)

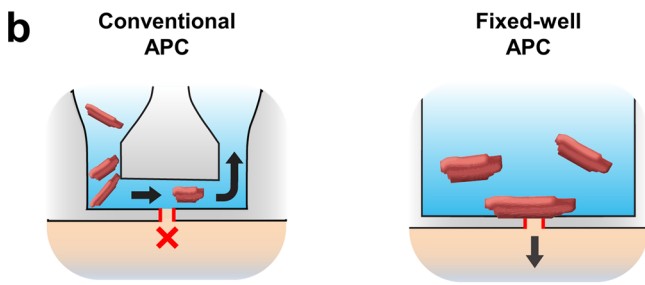

**b**
Conventional APC

Fixed-well APC

Native mammalian cardiomyocytes (Native-CM)

**Fig. 1 Simplified cross-sectional schematic of conventional microfluidic automated patch-clamp (APC) and fixed-well APC during assays with cultured cellular systems such as human induced pluripotent stem cell-derived cardiomyocytes (hiPSC-CM) or native mammalian cardiomyocytes (Native-CM). a** Schematic of hiPSC-CMs within APC systems. In a microfluidic APC system, cells and external bath solution are pipetted into the inlet well and travel through the channel to settle on the patch-clamp aperture (outlined in red). Waste liquid sequestration allows for fast flow-through perfusion and drug application (lateral arrow). In a fixed-well APC system, cells and bath solution are pipetted into the chamber and through gravity and suction are allowed to settle on the patch-clamp aperture (outlined in red). **b** Native cardiomyocytes within a microfluidic system cannot successfully reach or settle on the patch-clamp aperture due to size, sample debris, or high resistance channel-enhanced fluid velocity. These features are not present in a fixed well APC chamber. Cleanly isolated Native CMs can settle on the patch-clamp aperture for adequate seal formation and electrophysiological investigation.

showed no significant changes over successive experimental days. The overall success rate was comparable to that of high-quality hiPSC-CMs tested on the same APC system (Fig. 2c, f). Out of successful measurements in native cardiomyocytes, $29.3 \pm 5.9\%$ ($P < 0.05$) more recordings were obtained from atrial cells compared to ventricular cells. Seal quality, measured through cellular capacitance and $R_{series}$, remained stable in native cardiomyocytes during current acquisition similar to that of hiPSC-CMs (Fig. 2h). Analysis of $R_{series}$ and capacitance in native cardiomyocytes over different days (and therefore different animals) showed minimal, non-significant perturbations in native cardiomyocyte assay quality over subsequent days, indicating a reasonable rate of reproducibility (Fig. 2i). In addition, $Z$ factor analysis, a marker for high throughput assay reproducibility, consistently showed good to excellent values of assay robustness and reproducibility in multiple plates of primary cells over different days (Fig. 2c).

**L-type Ca$^{2+}$ current.** L-type calcium current ($I_{Ca,L}$) in native cardiomyocytes was measured with APC using a previously described voltage protocol[13,14], including a depolarizing ramp to inactivate

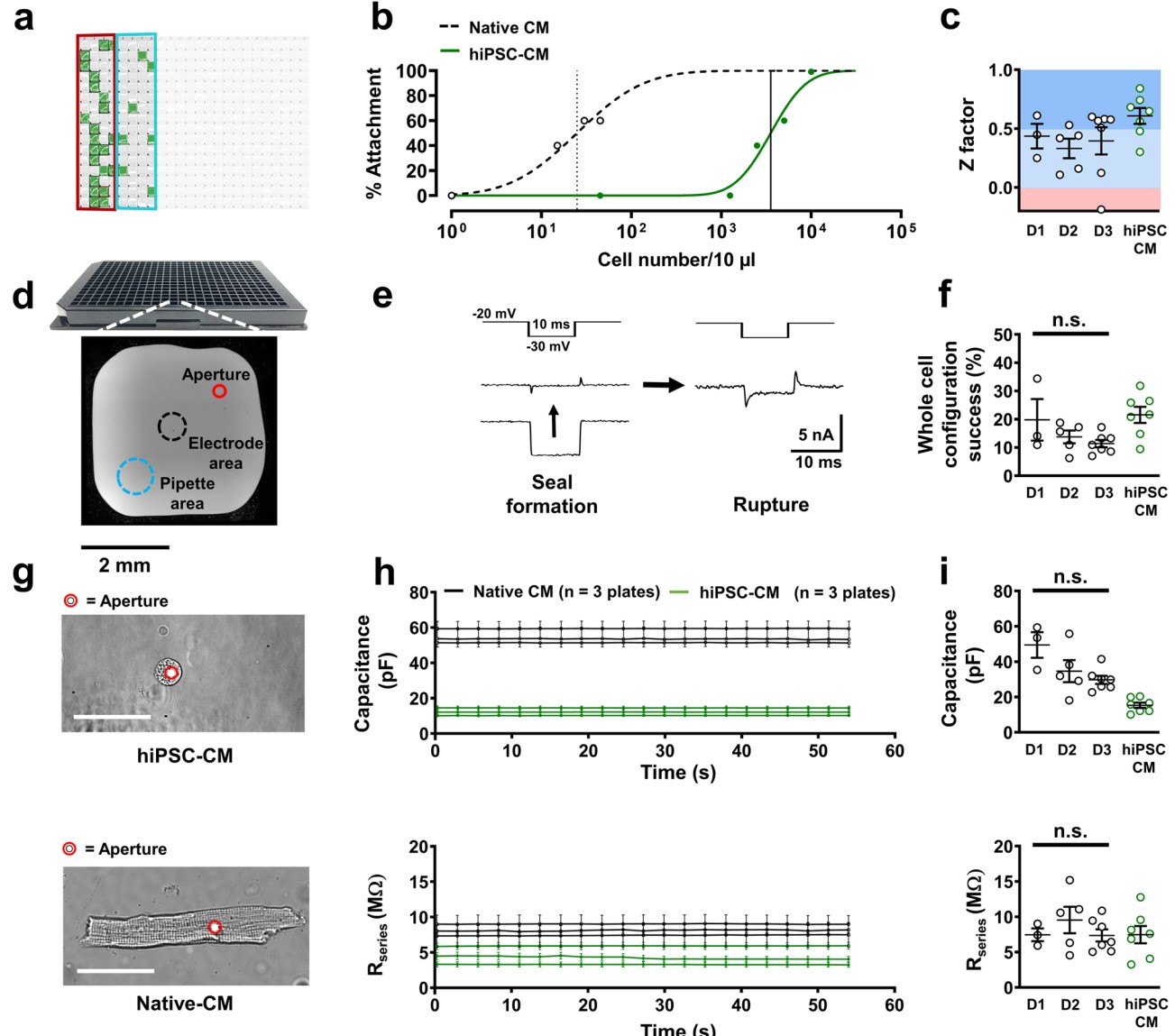

**Fig. 2 Quality control of automated patch-clamp of native atrial and ventricular cardiomyocytes (Native CM). a** Representative 384-well APC chip partially filled with native CMs is shown as a screenshot of Nanion DataControl 384 software with unused areas shaded out. Green boxes represent successful whole-cell configuration during L-type calcium current ($I_{Ca,L}$) measurement. Red and blue rings indicate atrial and ventricular CM partitions respectively. **b** Cellular density-dependent optical attachment rates of native ventricular cardiomyocytes and human ventricular induced pluripotent stem cell-derived cardiomyocytes (hiPSC-CM) onto the patch-clamp aperture. **c** $Z$ factor analysis of individual runs for assessment of reproducibility ($I_{Ca,L}$ currents) over three separate experimental days (D1, D2, D3) and therefore three animals. **d** NPC-384T chip with a photomicrograph of a single well with locations of interest labeled. **e** Representative membrane resistance during seal formation and membrane rupture of a native CM in the presence of a membrane test pulse. **f** Percentage success rates for whole-cell configuration from single plates of native CMs over three separate experimental days and hiPSC-CMs used in this protocol during $I_{Ca,L}$ acquisition. $P = 0.210$ **g** Photomicrograph of an attached ventricular hiPSC-CM and ventricular native CM. The patch-clamp aperture is obscured by the cell. Scale bar denotes 50 μm. **h** Time course of membrane capacitance and series resistance ($R_{series}$) in hiPSC-CMs (3 plates, same day) and native CMs (3 plates, same day) over successive $I_{Ca,L}$ experiments. **i** Cellular capacitance ($P = 0.054$) and $R_{series}$ ($P = 0.871$) averages from single plates over three separate experimental days. Data are mean ± SEM. n = mean of 24 wells (**b**) or single plates (**c**, **f**, **h**, **i**).

Na$^+$ channels prior to the pulse. Typical subtype characteristics were observed in $I_{Ca,L}$, with a larger density reported in ventricular myocytes compared to atrial (Fig. 3a–c). Peak $I_{Ca,L}$ (+10 mV) was also directly comparable to separate experiments utilizing traditional manual patch-clamp on the same cellular material, (Atrial: $-4.29 \pm 0.17$ [APC] vs. $-4.17 \pm 1.74$ [traditional patch-clamp] pA/pF; Ventricular: $-8.65 \pm 1.2$ [APC] vs. $-5.5 \pm 1.11$ [traditional patch-clamp] pA/pF; Fig. 3c and Supplementary Fig. 3). In addition, this clear subtype specificity was observed when this APC protocol was repeated using atrial and ventricular hiPSC-CMs using the

same system and APC chip design (Supplementary Fig. 4). In a separate cohort, I–V curves showed a non-linear current response to membrane depolarization, followed by a typical ohmic response after reaching peak current. Peak current in both atrial and ventricular native cardiomyocytes was reported at +10 mV (Fig. 3d). Further analysis of $I_{Ca,L}$ conductance revealed typical Ca$_V$1.2 activation curves in both groups (Fig. 3e). The application of an S1/S2 inactivation voltage protocol also produced robust inactivation curves in atrial and ventricular cells, further demonstrating the power of APC for detailed cellular characterization (Fig. 3e). Clear

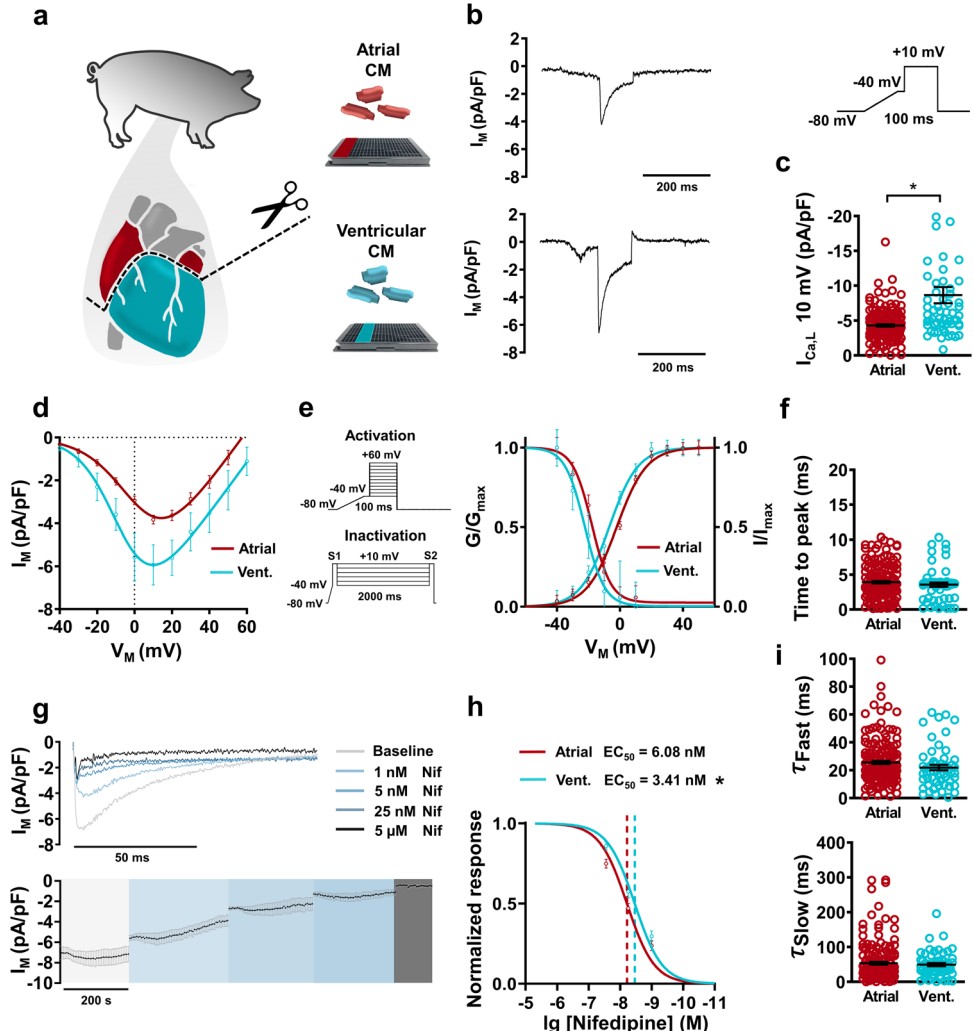

**Fig. 3 L-type calcium current ($I_{Ca,L}$) acquisition and analysis from native atrial and ventricular cardiomyocytes (CM) using automated patch-clamp.**
**a** Schematic of swine cardiac tissue harvesting and CM isolation. **b** Representative recordings of membrane current ($I_M$) showing $I_{Ca,L}$ in atrial and ventricular (vent.) CMs. Inset: Voltage protocol. **c** Peak $I_{Ca,L}$ density measured at +10 mV. **d** Current–voltage (I–V) relationship curves for $I_{Ca,L}$ in atrial ($n = 71$) and ventricular ($n = 26$) CMs. **e** Voltage protocols for I–V/activation experiments and S1/S2 voltage protocols for $I_{Ca,L}$ inactivation. Graph shows $I_{Ca,L}$ activation ($G/G_{max}$) in atrial ($n = 71$) and ventricular ($n = 26$) CMs, with corresponding inactivation($I/I_{max}$) curves in atrial ($n = 15$) and ventricular ($n = 6$) CMs. **f** Activation kinetics of $I_{Ca,L}$ at +10 mV expressed as a time to peak amplitude. **g** Representative concentration-dependent response to nifedipine (Nif) application in $I_{Ca,L}$ current with a time course of average $I_{Ca,L}$ from a single plate following nifedipine application ($n = 10$). **h** Concentration-response curve of $I_{Ca,L}$ (normalized to the current amplitude at the full block with 5 µM Nif) following nifedpine application in atrial and ventricular CMs with corresponding half maximal effective concentration (EC50). **l** Biphasic inactivation kinetics of $I_{Ca,L}$ at +10 mV expressed as fast decay ($\tau$fast) and slow decay ($\tau$slow). Data are mean ± SEM from three animals. *$P < 0.05$ vs. ventricular. n = number of atrial (176) and ventricular (58) CMs from 3 animals (**c**, **f**, **i**) or atrial (94) and Ventricular (39) (**h**).

temporal characteristics of $I_{Ca,L}$ activation and biphasic decay were also extracted at +10 mV (Fig. 3f, i). In order to test the possibility of high throughput drug screening of primary cardiomyocytes using APC, we applied successive concentration increases (1, 5, 25 nM and 5 µM) of nifedipine and assessed $I_{Ca,L}$ on atrial and ventricular preparations. Both subtypes showed appropriate concentration-dependent decreases of $I_{Ca,L}$ density (Fig. 3g). Full block was achieved with 5 µM nifedipine. To correct for potential current rundown over long experimental intervals, we kept the number of concentration steps low to ensure a short experimental time frame but still ensure useful concentration-response curve generation (Fig. 3h). When normalized to $I_{Ca,L}$ amplitude at full pharmacological block, atrial myocytes showed an $EC_{50}$ of $6.08 \pm 1.14$ nM and ventricular cardiomyocytes showed $3.41 \pm 0.71$ nM ($P < 0.05$).

**Action potential**. Robust APs were able to be elicited in both atrial and ventricular native cardiomyocytes using APC. Importantly, typical subtype-specific AP morphology was observed (Fig. 4a). Stimulated threshold current was significantly higher in ventricular myocytes (Fig. 4b). Atrial myocytes showed a shorter AP duration at 50% repolarization ($APD_{50}$) compared to ventricular (Fig. 4c). This typical discrepancy in chamber specific phenotype was also replicated using the same APC method in hiPSC-CMs generated using subtype-specific differentiation protocols (Supplementary Figure 4). In native cardiomyocytes, no change in resting membrane potential (RMP) between subtypes was detected (Fig. 4d). Indeed, during stimulation to elicit continuous APs, no diastolic depolarization or fluctuations in RMP were observed in native cardiomyocytes, indicating the capability for electrically stable current-clamp recordings with APC in native cardiomyocytes (Fig. 4e).

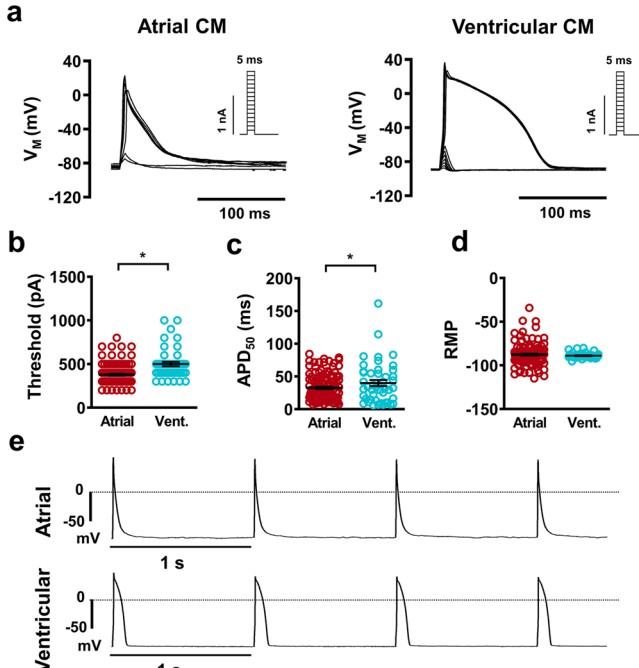

**Fig. 4 Action potential (AP) acquisition from native atrial and ventricular cardiomyocytes (CM) using automated patch-clamp. a** Representative traces membrane voltage ($V_M$) showing atrial and ventricular (vent.) triggered APs during successive increases in pulse current injection. Insets: current protocol. **b** Current at which AP take-off was first observed. **c** AP duration at 50% repolarization ($APD_{50}$). **d** Resting membrane potential (RMP) quantification. **e** Representative traces of AP trains at 1 Hz show stable RMP between pulses. Data are mean ± SEM. *$P < 0.05$ vs. ventricular. n = number of atrial (127) and ventricular (45) CMs from 3 animals (**b–d**).

**Inward rectifier currents**. Basal ($I_{K1}$) and atrial-specific acetylcholine-activated ($I_{K,ACh}$) inward rectifier currents in native cardiomyocytes were also assessed using APC (Fig. 5a, b). Using a previously described ramp voltage protocol and a high extracellular $K^+$ concentration (20 mM) to shift the reversal potential in a positive direction[15], atrial myocytes showed significantly less inward $I_{K1}$ density compared to ventricular which is in line with expected physiological values (Fig. 5c). Importantly, following the addition of the muscarinic receptor agonist carbachol (CCh), atrial cardiomyocytes displayed an extreme increase in inward current at −90 mV, which is quantified as $I_{K,ACh}$ (Fig. 5d). This current was completely absent in ventricular preparations. In-depth analysis of the CCh response shows a transient peak followed by typical desensitization to a steady state (Fig. 5b and Supplementary Fig. 5). Assessment of all inward rectifier currents also requires the addition of barium into the external solution to identify non-specific leak currents. As expected, inward rectifier activity in both atrial and ventricular myocytes was severely reduced following the addition of 1 mM $Ba^{2+}$ (Supplementary Fig. 5). This protocol gives a unique insight into the overall native cell APC assay quality due to the sequence of solution additions necessary to meaningfully complete the experiment. During the addition of CCh, and subsequently, the addition of $BaCl_2$, cellular capacitance and $R_{series}$ were continuously monitored to map the quality of the cell-aperture interface. Figure 5e, f shows representative time courses of both values, respectively, from one plate. The ratio of parameter change between the second solution addition (S2) and the third solution addition (S3) shows no significant difference from multiple plates over multiple days of

experimentation (Fig. 5g, h) indicating consistent seal quality during solution changes across all plates that were used.

**Multi-current (CAPER) protocol**. We aimed to exploit the solution exchange capabilities of APC by testing a multi-step, multi-current protocol to assess $I_{Ca,L}$, AP, and $I_{K1}$ sequentially from a single cell, provided electrical contact to the cytoplasm remained intact. Our Calcium—Action Potential—inward rectifiER (CAPER) protocol was applied through a sequence of internal (pipette) and external (bath) solution changes (Fig. 6a–c). This allows for unique visualization of cellular electrophysiology in three dimensions for assessment of how each current respectively and additively contributes to AP duration on a single cell basis (Fig. 6d). In this small cohort, we observed an increase of AP duration at 90% repolarization ($APD_{90}$) as $I_{Ca,L}$ amplitude increases in atrial cardiomyocytes. This uniformity was lost in ventricular cells and replaced with a stronger dependence on $I_{K1}$, as longer APs were generated by cells with lower $I_{K1}$ density (Fig. 6d). Both currents are crucial for AP physiology and morphology, and further investigation is required to verify and unravel this finding which is beyond the scope of the present work. in silico mathematical modeling is a powerful tool to investigate electrophysiological dynamics[16], and when comparing our CAPER results to our previously established model of porcine electrophysiology[17], some congruity was observed between experimental and modeled results (Supplementary Video 1). This proof of concept CAPER protocol was moderately successful, with 25% of successfully patched atrial cells providing traces for all three parameters, along with 24% of successful ventricular cells (Fig. 6b). Time traces of cell capacitance and $R_{series}$ from a representative plate reveal stable values throughout the CAPER protocol even with a slight decrease in $R_{series}$ after 13 min, possibly due to increased leak of repolarising currents in the presence of high external $K^+$ (Fig. 6e). Similar stability and current quality was observed in all successful experiments across different days. Representative screenshots of raw sequential CAPER measurements on single cells are shown in Fig. 6f and Supplementary Fig. 6. In the event of failure, an increase in $R_{series}$ could be observed as seal quality degraded. Cells that failed $I_{Ca,L}$ and AP assays sometimes showed clear inward rectifier currents. We assume this is due to the opposite phenomenon, where a suboptimal seal gains stability over time, possibly also due to a change in reversal potential.

## Discussion

Through the use of our fixed-well APC chip, we demonstrate that APC can provide stable subtype characterization of native mammalian cardiomyocytes. Numerous detailed electrophysiological parameters were able to be reproducibly extracted from separate cohorts of isolated swine native cardiomyocytes, which showed typical chamber subtype characteristics over a range of tests. Successful drug application highlighted the potential of this method for high throughput drug screening of native cardiomyocytes. In addition, our sequential CAPER protocol showed success in maximizing the data readout of live single cells, with moderate success of cell attachment rates during the exchange and the addition of multiple intracellular and extracellular solutions, respectively. Growing interest in APC expansion has so far produced successful recordings using native pancreatic cells, T-lymphocytes, and recently erythrocytes and cortical neurons[9,11,18]. After decades of failure to record from primary adult mammalian cardiomyocytes with APC, to our knowledge, we now report for the first time a highly versatile method for deep and user-unbiased electrophysiological phenotyping of primary cardiac material. These recordings are most likely possible due to the fixed-well format of the APC chip, and

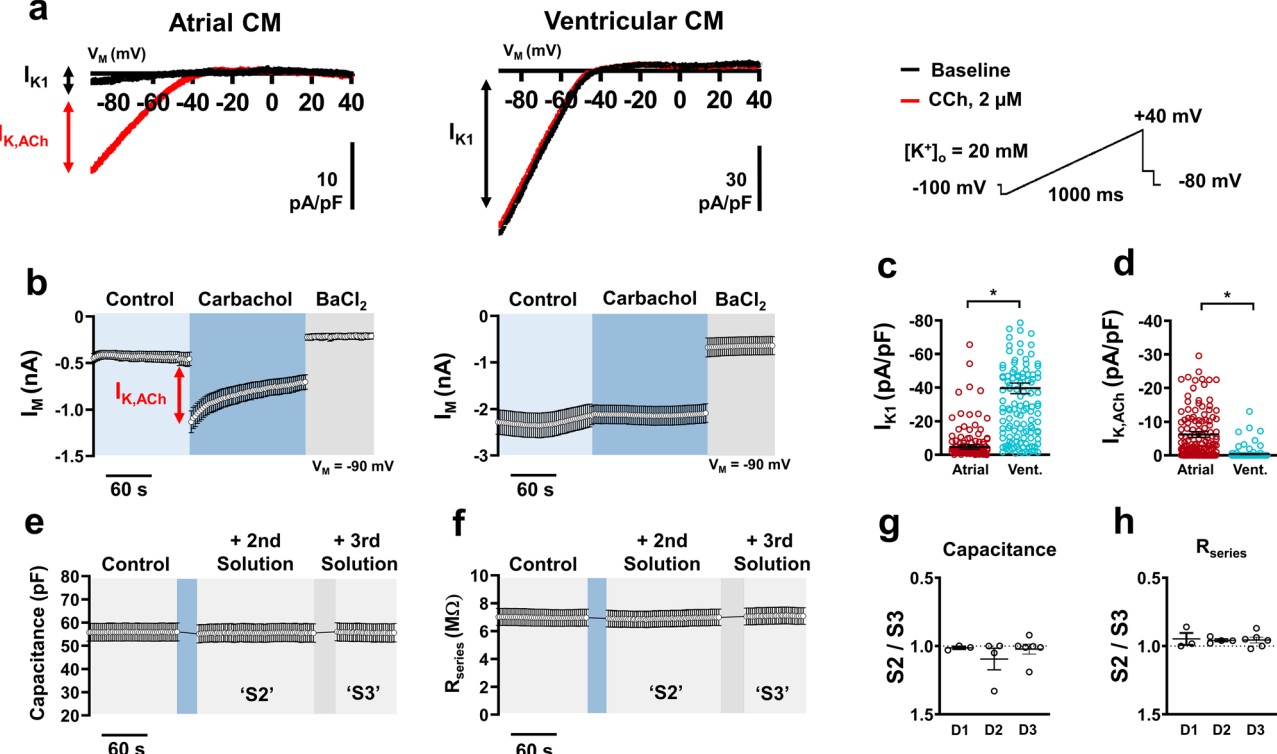

**Fig. 5 Inward rectifier acquisition from native atrial and ventricular cardiomyocytes (CM) using automated patch-clamp. a** Representative traces of membrane current ($I_M$) showing basal inward rectifier current ($I_{K1}$) in atrial and ventricular (vent.) with superimposed acetylcholine-activated inward rectifier ($I_{K,ACh}$) current following carbachol (CCh) application during a depolarizing ramp voltage protocol. **b** Time course of a single plate with atrial ($n = 10$) and ventricular ($n = 11$) CMs inward current at −90 mV during a typical experiment. Red arrow indicates peak $I_{K,ACh}$. **c** Peak inward $I_{K1}$ density measured at −90 mV. **d** Peak inward $I_{K,ACh}$ density measured at −90 mV. **e** Time course of membrane capacitance from a single plate (**b**, [$n = 10$]) over various external solution changes. **f** Time course of series resistance ($R_{series}$) from a single plate (**b**, [$n = 10$]) over various external solution changes. **g** Ratio of mean capacitance changes per plate between solution change 2 (S2) and solution change 3 (S3) over three separate experimental days (D1, D2, D3). **h** Ratio of $R_{series}$ between S2 and S3 over three separate experimental days. Data are mean ± SEM. *$P < 0.05$ vs. ventricular. $n =$ number of atrial (151) and ventricular (143) CMs from three animals (**c**, **d**).

our intensive cellular isolation procedure, which utilizes several filtration steps to produce clean cardiomyocyte populations with minimal debris and non-cardiomyocyte contaminants[19,20].

In this report, we highlight a broad variety of phenotyping techniques and parameters that are possible using APC in the study of primary mammalian cardiomyocytes. The protocols in this work allow for substantial conservation of resources, both biological and technical: (1) The absolute population of cardiomyocytes following isolation was lower than the amount of cultured biological material usually required for APC assays[7,12], yet robust results were regularly achieved. This is likely due to the large, bulky size of native isolated cardiomyocytes and the ease with which they could attach to the patch aperture (Fig. 2b). Indeed, despite the low cellular volume, whole-cell success rates in native cell assays were comparable to experiments on hiPSC-CM using the same system (Fig. 2f) and in conventional microfluidic APC systems[21]. (2) Traditional manual patch-clamp approaches require considerable experimenter skill and experience due to technically challenging setup requirements. In comparison, APC systems are operated in a more user-independent manner. Their relatively simple mode of operation facilitates the wide application of APC in many laboratories and hospitals. This more unbiased approach also increases data quality and reproducibility[22]. (3) Importantly, using APC to assess primary cellular material maximizes data output while conserving animal resources. In this study, we used three animals to generate a substantial amount of electrophysiological information for direct and detailed phenotyping and drug screening within a short time

period. In comparison, traditional manual patch-clamp methods utilizing the same swine construct yielded substantially fewer data from three animals (Supplementary Fig. 3). Primary cellular material is invaluable for disease phenotyping and drug development. However, due to ethical considerations, the use of animals must be regulated and kept to a minimum. Utilization of higher throughput methods such as APC for functional screening is therefore a desirable step towards reducing the number of animals sacrificed for disease modeling and cardiac safety screening purposes. Multi-current approaches with APC, such as the CAPER protocol, also inherently allow for maximal data output from minimal biological material. Such integrated protocols are possible with APC because the internal solution can be washed out and exchanged easily. In contrast, the corresponding pipette solution in a conventional patch-clamp system usually needs to be maintained throughout a complete experiment and is extremely difficult to exchange during an active recording[23].

High throughput characterization of native cardiomyocytes using APC is not only limited to investigating samples obtained from experimental animal models. Human cardiac samples can be obtained during various surgical procedures, for example right atrial biopsies from patients undergoing open heart surgery or left ventricular spindle biopsies during implantation of left ventricular assist devices in patients with end-stage heart failure[13,20,24]. Given the availability of standardized protocols for the isolation of cardiomyocytes from these samples[19,20], the application of APC systems for comprehensive patient-specific characterization of cellular electrophysiology will represent an important step to realistically

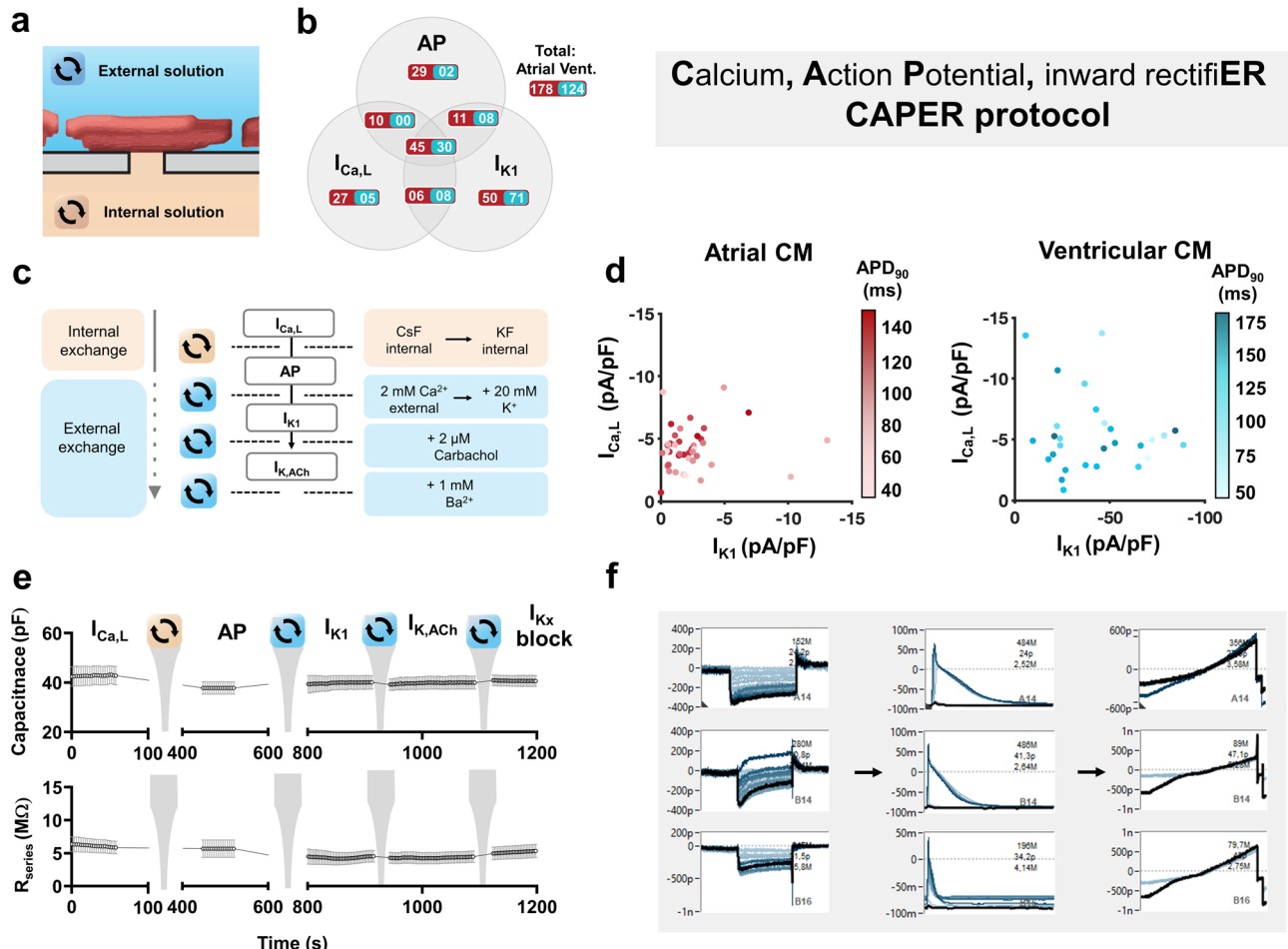

**Fig. 6 Overview of the multi-current Calcium, Action Potential and inward rectifiER (CAPER) protocol in native atrial and ventricular cardiomyoctes (CM) using automated patch-clamp. a** Schematic of the cardiomyocyte-aperture interface with corresponding external (blue) and internal (orange) solutions. The substitution of which allows for multi-current acquisition from a single cell. **b** Shares of total CM numbers that showed successful measurements of any of the three parameters of interest (L-type calcium current ($I_{Ca,L}$), action potential (AP) duration at 90% repolarization ($APD_{90}$) and basal inward rectifier current ($I_{K1}$). The central total indicates the successful cohort of atrial and ventricular (Vent.) CMs in which all three parameters were successfully acquired. **c** A detailed overview of the CAPER protocol including sequential internal and/or external solution exchange during a single experimental run. The acetylcholine-activated inward rectifier current ($I_{K,ACh}$) is an optional addition to the protocol prior to the inward rectifier block with $BaCl_2$. **d** Three-dimensional visualization of $I_{Ca,L}$, $APD_{90}$, and $I_{K1}$ relationships in atrial (left) and ventricular (right) CMs. $APD_{90}$ is expressed as a shadow spectrum where darker colors indicate longer AP duration. **e** Time course of membrane capacitance (upper) and series resistance ($R_{series}$; lower) from a single plate ($n = 5$ CMs) over a full CAPER run. **f** Direct acquisition software screenshots from a single plate (single animal) showing three complete electrophysiological measurements from single cells. Y-axis indicates the membrane current expressed as picoamperes (p) or nanoamperes (n) in the screenshots of $I_{Ca,L}$ (left) and $I_{K1}$ (right), or membrane voltage expressed as millivolts (m; center).

achieving the concept of personalized medicine. This could be particularly powerful when paired with artificial intelligence (AI) deep learning assemblies. Previous studies have implemented AI networks to predictively and precisely categorize patients based on minuscule patterns and repetitions within their clinical ECG[25]. Similar learning methods applied to high throughput data concerning human cardiac ionic activity could aid tremendously in highly sensitive patient-specific diagnosis and therapeutic treatment regimes.

As a proof of concept study, several limitations exist in our presently reported methodology. Our output cannot yet be considered truly high throughput, as our conservative partial plate utilization (max. 128 wells) restricts our data point sample size to a lower value than would be possible with a full 384-well plate. In addition, a high failure rate was observed during our CAPER protocol. This is an unfortunate consequence of attempting to keep an adequate cell–aperture assembly over a 30 min intensive experiment with many disruptive solution changes and additions.

Further optimization is required in order to increase the CAPER success rate, for example by increasing working temperature to more physiological temperatures while maintaining cell viability or selectively targeting the composition of extracellular and intracellular solutions to enhance seal quality. Such modifications should also be targeted toward reducing the heterogeneity of the presently reported experimental data within the cellular subtype cohorts. In addition, alternative approaches such as perforated patch-clamp could be considered for future experiments[26].

In typical APC experiments, fluoride ions in the internal solution seem to foster giga seal formation and help to increase electrical resistance and seal stability[27]. Fluoride-free recordings are a new benchmark in many APC applications to enable experiments under substantially more physiological conditions[27,28]. Such approaches were not tested during the present experiments.

Imperfect extracellular conditions may also facilitate the appearance of other 'contaminant' currents in the recordings. Our extracellular solution for $I_{Ca,L}$ measurement contains no $Na^+$

current blockers, therefore we rely on our voltage protocol depolarizing ramp to inactivate available $Na^+$ channels before each $I_{Ca,L}$-eliciting pulse. Should cells express pathological $Na_V1.5$ function, this voltage protocol alone may not be adequate to suppress ionic $Na^+$ activity. Successful pharmacological blockade of $I_{Ca,L}$ with nifedipine indicates trace contamination was unlikely in the presently reported summary (Fig. 3g, h).

The dependence on extracellular conditions during multi-current protocols exposes a disadvantage of the fixed-well format compared to conventional microfluidic systems. With our fixed-well APC chip, extracellular solutions cannot be fully exchanged or washed out after more than a few seconds of drug exposure. This means careful planning must be made when attempting sequential multi-current protocols such as CAPER to ensure each successive recording can proceed with adequate extracellular conditions. This is a clear advantage of microfluidic APC installations, as both intracellular and extracellular fluid can be completely and efficiently washed out and replaced with minimal fuss[7,11,12,26]. Detailed customization of multi-current protocols would therefore be much more versatile with microfluidic APC platforms.

Taken together, these experiments comprise a successful first step into high throughput, multi-parameter assessment of native cardiomyocyte electrophysiology using APC. This feat has been a desirable goal for many years. Further studies and methodologies can build upon these protocols for future mechanistic and phenotyping studies of human cardiac electrophysiology. Effectively bringing the proverbial 'bench' closer to the bedside, in the future this will reduce costs, increase data availability and more efficiently facilitate robust cardiac disease modeling studies, drug development initiatives, and personalized treatment strategies.

## Methods

**Native cardiomyocyte isolation.** All protocols were approved by the Regierung von Oberbayern (ROB-55.2-2532.Vet_02-18-69) and were conducted following the "Guide for the Care and Use of Laboratory Animals" (National Institute of Health, 8th Edition 2011). German landrace swine atrial and ventricular biopsies were obtained during animal surgery and subsequent animal sacrifice. Clean cardiomyocyte isolation was carried out according to our previously published standard protocol[19]. The tissue was cut into chunks of ~1 mm³ within a $Ca^{2+}$-free solution of (in mM): 20 glucose, 10 KCl, 1.2 $KH_2PO_4$, 5 $MgSO_4$, 5 MOPS, 100 NaCl, 50 taurine; pH 7.0 (with NaOH) at 4 °C. Chunks were subsequently washed, strained and resuspended in $Ca^{2+}$-free solution containing 286 U/ml collagenase type I (Worthington, USA) and 5 U/ml protease XXIV (Sigma-Aldrich) and stirred gently for 45 min at 37 °C. After 10 min, $CaCl_2$ was added to the solution for a final concentration of 20 μM. Following straining and resuspension with $Ca^{2+}$-free solution containing 286 U/ml collagenase type I and an additional 20 μM $CaCl_2$, cells were gently stirred for 5 min at 37 °C. Once isolated cardiomyocytes were detected in the solution, the supernatant was discarded, the remaining chunks were resuspended and mechanically titrated into separate cardiomyocytes in a storage solution of 1% w/v albumin and (in mM) 10 glucose, 10 β-hydroxybutyric acid, 70 L-glutamic acid, 20 KCl, 10 $KH_2PO_4$, 10 taurine; pH 7.4 (with KOH). Following centrifugation (90×g, 7 min, 37 °C) and resuspension of the pellet in storage solution, the viable cardiomyocytes were manually counted under bright-field conditions. 10 mM $CaCl_2$ was then gradually added to the solution over a period of 40 min for a final concentration of 0.2 mM $CaCl_2$. Samples were immediately used for APC analysis within 30 min of digestion cessation and within a window of 3 h after organ excision. Cells were not kept in the solution overnight, therefore every day of experimentation was conducted using cellular material from a different animal. Myocyte suspension was pipetted into 128 wells of the APC chip with 40 μl per well. This partial plate utilization concentrates cellular density over a smaller chip area, effectively increasing the chances of a successful experiment. Atrial and ventricular cardiomyocytes were investigated simultaneously on the same APC chip.

**Automated patch-clamp recordings.** All experiments were recorded using the SyncroPatch 384 (Nanion Technologies, Munich, Germany) at room temperature (21 °C). Negative pressure application between 150–200 mbar achieved whole-cell configuration. PatchControl 384 (Nanion Technologies) software allowed for the digitization and acquisition of data (system digitization rate: 10 kHz). Native cardiomyocytes were analyzed on thin borosilicate glass, single-aperture, low resistance (~2 μm diameter) 384-well planar fixed-well APC chips (1xS-type NPC-384T; Fig. 2a, d). The success rate was defined as the percentage of wells able to reach whole-cell configuration. Series resistance ($R_{series}$) and cell capacitance were

continuously measured from each well via test pulse application (10 ms negative square pulse from −20 to −30 mV before each sweep of the relevant voltage/current protocol (Fig. 2e). When starting an experiment, the APC chip is loaded with the desired pipette solution and separately, 30 μl of a divalent-free solution is automatically pipetted into each well. This initial extracellular solution contains (in mM): 10 HEPES, 140 NaCl, 4 KCl, 5 glucose. 20 μl of cell suspension is then administered to each well. Next, 40 μl of a solution containing (in mM):10 HEPES, 130 NaCl, 4 KCl, 10 $CaCl_2$, 1 $MgCl_2$, 5 glucose is temporarily added to each well to foster giga seal formation of captured cells. 40 μl of the solution is then removed from each well and replaced with the desired bath solution, taking into account the existing ionic concentrations already within the APC chip. A time course of seal resistance over these initial solution changes is shown in Supplementary Fig. 2. Solution withdrawal and addition throughout all experiments are precisely calibrated so when removing or adding solution, the pipettes constantly keep the maximum distance possible from the cells at the bottom of the APC chip. This limits potential cellular disruption and loss of seal integrity during solution exchanges.

L-type calcium currents ($I_{Ca,L}$) were measured at 0.5 Hz using a voltage-step protocol with a holding potential of −80 mV and a 100 ms ramp pulse to −40 mV. This is held for 20 ms before a 100 ms test pulse to +10 mV. Internal solution contained (in mM): 10 EGTA, 10 HEPES, 10 CsCl, 10 NaCl, 110 CsF, pH 7.2 (with CsOH). Bath solution contained (in mM): 10 HEPES, 140 NaCl, 5 glucose, 4 KCl, 2 $CaCl_2$, 1 $MgCl_2$, pH 7.4 (with KOH). The current–voltage (I–V) relationship and activation curves were measured by altering the test pulse from −40 mV by 10 mV every sweep including a final pulse of 60 mV. The I–V curves were fitted with a modified Boltzmann equation:

$$I_{Ca,L} = \frac{G_{max}(V - E_{rev})}{1 + \exp\left(\frac{[V_{50} - V]}{k}\right)} \tag{1}$$

where $G_{max}$ is the maximal conductance, $E_{rev}$ is the reversal potential, $V_{50}$ is the half-activation potential, and $k$ is the slope factor. $I_{Ca,L}$ inactivation was assessed with the application of an S1–S2 inactivation protocol consisting of a normal pulse (S1) to +10 mV, followed by a 2 s holding potential before a second +10 mV pulse (S2). The holding potential was altered from −40 to +10 mV during each sweep. The fast and slow time constants of the biphasic $I_{Ca,L}$ decay at peak current (+10 mV) were measured by fitting two standard exponential functions to each recording and assessing the tau (τ) for each fit. For method comparison, we also performed $I_{Ca,L}$ measurements in freshly isolated cardiomyocytes using a manual patch-clamp and application of the same original electrophysiological protocol (Supplementary Fig. 3). These manual experiments were carried out at 37 °C.

APs were evoked in current-clamp configuration with the application of 5 ms current pulses, with steps of 100 pA, starting at 100 pA until 1.5 nA. AP were analyzed at a stimulus 20% above that required to reach the threshold. Once the threshold was determined, AP trains at 1 Hz allowed for visualization of RMP stability as an index of seal quality. Internal (pipette) solution contained (in mM) 10 EGTA, 10 HEPES, 10 KCl, 10 NaCl, 110 KF, pH 7.2 (with KOH). Bath solution contained (in mM): 10 HEPES, 140 NaCl, 5 glucose, 4 KCl, 2 $CaCl_2$, 1 $MgCl_2$, pH 7.4 (with KOH). Injected holding current (Atrial: −3.5 ± 0.7 pA/pF, Ventricular: −12.8 ± 2.6 pA/pF) was administered independently to each well as required.

Basal inward rectifier current ($I_{K1}$) was measured in voltage-clamp configuration at 1 Hz using a voltage protocol with a holding potential at −80 mV followed by a depolarizing ramp pulse from −90 to +40 mV. Internal (pipette) solution contained (in mM) 10 EGTA, 10 HEPES, 10 KCl, 10 NaCl, 110 KF, pH 7.2 (with KOH). Bath solution contained (in mM): 10 HEPES, 140 NaCl, 5 glucose, 20 KCl, 2 $CaCl_2$, 1 $MgCl_2$, pH 7.4 (with KOH). Here, an external KCl concentration of 20 mM was used to facilitate a positive shift in $I_{K1}$ reversal potential and allows for a larger current acquisition of the inward component of $I_{K1}$ at −90 mV. $I_{K1}$ was identified as current responsive to $Ba^{2+}$ blockade (1 mM).

Acetylcholine-activated inwardly rectifying current ($I_{K,ACh}$) was identified using the same protocol as that for $I_{K1}$ following the external application of 2 μM carbachol (CCh), a muscarinic receptor agonist, which selectively opens atrial specific $I_{K,ACh}$ channels. $I_{K,ACh}$ was defined as the initial (peak) CCh-dependent current increase of inward current which is measured at −90 mV[29].

Using an integrated multi-current protocol, we aimed to measure $I_{Ca,L}$, AP and $I_{K1}$ (Calcium, Action Potential, inward rectifiER; CAPER) from single cells during a single experiment. The CAPER protocol consisted of sequential programs of external and internal fluid exchange and voltage-step protocols to allow for the acquisition of $I_{Ca,L}$, AP, and $I_{K1}$. Using the measurement parameters outlined above, $I_{Ca,L}$ was measured first, subsequent to a change of pipette solution and amplifier configuration to current-clamp to measure APs. The amplifier was then switched back to the voltage clamp, and the ramp voltage protocol was applied with an increased external $K^+$ concentration (20 mM) to measure inward $I_{K1}$ components. The same cardiomyocytes remained attached to the patch-clamp aperture throughout the experimental run. $R_{series}$ and capacitance were monitored throughout.

Assay quality was quantified through $R_{series}$ and cellular capacitance stability during a typical $I_{Ca,L}$ experiment, and also through a previously described reproducibility test (Z factor, Z') that has been designed to evaluate the success of a high-performance assay performed on a single plate in the absence of

pharmacological modulation[7,30]. $Z'$ is primarily used to assess how reproducible an assay technique is on a day-to-day, or assay-to-assay basis. $Z'$ was calculated using the equation:

$$Z' = 1 - \frac{3(s_1 + s_2)}{\bar{x}_1 - \bar{x}_2} \quad (2)$$

where $s$ is the standard deviation, $\bar{x}$ = mean, $_1$ = experimental group expressing maximal current (native CM), $_2$ = control group expressing minimal current. The control group used to assess minimal current consisted of Chinese hamster ovary (CHO) cells stably expressing Nav1.5 (Charles River Laboratories). These were measured in the same plate as the native cells for each experiment. The $I_{Ca,L}$ protocol was used to assess $Z'$. $Z' < 0$ indicates low reproducibility between assays, $0 < Z' < 0.5$ indicates a good and reproducible assay, $Z' > 0.5$ indicates excellent quality.

**Statistics and reproducibility**. Current amplitudes, density, and decay were analyzed offline using DataControl 384 (Nanion Technologies) software. $I_{Ca,L}$, $I_{K1}$, and $I_{K,ACh}$ of atrial and ventricular native cardiomyocytes were ratioed to cell capacitance and expressed as current density. Results were only included if they showed a seal resistance of >100 MΩ, a peak current ($I_{Ca,L}$: +10 mV, $I_{K1}/I_{K,ACh}$: −90 mV) of >50 pA, and a $R_{series}$ of <200 MΩ (at 10 mV). AP parameters such as RMP and AP duration were analyzed offline using DataControl 384 software. Using the AP search feature of the software, APs that did not display a clear threshold take-off potential following increasing current stimuli were excluded. Summarized data are reported as mean ± SEM. Success metrics were measured over 3 successive days using 1 animal per day. Across different days, significance was assessed using one-way ANOVA with the Bonferroni correction. Numerical data were analyzed using an unpaired two-tailed Student's $t$-test. $P < 0.05$ was considered to be statistically significant.

**hiPSC reprogramming and culture**. All protocols were approved by the ethics committee of the University Medical Center Göttingen (10/9/15). Human induced pluripotent stem cell (hiPSC) line UMGi014-C clone 14 (isWT1.14) was derived from the dermal fibroblasts of a healthy male donor. Reprogramming was achieved with the integration-free CytoTune iPS 2.0 Sendai Reprogramming Kit (Thermo Fisher Scientific) with reprogramming factors OCT4, KLF4, SOX2, c-MYC. hiPSC were cultured on 1:120 Matrigel[TM] (BD Biosciences) coated plates and maintained with Stem MACS IPS-Brew XF medium (Miltenyi Biotec) daily. Cells underwent passaging twice a week.

**Directed cardiac differentiation**. Subtype-directed differentiation was achieved following our previously published standard protocols[31,32]. Here, 80–90% confluent hiPSC monolayers were supplied with 4 μM CHIR99021 (Sigma-Aldrich) on day 0 (d0) in a 'Differentiation Medium' containing: RPMI 1640 with GlutaMAX (Thermo Fisher Scientific), 0.5 mg/ml human recombinant albumin and 0.2 mg/ml L-ascorbic acid 2-phosphate (all Sigma-Aldrich) for 48 h. Subsequent application of 2.5 μM IWP2 (Sigma-Aldrich) for a further 48 h stimulates WNT signaling cessation. Atrial subtype differentiation was achieved with the application of 1 μM retinoic acid during d3–d6. After day 8 medium was changed to a 'Culture Medium' containing: RPMI 1640 with GlutaMAX, and 2% B27 (Thermo Fisher Scientific). Cardiomyocyte purification by lactate selection was performed between d15 and d20 with a 'Selection Medium' containing RPMI 1640 without glucose (Thermo Fisher Scientific), 0.5 mg/ml human recombinant albumin, 0.2 mg/ml L-ascorbic acid 2-phosphate, and 4 mM lactate (all Sigma-Aldrich). Subsequently, hiPSC-CMs were maintained with a culture medium every 2–3 days. For experimentation, hiPSC-CMs were incubated with PBS/EDTA (Thermo Fisher Scientific) for 5 min at 4 °C. A mixture of 25% Dispase (1:10 dilution), 75% Accumax (all Sigma-Aldrich) was applied to the cells for 35–45 min at 37 °C. Cells were returned to 4 °C for 10 min and then exposed to pre-cooled HBSS (Thermo Fisher Scientific) at a volume to provide sufficient cellular density (>100,000 cells per ml) and sparingly resuspended with a glass pipette in order to detach cells into isolated bodies.

hiPSC-CM experiments were conducted under the same conditions as native cardiomyocytes. Inward rectifier ($I_{K1}$) measurements and multi-parameter protocols such as CAPER were not attempted with hiPSC-CM. During AP acquisition, injected holding current (−9.7 ± 1.5 pA/pF) was administered independently to each well as required.

**In silico modeling**. Though combining our experimental CAPER data from atrial cardiomyocytes with an established mathematical in silico model of swine atrial physiology[17], we see a glimpse of the reliability of our multi-dimensional experimental results (Supplementary Movie 1). A three-dimensional surface plot was generated by recording AP characteristics and currents from 10,000 individual in silico swine atrial cardiomyocytes whose parameters (ion channel conductances, time constants, and rate constants for reaction kinetics) were randomly varied within a preset spread (50–150%) of the average model value. After reaching a steady state (5 s), the in silico cells were stimulated with a 4 ms pulse of 20 pA. APD$_{90}$ was acquired from all simulations able to generate an AP (~40%) and the corresponding peak $I_{Ca,L}$ and $I_{K1}$ were obtained from each successful cell at +10 and −90 mV, respectively, as was used experimentally. Note: our experimental recording temperature (21 °C) differs from that of the model (37 °C).

**Reporting summary**. Further information on research design is available in the Nature Research Reporting Summary linked to this article.

## Data availability
All data generated and analyzed during this study are included in this published article and in Supplementary Data 1, which contains the source data underlying figures.

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

## Acknowledgements

The authors thank Maren Dilaj for excellent secretarial help, Valerie Pauly and Nora Hesse for their exceptional veterinary support, and the team at Nanion Technologies GmbH for their exceptional assistance and warm collaborative welcome to their facility.

## Author contributions

F.S., M.R., N.F., and N.V. designed the studies. F.S., F.E.F., M.R., N.B., R.M., and N.V. performed the research and analyzed the data. P.T. and S.C. performed the animal surgery, providing tissue samples and medical expertise. L.C. and A.L. provided cellular material and reprogramming and differentiation expertise. F.S., N.F., and N.V. wrote the manuscript. All authors give their consent for the publication of the above manuscript.

## Funding

This work was funded by the Deutsche Forschungsgemeinschaft (DFG, German Research Foundation) under Germany's Excellence Strategy—EXC 2067/1- 390729940. We acknowledge the support from the DFG to N.V. (VO 1568/3-1, VO1568/4-1, IRTG1816, SFB1002 project A13) and to P.T. (Clinician Scientist Program In Vascular Medicine (PRIME, MA 2186/14-1)), from the Else-Kröner-Fresenius Foundation to N.V.

(EKFS 2016_A20), from the German Center for Cardiovascular Research to N.V. (DZHK, 81X2300189, 81X4300102, 81X4300115, 81X4300112) and S.C. (81X2600255), from the Corona Foundation to S.C. (S199/10079/2019), and from the ERA-NET on Cardiovascular Diseases to S.C. (ERA-CVD; 01KL1910). Open Access funding enabled and organized by Projekt DEAL.

## Competing interests

The authors declare the following competing interests: M.R. and N.B. are employed by, and N.F. is CEO of, Nanion Technologies GmbH. All other authors declare no competing interests.
