## [Peer Review File · Communications Biology]

Reviewer #1 (Remarks to the Author):

SUMMARY

In this manuscript, the authors present patch clamp data from native cardiomyocytes that were obtained with their SyncroPatch 384 (Nanion Technologies, Munich, Germany) automated patch clamp apparatus that was modified to facilitate recordings from native cardiomyocytes in addition to small and round human induced pluripotent stem cell derived cardiomyocytes (hiPSC-CMs).

OVERALL IMPRESSION

I read this manuscript on automated patch-clamp of native cardiomyocytes with great interest. Unfortunately, it read like a salesperson presentation of Nanion Technologies rather than a scientific paper. It is up to the Editorial Board members of Communications Biology whether the journal is willing to function as a platform for promoting specific commercial products.

As an interested reader, I was constantly skipping back and forth between the main manuscript and its supplement, demonstrating that parts of the supplement should be included in the main manuscript. If the word count and/or number of figures of the main manuscript are limited, which I did not check, the authors should consider to fuse the two figures of the main manuscript and the five of the supplement into only a few figures that contain the essential information, focusing on the actual patch clamp recordings. In its current form, the seven figures look like parts of a PowerPoint presentation with many non-essential panels.

SPECIFIC COMMENTS

Abstract, lines 33–34: “Automated patch-clamp (APC) approaches are experimenter independent and offer high-throughput, (...)” . In my experience, detailed electrophysiological data require time-consuming analysis by a patch clamp expert to avoid misinterpretation, thus seriously limiting the throughput and not being “experimenter independent”.

Abstract, lines 84–86: “Native cardiomyocytes were freshly isolated from pig hearts as described previously [6] and in the Methods section (Supplementary Information), resulting in an average cell yield of 6,000 cells per ml.” How was this “average cell yield” determined and how large was its variation between isolations?

Main text, lines 95–96: “The same cardiomyocytes remained attached to the patch clamp aperture throughout the experimental run.” So, why does the number of myocytes differ between parts of the experimental run shown in Figure 2C? As an experimenter, I would like to know how many cells (absolute and percentage) survive the entire protocol and how many of these provided useful experimental data.

Main text, lines 109–112: “By quantifying a broad range of electrophysiological parameters we furthermore demonstrate that APC is applicable for a detailed range of mechanistic studies using primary cardiac cellular material to suit the specific needs of the experimenter.” This reads as an overstatement, given that the obtained data (Figure 2; Main text, lines 98–105) are limited to some general electrophysiological properties of atrial and ventricular myocytes. As an experimenter, I would like to know how well detailed patch clamp data, like current-voltage relationships, steady-state (in)activation curves and time constants of (in)activation of one or more individual membrane currents, can be achieved. According to the “Automated patch-clamp recordings” subsection of the Methods section of the Supplementary Information, at least the current-voltage relationship of the L-type calcium current was measured using the APC technique “by altering the test pulse from -40 mV by 5 mV every sweep including a final pulse of 60 mV”. However, this current-voltage relationship is neither shown in the manuscript nor in its supplement.

Main text, lines 129–132: “APC systems allow for consecutive measurements of APs and various membrane currents in the very same cell. This is due to the fact that the internal solution of APC platforms, which is directly connected to the cytosol, can be exchanged relatively easily.” As an experimenter, I would like to know whether perforated patch clamp has been tried. This is important because this is required for reliable detailed investigations of specific membrane currents like the aforementioned L-type calcium current.

Figure 2. Panel C shows “ $I_{Ca,L}$ current density” and “ Ba^{2+} sensitive I_{K1} and $I_{K,ACh}$ current densities”, but it is not entirely clear how these “densities” are defined. Is the “ $I_{Ca,L}$ current density” the (negative) peak at -10 mV? And are the “ Ba^{2+} sensitive I_{K1} and $I_{K,ACh}$ current densities” determined at -100 mV?

Figure 2. As an experimenter, I would like to see the actual recordings of panel A at a larger scale, so that the actual recordings can be appreciated. In its present form, it is difficult to make an estimate, but it seems that the reversal potential of the potassium currents differs between atrial and ventricular cardiomyocytes. If so, can you explain?

Figure 2. Do the two rightmost time scale bars of panel A (also) apply to the voltage clamp protocol?

Supplementary Information, lines 64–66: “Enzyme digestion procedures that produce cleanly isolated cardiomyocyte populations with minimal debris and non-cardiomyocyte contaminants are crucial to the eventual success of the assay.” Did you take special steps to arrive at these “cleanly isolated cardiomyocyte populations” or was it sufficient to strictly adhere to the protocol that you refer to at lines 46–47 (“Clean cardiomyocyte isolation was carried out according to our previously published standard protocol [1]”)?

Supplementary Information, lines 66–67: “Isolates used in this study contained 60 cells per $10 \mu\text{l}$ of external solution.” Is this number, which is equivalent to the aforementioned “average cell yield of 6,000 cells per ml” an estimate? How did you arrive at this number?

Supplementary Information, lines 71–72: “PatchControl 384 (Nanion Technologies) software allowed for the digitalization and acquisition of data.” At which frequencies were the data acquired and digitized?

Supplementary Information, lines 72–73: “Native cardiomyocytes were analysed at room temperature (...)”. It is only here that it is mentioned that the data were acquired at room temperature. This should (also) be mentioned in the main text. Recording at room temperature may be inherent to the SyncroPatch 384 apparatus. If so, it is a clear limitation that should be addressed in the main text.

Supplementary Information, lines 74–78: “Seal resistance, series resistance and cell capacitance were measured from each well via test pulse application and were continuously recorded over the entire experiment, which involved sequential acquisition of all currents detailed below (Supplementary Figure 3).” What was the composition of these test pulses, at which time during the experiment were these applied, and were these test pulses changed during the experiment to account for the changes in the internal or external solutions?

Supplementary Information, lines 79–82: “L-type calcium currents ($I_{Ca,L}$) were measured at 0.5 Hz using a voltage-step protocol with a holding potential of -80 mV and a 100 ms ramp pulse to -40 mV followed by a 100 ms test-pulse to $+10$ mV. The I-V relationship was measured by altering the test pulse from -40 mV by 5 mV every sweep including a final pulse of 60 mV.” Was this protocol repeated to assess any changes in the outcome over time? In other words: is this calcium current stable over time?

Supplementary Information, lines 105–108: “Acetylcholine-activated inwardly rectifying current (IK,ACh) was identified using the same protocol as that for IK1 following the application of 2 μ M Carbachol, an M-receptor agonist, which selectively opens atrial specific IK,ACh channels. Both IK1 and IK,ACh were identified as current responsive to Ba²⁺ blockade (1 mM).” It would be helpful to show a typical example of the original current traces in the order of their recording, together with the current differences that were used to arrive at the IK1 and IK,ACh traces.

Supplementary Figure 2. Panel E shows the “overall whole-cell success rates of native CM and hiPSC-CM used in this protocol”, but it is not explained how these “success rates” were defined and determined.

Reviewer #2 (Remarks to the Author):

The authors present automated patch-clamp (APC) recordings of cardiomyocytes (CMs) from swine and human PSCs. The innovation is the use of “newly developed APC plate format”.

Assessment

This is an interesting and important technological advance. I believe the authors can improve the manuscript and enhance its (future) impact by providing some additional data and modifying the figures.

Specific comments

1. Figure 1 should explicitly show the dimensions of fixed well and this can be complemented by showing a higher magnification of a region of the fixed well plates in Figure 2 of the Supplement.
2. I am not sure I know what “longitudinal section area” means? Do you mean the surface area of the cells as estimated from (microscopic) images? How were the images obtained? I could not understand the point of showing “longitudinal section area”. I think it would be more useful to show cell capacitance, both the raw capacity transients and the estimated capacitance.
3. The AP data (for pig and human) should be supplemented with data on the resting membrane potentials (RMPs) and I would also suggest adding (if available) a series of APs recorded continuously to Figure 2, in order to illustrate the stability of the RMPs.
4. Since the authors have (appropriately) used Ba²⁺ for the estimation of Ik1 (and IK,ACh?), I think it would be extremely useful to assess and present data on the membrane resistance in the presence of Ba²⁺ (estimated between -90 and -60 mV). This is critical data to give a sense of the seal tightness.
5. The authors use an appropriate current-clamp protocol to generate APs. It would be very useful to illustrate a typical series of current injections. I note that the APs shown do not show a typical “take-off” point. A better example might be considered.
6. How much current (charge) injection was typically needed to evoke APs? How did this differ between human and swine CMs.
7. I wonder about the human CM data. The RMPs for the sample data are very negative and this seems unlikely to be representative. A continuous recording (requested above already) of a series of APs would be helpful for these CMs, along with RMP data and membrane resistance.
8. CMs from PSCs are typically spontaneous. How did this impact on the 2 Hz recordings?

Reviewer #3 (Remarks to the Author):

This manuscript by Seibertz et. al. describes the use of the Nanion Syncropatch 384 for recording currents from cardiomyocytes derived from pig hearts or from human iPSCs. The Syncropatch has been used extensively for automated patch clamp (APC), but there has been limited use for these harder to manipulate cells. The manuscript provides proof-of-concept data that APC can be applied for these cell types, and is novel and impactful in that respect. Successful deployment of this technology in native or native-like cells will be beneficial for high-throughput drug screening for safety (e.g. hERG) or efficacy. Although I am enthusiastic about the manuscript, my main concern is the somewhat hyperbolic claims made, which suggest that this technology is revolutionary and will allow rapid screening of hundreds of thousands of cells, and the far more modest data provided, which suggest a more incremental benefit.

1. The main concern is with reproducibility of the system. There are often problems with APC systems over repeated use, including plugging of holes, variability in recordings between different batches of cells, degradation of the borosilicate substrate, degradation in the quality of recording over time, etc. There is nothing in the manuscript to address reproducibility or any of these issues. At the very least, the authors should provide measures of reproducibility between different runs on the same day as well as runs on at least 3 different days using several different CM isolations.

2. Showing a single exemplar trace for a high-throughput system does not seem very convincing in Fig. 2 or S5. In Fig. 2 are the separate traces even from the same cell? Again, when collecting data from multiple trials, the authors should show full traces from multiple different cells and different plates, to allow an evaluation of reproducibility. Please also quantify how effective the system is for performing all four protocols listed on single cells. I imagine there will be some failure rate or degradation in quality, and it would be useful to quantify or visualize this.

3. Please also provide figures depicting stability in seal quality (e.g. series resistance, cell capacitance) across the various solution changes for the protocol in experiment 2. Again, to support the claims made in the paper, it would be necessary to show that these are reproducible across different experiments on the same day, across different plates, and across experiments performed on different days. A common problem with cardiomyocyte patch clamp is loss of seal quality with repeated solution changes. It would be very helpful to show how much better this system is compared to traditional electrophysiology for these changes.

4. Given the major likely application for this system will be in high-throughput drug screening, it would be useful to provide some data showing reproducible measurement of a dose-response drug effect on cardiomyocytes, perhaps something simple like a calcium or potassium channel blocker.

5. There are many experimental details missing:

- are cardiomyocytes from the same animal used in multiple runs during the day, or does each run require a fresh isolation?
- are plates reused? If so, how are they treated between runs?
- if plates are not reused, how reproducible are recordings across different plates using cardiomyocytes from the same animal? Are there substantial differences in failure rates?
- how is successful solution exchange measured, and, since there is no laminar flow microfluidics to control this, how complete and reproducible are automated solution exchanges? This strikes me as a particular concern in using microwells, as the more complete a solution exchange will be, the more likely it will jostle a cardiomyocyte and damage the seal.
- for the data shown in Fig. 2 and S5, is data from a single experiment/plate/isolation? Across multiple trials?

6. In general, though the Syncropatch 384 may ultimately prove quite useful for high-throughput voltage-clamping of cardiomyocytes and other primary cells, the data presented in this manuscript is far more modest and preliminary. I would suggest toning down the manuscript in terms of claims made for this system. None of the data displayed actually shows high-throughput data collection, for example (it's more in the moderate-range throughput), nor detailed range of mechanistic studies.

7. A fairly minor issue, but if available, for those of us who are electrophysiologists it would be very neat to see the membrane resistance versus time trace as a cell forms a gigaOhm seal and the automated suction causes cell break-in.

Response to comments of Reviewer 1

In this manuscript, the authors present patch clamp data from native cardiomyocytes that were obtained with their SyncroPatch 384 (Nanion Technologies, Munich, Germany) automated patch clamp apparatus that was modified to facilitate recordings from native cardiomyocytes in addition to small and round human induced pluripotent stem cell derived cardiomyocytes (hiPSC-CMs).

OVERALL IMPRESSION

I read this manuscript on automated patch-clamp of native cardiomyocytes with great interest. Unfortunately, it read like a salesperson presentation of Nanion Technologies rather than a scientific paper. It is up to the Editorial Board members of Communications Biology whether the journal is willing to function as a platform for promoting specific commercial products.

As an interested reader, I was constantly skipping back and forth between the main manuscript and its supplement, demonstrating that parts of the supplement should be included in the main manuscript. If the word count and/or number of figures of the main manuscript are limited, which I did not check, the authors should consider to fuse the two figures of the main manuscript and the five of the supplement into only a few figures that contain the essential information, focusing on the actual patch clamp recordings. In its current form, the seven figures look like parts of a PowerPoint presentation with many non-essential panels.

We thank the reviewer for his/her evaluation of our manuscript and the constructive suggestions. A point-by-point response is detailed below.

We are extremely enthusiastic about the possibilities of such an automated patch clamp device for native cardiomyocyte measurement. This has been a hitherto unachievable benchmark for the entire 20 year history of APC technology. In an effort to publish our first proof of concept results, our enthusiasm may have come across as too strong, with our wording too overstretched so it sounds like sales-person pitch. Though we still fervently believe that this new development deserves to be shared, we have critically reviewed the manuscript and removed many instances of extreme language or out of context hyperbole in a way to convey our message in a more neutral fashion. Many of the detailed and careful comments from this reviewer helped us in this regard.

We have also extensively re-worked the structure of the manuscript, its supplements and its figures in order to more coherently convey our message based on this reviewers recommendation. Much of the supplement has now been added to the main manuscript. Our changes will be detailed in our following responses.

SPECIFIC COMMENTS

1. Abstract, lines 33–34: “Automated patch-clamp (APC) approaches are experimenter independent and offer high-throughput, (...)” . In my experience, detailed electrophysiological data require time-consuming analysis by a patch clamp expert to avoid misinterpretation, thus seriously limiting the throughput and not being “experimenter independent”.

We agree with this reviewer that this sentence does not take the potentially complex nature of electrophysiological data into account. With high throughput systems such as ours, commercial software (DataControl 384) provides an easy to use interface to analyze multiple data parameters in a batch-based manner. The user can automatically acquire peak current, capacitance, inactivation time constants and much more information about all viable cells in minutes with minimal clicks. Once the data is acquired, minimal amounts of manual data handling is required to isolate recordings of interest, and perform the relevant statistical tests. The basic training needed for such exercises is minimal, and on par with any other branch of academic work. We believe this workflow, innate to APC, is substantially more user independent than manual patch-clamp. Nonetheless, we have included the following sentence in the revised version of the manuscript.

“APC systems are operated in a more user-independent manner. Their relatively simple mode of operation facilitates the wide application of APC in many laboratories and hospitals. This more unbiased approach also increases data quality and reproducibility” (page 17, line 367).

In addition, in the revised discussion section, we explore the budding possibilities offered by machine learning techniques. Algorithms which create convoluted neural networks to recognise deep patterns within data sets are currently being used clinically to read and successfully recognise and diagnose patients based only on their ECG trace. Similar machine learning capabilities could be relatively easily applied to data from high throughput methods such as APC. This would further streamline data analysis and contribute to a more experimenter independent work flow. The following paragraph has been added to the discussion.

“[...] the application of APC systems for comprehensive patient-specific characterization of cellular electrophysiology will represent an important step to realistically achieve the concept of personalized medicine. This could be particularly powerful when paired with artificial intelligence (AI) deep learning assemblies. Previous studies have implemented AI networks to predictively and precisely categorise patients based on miniscule patterns and repetitions within their clinical ECG. Similar learning methods applied to high throughput data concerning human cardiac ionic activity could aid tremendously in highly sensitive patient-specific diagnosis and therapeutic treatment regimes.” (page 18, line 393).

2. Abstract, lines 84–86: “Native cardiomyocytes were freshly isolated from pig hearts as described previously [6] and in the Methods section (Supplementary Information), resulting in an average cell yield of 6,000 cells per ml.” How was this “average cell yield” determined and how large was its variation between isolations?

This point has now been addressed and clarified in the substantial additions to the revised methods section. A representative fraction of cells were counted manually and then this number extrapolated to account for all cells in solution as per our previously published cell isolation and counting protocols^{1,2}. This is now stated in the revised methods section, as well as the variation in cell number between animals.

“Following centrifugation (90 g, 7 min, 37 °C) and resuspension of the pellet in storage solution, the viable cardiomyocytes were manually counted under bright-field conditions.” (page 5, line 102).

“Native cardiomyocyte isolation produced 8790±1610 viable atrial and 7200±3903 viable ventricular cardiomyocytes per isolated heart (n=3).” (page 11, line 220).

3. Main text, lines 95–96: “The same cardiomyocytes remained attached to the patch clamp aperture throughout the experimental run.” So, why does the number of myocytes differ between parts of the experimental run shown in Figure 2C? As an experimenter, I would like to know how many cells (absolute and percentage) survive the entire protocol and how many of these provided useful experimental data.

We thank this reviewer for pointing out this lack of information about our multi-current protocol. In response to this comment, we have added a major figure to the manuscript to make this data completely transparent (Figure 6 below). We have also detailed the amount of cellular survival in a new paragraph within the results section. We have now made clear that other, larger data sets, not included in our multi current protocol, contribute to our data for individual currents.

Figure 6. Overview of the multi-current Calcium, Action Potential and inward rectifier (CAPER) protocol in native atrial and ventricular cardiomyocytes (CM) using automated patch-clamp. **A**, Schematic of the cardiomyocyte-aperture interface with corresponding external (blue) and internal (orange) solutions. The substitution of which allows for multi-current acquisition from a single cell. **B**, Shares of total CM numbers that showed successful measurements of any of the three parameters of interest (L-type calcium current ($I_{Ca,L}$), action potential (AP) duration at 90% repolarization (APD_{90}) and basal inward rectifier current (I_{K1}). The central total indicates the successful cohort of atrial and ventricular (Vent.) CM in which all three parameters were successfully acquired. **C**, A detailed overview of the CAPER protocol including sequential internal and/or external solution exchange during a single experimental run. The acetylcholine activated inward rectifier current ($I_{K,ACh}$) is an optional addition to the protocol prior to inward rectifier block with $BaCl_2$. **D**, Three-dimensional visualization of $I_{Ca,L}$, APD_{90} and I_{K1} relationships in atrial (left) and ventricular (right) CM. APD_{90} is expressed as a shadow spectrum where darker colours indicate longer AP duration. **E**, Time course of membrane capacitance (upper) and series resistance (R_{series} ; lower) from a single plate ($n=5$ CM) over a full CAPER run. **F**, Direct acquisition software screenshots from a single plate (single animal) showing three complete electrophysiological measurements from single cells.

“This proof of concept CAPER protocol was moderately successful, with 25% of successfully patched atrial cells providing traces for all three parameters, along with 24% of successful ventricular cells (Figure 6B). Time traces of cell capacitance and R_{series} from a representative plate reveal stable values throughout the CAPER protocol even with a slight decrease in series resistance after 13 minutes, possibly due to increased leak of repolarising currents in the presence of high external K^+ . (Figure 6E). Similar stability and current quality was observed in all successful experiments across different days (Figure 6F, Supplementary Figure 7). In the event of failure, an increase in R_{series} could be observed as seal quality degraded. Cells that failed $I_{Ca,L}$ and AP assays sometimes showed clear inward rectifier currents. We assume this is due to the opposite phenomenon, where a suboptimal seal gains stability over time, possibly also due to a change in reversal potential and resulting cellular stability.” (page 15, line 323)

4. Main text, lines 109–112: “By quantifying a broad range of electrophysiological parameters we furthermore demonstrate that APC is applicable for a detailed range of mechanistic studies using primary cardiac cellular material to suit the specific needs of the experimenter.” This reads as an overstatement, given that the obtained data (Figure 2; Main text, lines 98–105) are limited to some general electrophysiological properties of atrial and ventricular myocytes. As an experimenter, I would like to know how well detailed patch clamp data, like current-voltage relationships, steady-state (in)activation curves and time constants of (in)activation of one or more individual membrane currents, can be achieved. According to the “Automated patch-clamp recordings” subsection of the Methods section of the Supplementary Information, at least the current-voltage relationship of the L-type calcium current was measured using the APC technique “by altering the test pulse from -40 mV by 5 mV every sweep including a final pulse of 60 mV”. However, this current-voltage relationship is neither shown in the manuscript nor in its supplement.

We thank the reviewer for this comment. As we believe this technology will be useful in future phenotyping studies, we therefore are very motivated to live up to our claim of it being able to provide a ‘detailed’ range of measurements. We have substantially increased our data sets with multiple new experiments, in which we can quantify IV relationships, activation and inactivation curves as well as time constants for $I_{Ca,L}$ decay and time to peak. This is shown in the new Figure 3 below.

Automated patch-clamp of native cardiomyocytes
Commun Biol COMMSBIO-21-3514-A

Figure 3. L-type calcium current ($I_{Ca,L}$) acquisition and analysis from native atrial and ventricular cardiomyocytes (CM) using automated patch-clamp. **A**, Schematic of swine cardiac tissue harvesting and CM isolation. **B**, Representative recordings of $I_{Ca,L}$ in atrial (upper) and ventricular (vent.; lower) CM. Inset: Voltage protocol (upper right). **C**, Peak $I_{Ca,L}$ density measured at +10 mV. **D**, Current-voltage (I - V) relationship curves for $I_{Ca,L}$ in atrial ($n=71$) and ventricular ($n=26$) CM. **E**, Right: Voltage protocols for I - V /activation experiments (upper) and S1/S2 voltage protocols for $I_{Ca,L}$ inactivation (lower). Left: $I_{Ca,L}$ activation, measured as G/G_{max} in atrial ($n=71$) and ventricular ($n=26$) CM, with corresponding inactivation curves (I/I_{max}) in atrial ($n=15$) and ventricular ($n=6$) CM. **F**, Activation kinetics of $I_{Ca,L}$ expressed as time to peak amplitude. **F**, Activation kinetics of $I_{Ca,L}$ expressed as time to peak amplitude. **G**, Concentration dependent response to nifedipine (Nif) application in $I_{Ca,L}$ current (upper), time course of average $I_{Ca,L}$ from a single plate following nifedipine application (lower, $n=10$). **H**, Normalized concentration response curve of $I_{Ca,L}$ following nifedipine application in atrial and ventricular CM with corresponding half maximal effective concentration (EC_{50}). **I**, Biphasic inactivation

kinetics of $I_{Ca,L}$ expressed as fast decay (τ_{fast} ; upper) and slow decay (τ_{slow} , lower) Data are mean \pm SEM. * $P < 0.05$ vs ventricular. n =number of atrial (176) and ventricular (58) CM from 3 animals (B, E, F).

Our new methods are outlined in the revised methods section of the main manuscript, where we have also fixed our sloppy mistake of not differentiating between normal $I_{Ca,L}$ voltage protocols and those for the IV/activation relationship.

“L-type calcium currents ($I_{Ca,L}$) were measured at 0.5 Hz using a voltage-step protocol with a holding potential of -80 mV and a 100 ms ramp pulse to -40 mV. This is held for 20 s before a 100 ms test-pulse to +10 mV. The current voltage (I-V) relationship and activation curves were measured by altering the test pulse from -40 mV by 10 mV every sweep including a final pulse of 60 mV. Internal solution contained (in mM): 10 EGTA, 10 HEPES, 10 CsCl, 10 NaCl, 110 CsF, pH 7.2 (with CsOH). Bath solution contained (in mM): 10 HEPES, 140 NaCl, 5 glucose, 4 KCl, 2 $CaCl_2$, 1 $MgCl_2$, pH 7.4 (with KOH). $I_{Ca,L}$ inactivation was assessed with the application of an S1-S2 inactivation protocol consisting of a normal pulse (S1) to +10 mV, followed by a 2 second holding potential before a second +10 mV pulse (S2). The holding potential was altered from -40 mV to +10 mV during each sweep. The fast and slow time constants of the biphasic $I_{Ca,L}$ decay were measured by fitting two standard exponential functions to each recording and assessing the tau (τ) for each fit. For method comparison, we also performed $I_{Ca,L}$ measurements in freshly isolated cardiomyocytes using manual patch-clamp and application of the same original electrophysiological protocol (Supplementary Figure 3).” (page 7, line 138).

5. Main text, lines 129–132: “APC systems allow for consecutive measurements of APs and various membrane currents in the very same cell. This is due to the fact that the internal solution of APC platforms, which is directly connected to the cytosol, can be exchanged relatively easily.” As an experimenter, I would like to know whether perforated patch clamp has been tried. This is important because this is required for reliable detailed investigations of specific membrane currents like the aforementioned L-type calcium current.

Perforated patch has not been tried in these proof of concept experiments due to time constraints and indeed the scope and intention of this manuscript. Nonetheless, we acknowledge the comment of this reviewer and have added a sentence in the limitations component of the discussion section of the revised version of the manuscript.

“In addition, alternative approaches such as perforated patch clamp could be considered for future experiments” (page 15, line 413).

6. Figure 2. Panel C shows “ICa,L current density” and “Ba2+ sensitive IK1 and IK,ACh current densities”, but it is not entirely clear how these “densities” are defined. Is the “ICa,L current density” the (negative) peak at –10 mV? And are the “Ba2+ sensitive IK1 and IK,ACh current densities” determined at –100 mV?.

We thank this reviewer for pointing out these omissions. These have been clarified in the methods section of the revised version of the manuscript.

“L-type calcium currents ($I_{Ca,L}$) were measured at 0.5 Hz using a voltage-step protocol with a holding potential of -80 mV and a 100 ms ramp pulse to -40 mV. This is held for 20 s before a 100 ms test-pulse to +10 mV” (page 7, line 138).

“Here, an external KCl concentration of 20 mM was used to facilitate a positive shift in I_{K1} reversal potential and allows for larger current acquisition of the inward component of I_{K1} at -90 mV” (page 8, line 168).

“ $I_{Ca,L}$, I_{K1} and $I_{K,ACh}$ of atrial and ventricular native cardiomyocytes were ratioed to cell capacitance and expressed as current density”(page 9, line 207).

7. Figure 2. As an experimenter, I would like to see the actual recordings of panel A at a larger scale, so that the actual recordings can be appreciated. In its present form, it is difficult to make an estimate, but it seems that the reversal potential of the potassium currents differs between atrial and ventricular cardiomyocytes. If so, can you explain?

Following substantial changes to the layout of the figures, representative trace data is now larger and shown with appropriate clarity.

We also apologize for our sloppy choice of representative traces, which is partly responsible for a lower V_{rev} in the ventricular representative figure. In addition, the traces in the original manuscript already undergone Ba^{2+} correction, which could distort the precise characteristics of the rectification. After thorough review, we have screened our original ventricular traces and new data from our additional cohort. In the revised version of the manuscript we include a more accurate representation. In addition, the subsequent trace under full block conditions has also been provided in a new Supplemental Figure 5. The visual V_{rev} is now more comparable between subtypes, and indeed fits calculation of the Nernst potential. Under the conditions we used, the reversal potential is solely determined by the internal and external K^+ concentrations following the Nernst Equation shown below.

$$E_{K^+} = \frac{R \cdot T}{F \cdot z} \cdot \ln \frac{[K^+]_o}{[K^+]_i}$$

$$E_{K^+} = \frac{8.314 \text{ J} \cdot 294.15 \text{ K}}{96485 \text{ C}} \cdot \ln \frac{20 \text{ mM}}{110 \text{ mM}}$$

$$E_{K^+} = 43 \text{ mV}$$

The changes outlined here are incorporated in Figure 5 and Supplemental Figure 5 of the revised manuscript and shown below.

Figure 5. Inward rectifier acquisition from native atrial and ventricular cardiomyocytes (CM) using automated patch-clamp. **A**, Representative basal inward rectifier current (I_{K1}) in atrial (left) and ventricular (vent.; middle) with superimposed acetylcholine-activated inward rectifier ($I_{K,ACh}$) current following carbachol (CCh) application during a depolarizing ramp voltage protocol (right). **B**, Time course of a single plate with atrial ($n=10$; left) and ventricular ($n=11$; right) CM inward current at 90 mV during a typical experiment. Red arrow indicates peak $I_{K,ACh}$. **C**, Peak inward I_{K1} density measured at -90 mV. **D**, Peak inward $I_{K,ACh}$ density measured at -90 mV. **E**, Time course of membrane capacitance from a single plate (**B**, left) over various external solution changes. **F**, Time course of series resistance (R_{series}) from a single plate (**B**, left) over various external solution changes. **G**, Ratio of mean capacitance changes per plate between solution change 2 (S2) and solution change 3 (S3) over 3 separate experimental days (D1, D2, D3). **H**, Ratio of R_{series} between S2 and S3 over 3 separate experimental days. Data are mean \pm SEM. * $P < 0.05$ vs ventricular. n = number of atrial (151) and ventricular (143) CM from 3 animals (**C**, **D**).

Supplementary Figure 5. Features of inward rectifier currents measured in native atrial and ventricular cardiomyocytes (CM) using automated patch-clamp. *A*, Representative uncorrected basal inward rectifier current (I_{K1}) in atrial (left) and ventricular (vent.; right) with superimposed acetylcholine-activated inward rectifier ($I_{K,ACh}$) current following carbachol (CCh) application, and during full block with BaCl₂ application, all during a depolarizing ramp voltage protocol (lower). *B*, Cell capacitance. *C*, Membrane resistance, calculated through dividing the driving force of K⁺ (30 mV) by the absolute Ba²⁺ sensitive current at -90 mV. *D*, Detailed representative timecourse of CCh application and the eventual desensitization to a quasi steady-state (QSS). *E*, QSS current. Data are mean \pm SEM. * P <0.05 vs ventricular. n =number of atrial (151) and ventricular (143) CM from 3 animals (*B*, *C*; *E*).

8. Figure 2. Do the two rightmost time scale bars of panel A (also) apply to the voltage clamp protocol?

In the original manuscript, they did not. The voltage protocol presentation has been substantially updated and all time scales for both recordings and protocols are now clearly shown.

9. Supplementary Information, lines 64–66: “Enzyme digestion procedures that produce cleanly isolated cardiomyocyte populations with minimal debris and non-cardiomyocyte contaminants are crucial to the eventual success of the assay.” Did you take special steps to arrive at these “cleanly isolated cardiomyocyte populations” or was it sufficient to strictly adhere to the protocol that you refer to at lines 46–47 (“Clean cardiomyocyte isolation was carried out according to our previously published standard protocol [1]”)?

We thank the reviewer for pointing out this potentially confusing wording. Our aim was to highlight that “clean” preparations without any debris are of utmost importance for the success of the native cardiomyocyte assay. During isolation we feel that washing is important (see Voigt *et al.* JMCC 2015)² along with the size of strainer. In addition, the tissue must be as fresh as possible following isolation. Since these are standard procedures described elsewhere, and are included in our manuscript already, we have removed our particular focus on the word “clean” in the revised version of the manuscript.

10. Supplementary Information, lines 66–67: “Isolates used in this study contained 60 cells per 10 μ l of external solution.” Is this number, which is equivalent to the aforementioned “average cell yield of 6,000 cells per ml” an estimate? How did you arrive at this number?

The freshly isolated cells were manually counted and then this number extrapolated to account for all cells in solution as per normal cell counting protocols^{1,2}. To clarify this, we have removed this sentence containing “average cell yield” in the revised version of the manuscript. We also kindly refer the reader to our response to comment 2 from this reviewer.

11. Supplementary Information, lines 71–72: “PatchControl 384 (Nanion Technologies) software allowed for the digitalization and acquisition of data.” At which frequencies were the data acquired and digitized?

The following sentence has now been added to the methods section of the revised version of the manuscript.

“PatchControl 384 (Nanion Technologies) software allowed for the digitization and acquisition of data (Digitization: 10 kHz).” (page 6, line 116).

12. Supplementary Information, lines 72–73: “Native cardiomyocytes were analysed at room temperature (...).” It is only here that it is mentioned that the data were acquired at room temperature. This should (also) be mentioned in the main text. Recording at room temperature may be inherent to the SyncroPatch 384 apparatus. If so, it is a clear limitation that should be addressed in the main text.

We thank this reviewer for pointing out this very important point and apologize for our oversight in our methods section. The experiments in this work were all performed at room temperature to provide an efficient and clear overview of the feasibility of our protocols. We have now stated this clearly in the revised version of the methods section, and have addressed this as a limitation of our work in the discussion.

“All experiments were recorded using the SyncroPatch 384 (Nanion Technologies, Munich, Germany) at room temperature” (page 6, line 114).

“Further optimization is required in order increase the CAPER success rate, for example through increasing working temperature to more physiological temperatures while maintaining cell viability or selectively targeting the composition of extracellular and intracellular solutions to enhance seal quality” (page 19, line 409).

13. Supplementary Information, lines 74–78: “Seal resistance, series resistance and cell capacitance were measured from each well via test pulse application and were continuously recorded over the entire experiment, which involved sequential acquisition of all currents detailed below (Supplementary Figure 3).” What was the composition of these test pulses, at which time during the experiment were these applied, and were these test pulses changed during the experiment to account for the changes in the internal or external solutions?

Test pulse information, as well as readouts of seal resistance, series resistance and cell capacitance have now been included into a major figure (Figure 2) in the revised version of the manuscript. In addition, the following text has been added to the revised methods section.

Figure 2. Quality control of automated patch-clamp of native atrial and ventricular cardiomyocytes (Native CM). **A**, Representative 384-well APC chip partially filled with native CM shown as a screenshot of Nanion DataControl 384 software with unused areas shaded out. Green boxes represent successful whole cell configuration during L-type calcium current measurement. Red and Blue rings indicate atrial and ventricular CM partitions respectively. **B**, Cellular density-dependent optical attachment rates of native ventricular cardiomyocytes and

Automated patch-clamp of native cardiomyocytes

Commun Biol COMMSBIO-21-3514-A

human ventricular induced pluripotent stem cell derived cardiomyocytes (hiPSC-CM) onto the patch-clamp aperture. **C**, Z factor analysis of individual runs for assessment of reproducibility ($I_{Ca,L}$ currents). **D**, NPC-384T chip (upper) with a photomicrograph of a single well with locations of interest labelled (lower). **E**, Representative membrane resistance during seal formation (left) and membrane rupture (right) of native CM in the presence of a membrane test pulse (upper). **F**, Absolute success rates of native CM and hiPSC-CM used in this protocol. **G**, Photomicrograph of attached ventricular hiPSC-CM (upper) and ventricular native CM (lower). The patch-clamp aperture is obscured by the cell. **H**, Time course of membrane capacitance (upper) and series resistance (R_{series} ; lower) in hiPSC-CM (3 plates, same day) and native CM (3 plates, same day) over successive $I_{Ca,L}$ experiments. **I**, Cellular capacitance (upper) and R_{series} (lower) averages from single plates over 3 separate experimental days (D1, D2, D3) and therefore 3 animals. Data are mean \pm SEM. n = mean of 24 wells (**B**) or single plates (**C, F, H, I**).

“Series resistance (R_{series}) and cell capacitance were continuously measured from each well via test pulse application (10 ms negative square pulse from -20 mV to -30 mV) before each sweep of the relevant voltage/current protocol” (page 6, line 121).

14. Supplementary Information, lines 79–82: “L-type calcium currents ($I_{Ca,L}$) were measured at 0.5 Hz using a voltage-step protocol with a holding potential of -80 mV and a 100 ms ramp pulse to -40 mV followed by a 100 ms test-pulse to +10 mV. The I-V relationship was measured by altering the test pulse from -40 mV by 5 mV every sweep including a final pulse of 60 mV.” Was this protocol repeated to assess any changes in the outcome over time? In other words: is this calcium current stable over time?

Our protocols were not repeated to test for this since our primary aim is to show the feasibility of recording primary cardiomyocytes with the APC system. Calcium current appears stable over time. Over longer recording times in the realm of 30 minutes or more, we expect a rundown of $\sim 20\%^3$. In our longest $I_{Ca,L}$ measurements using pharmacology, no obvious rundown more than this was seen over a 5 minute continuous recording.

15. Supplementary Information, lines 105–108: “Acetylcholine-activated inwardly rectifying current ($I_{K,ACh}$) was identified using the same protocol as that for I_{K1} following the application of 2 μ M Carbachol, an M-receptor agonist, which selectively opens atrial specific $I_{K,ACh}$ channels. Both I_{K1} and $I_{K,ACh}$ were identified as current responsive to Ba^{2+} blockade (1 mM).” It would be helpful to show a typical example of the original current traces in the order of their recording, together with the current differences that were used to arrive at the I_{K1} and $I_{K,ACh}$ traces.

In order to better visualize the differences in inward current in the order of their recording, we have added a clear time course of inward current amplitude for a representative plate of atrial and ventricular cardiomyocytes. These graphs have been incorporated in Figure 5 of the revised version of the manuscript.

Figure 5. Inward rectifier acquisition from native atrial and ventricular cardiomyocytes (CM) using automated patch-clamp. **A**, Representative basal inward rectifier current (I_{K1}) in atrial (left) and ventricular (vent.; middle) with superimposed acetylcholine-activated inward rectifier ($I_{K,ACh}$) current following carbachol (CCh) application during a depolarizing ramp voltage protocol (right). **B**, Time course of a single plate with atrial ($n=10$; left) and ventricular ($n=11$; right) CM inward current at -90 mV during a typical experiment. Red arrow indicates peak $I_{K,ACh}$. **C**, Peak inward I_{K1} density measured at -90 mV. **D**, Peak inward $I_{K,ACh}$ density measured at -90 mV. **E**, Time course of membrane capacitance from a single plate (**B**, left) over various external solution changes. **F**, Time course of series resistance (R_{series}) from a single plate (**B**, left) over various external solution changes. **G**, Ratio of mean capacitance changes per plate between solution change 2 (S2) and solution change 3 (S3) over 3 separate experimental days (D1, D2, D3). **H**, Ratio of R_{series} between S2 and S3 over 3 separate experimental days. Data are mean \pm SEM. * $P < 0.05$ vs ventricular. n = number of atrial (151) and ventricular (143) CM from 3 animals (**C**, **D**).

16. Supplementary Figure 2. Panel E shows the “overall whole-cell success rates of native CM and hiPSC-CM used in this protocol”, but it is not explained how these “success rates” were defined and determined.

We thank this reviewer for their careful and detailed evaluation of our manuscript and apologize for our lack of method description. Success rates were quantified through the amount of cells that successfully reached whole cell configuration during an experiment, and through determination of the experimental Z factor, which is now detailed in the revised version of the manuscript.

Automated patch-clamp of native cardiomyocytes

Commun Biol COMMSBIO-21-3514-A

“Success rate was defined as the absolute population of wells able to reach whole cell configuration.” (page 6, line 120).

“Assay quality was quantified through R_{series} and cellular capacitance stability during a typical $I_{Ca,L}$ experiment, and also through a previously described reproducibility test (Z factor, Z') that has been designed to evaluate the success of a high performance assay performed on a single plate in the absence of pharmacological modulation. Z' is primarily used to assess how reproducible an assay technique is on a day to day, or assay to assay basis” (page 9, line 191).

Response to comments of Reviewer 2

The authors present automated patch-clamp (APC) recordings of cardiomyocytes (CMs) from swine and human PSCs. The innovation is the use of "newly developed APC plate format".

Assessment: This is an interesting and important technological advance. I believe the authors can improve the manuscript and enhance its (future) impact by providing some additional data and modifying the figures.

We thank the reviewer for his/her helpful comments and constructive critiques of our original manuscript that led us to substantially improve the revised manuscript. Detailed responses to individual points are provided below.

Specific comments

1. Figure 1 should explicitly show the dimensions of fixed well and this can be complemented by showing a higher magnification of a region of the fixed well plates in Figure 2 of the Supplement.

We thank this reviewer for this helpful and informative idea. We have now included a picture of a single, empty well into the respective figure (Figure 2) in the revised version of the manuscript.

Automated patch-clamp of native cardiomyocytes
Commun Biol COMMSBIO-21-3514-A

Figure 2. Quality control of automated patch-clamp of native atrial and ventricular cardiomyocytes (Native CM). **A**, Representative 384-well APC chip partially filled with native CM shown as a screenshot of Nanion DataControl 384 software with unused areas shaded out. Green boxes represent successful whole cell configuration during L-type calcium current measurement. Red and Blue rings indicate atrial and ventricular CM partitions respectively. **B**, Cellular density-dependent optical attachment rates of native ventricular cardiomyocytes and human ventricular induced pluripotent stem cell derived cardiomyocytes (hiPSC-CM) onto the patch-clamp aperture. **C**, Z factor analysis of individual runs for assessment of reproducibility ($I_{Ca,L}$ currents). **D**, NPC-384T chip (upper) with a photomicrograph of a single well with locations of interest labelled (lower). **E**, Representative membrane resistance during seal formation (left) and membrane rupture (right) of native CM in the presence of a membrane test pulse (upper). **F**, Absolute success rates of native CM and hiPSC-CM used in this protocol. **G**, Photomicrograph of attached ventricular hiPSC-CM (upper) and ventricular native CM (lower). The patch-clamp aperture is obscured by the cell. **H**, Time course of membrane capacitance (upper) and series resistance (R_{series} ; lower) in hiPSC-CM (3 plates, same day) and native CM (3 plates, same day) over successive $I_{Ca,L}$ experiments. **I**, Cellular capacitance (upper) and R_{series} (lower) averages

from single plates over 3 separate experimental days (D1, D2, D3) and therefore 3 animals. Data are mean±SEM. n= mean of 24 wells (B) or single plates (C, F, H, I).

2. I am not sure I know what "longitudinal section area" means? Do you mean the surface area of the cells as estimated from (microscopic) images? How were the images obtained? I could not understand the point of showing "longitudinal section area". I think it would be more useful to show cell capacitance, both the raw capacity transients and the estimated capacitance.

As recommended by this reviewer, the results showing longitudinal section area, a proxy for "size", have been removed in the revised version of the manuscript. Indeed, it is not related to cellular electrophysiology which is the key focus of this work. Throughout experiments, a test pulse was applied throughout experimental procedures to give a continuous readout of cell capacitance, as stated in the response to comment 13 from reviewer 1. An example of a capacity transient been included in the updated Figure 2. Average cellular capacitance across multiple experiments and multiple animals/days has also been reported as per this reviewer's suggestion in Figure 2 (shown above).

3. The AP data (for pig and human) should be supplemented with data on the resting membrane potentials (RMPs) and I would also suggest adding (if available) a series of APs recorded continuously to Figure 2, in order to illustrate the stability of the RMPs.

We have included data on all resting membrane potentials to the new Figure 4. We have also taken this reviewers suggestions into account and, after new experiments, are able to provide representative traces of successive action potentials during 1 Hz stimulation obtained from both atrial and ventricular cardiomyocytes. These changes are shown in the revised figure below.

Figure 4. Action potential acquisition from native atrial and ventricular cardiomyocytes (CM) using automated patch-clamp. **A**, Representative traces of atrial (left) and ventricular (vent.; right) triggered action potentials during successive increases in pulse current injection. Insets: current protocol. **B**, Current at which action potential take off was first observed. **C**, Action potential duration at 50% repolarization (APD_{50}). **D**, Resting membrane potential (RMP) quantification. **E**, Representative traces of AP trains at 1 Hz showing stable RMP between pulses. Data are mean \pm SEM. * $P < 0.05$ vs ventricular. n =number of atrial (127) and ventricular (45) CM from 3 animals (**C**, **D**; **F**).

4. Since the authors have (appropriately) used Ba^{2+} for the estimation of I_{K1} (and $I_{K,ACH}$?), I think it would be extremely useful to assess and present data on the membrane resistance in the presence of Ba^{2+} (estimated between -90 and -60 mV). This is critical data to give a sense of the seal tightness.

In the main manuscript, the plotted curves were visualized following Ba^{2+} correction. This is still the case in the revised Figure 5, however we have included a new supplemental figure in which this correction has been removed and a new representative trace of full block with Ba^{2+} has been included for the readers interest. This shows the absolute current densities measured during the ramp pulse under different external conditions. As suggested by this

reviewer, we have calculated membrane resistance by dividing the driving force of K^+ (30 mV) by the absolute Ba^{2+} sensitive current at -90 mV. This is also been plotted in Supplemental Figure 5 which is shown below.

Supplementary Figure 5. Features of inward rectifier currents measured in native atrial and ventricular cardiomyocytes (CM) using automated patch-clamp. **A**, Representative uncorrected basal inward rectifier current (I_{K1}) in atrial (left) and ventricular (vent.; right) with superimposed acetylcholine-activated inward rectifier ($I_{K,ACh}$) current following carbachol (CCh) application, and during full block with $BaCl_2$ application, all during a depolarizing ramp voltage protocol (lower). **B**, Cell capacitance. **C**, Membrane resistance, calculated through dividing the driving force of K^+ (30 mV) by the absolute Ba^{2+} sensitive current at -90 mV. **D**, Detailed representative timecourse of CCh application and the eventual desensitization to a quasi steady-state (QSS). **E**, QSS current. Data are mean \pm SEM. * $P < 0.05$ vs ventricular. n =number of atrial (151) and ventricular (143) CM from 3 animals (**B**, **C**; **E**).

5. The authors use an appropriate current-clamp protocol to generate APs. It would be very useful to illustrate a typical series of current injections. I note that the APs shown do not show a typical "take-off" point. A better example might be considered.

We thank this reviewer for another insightful piece of advice. Our new Figure 4 has been substantially modified based on the suggestions from this reviewer. A trace of current injections and their resulting effects on membrane voltage leading to a typical AP take-off have been added to Figure 4 in the revised version of the manuscript.

Figure 4. Action potential acquisition from native atrial and ventricular cardiomyocytes (CM) using automated patch-clamp. **A**, Representative traces of atrial (left) and ventricular (vent.; right) triggered action potentials during successive increases in pulse current injection. Insets: current protocol. **B**, Current at which action potential take off was first observed. **C**, Action potential duration at 50% repolarization (APD₅₀). **D**, Resting membrane potential (RMP) quantification. **E**, Representative traces of AP trains at 1 Hz showing stable RMP between pulses. Data are mean±SEM. *P<0.05 vs ventricular. n=number of atrial (127) and ventricular (45) CM from 3 animals (C, D; F).

6. How much current (charge) injection was typically needed to evoke APs? How did this differ between human and swine CMs.

This data, along with the RMP has been included in the results section of the revised version of the manuscript for native cardiomyocytes.

“Injected current (Atrial: -3.5 ± 0.7 , Ventricular: -12.8 ± 2.6 pA/pF) was administered independently to each well as required.” (page 8, line 160).

7. I wonder about the human CM data. The RMPs for the sample data are very negative and this seems unlikely to be representative. A continuous recording (requested above already) of a series of APs would be helpful for these CMs, along with RMP data and membrane resistance.

Indeed, hyperpolarizing currents were injected in order to generate stable APs. We have relegated most information about hiPSC-CM to the supplements as this was not the key focus of this work. The following information about hyperpolarizing currents has been added to the the supplementary material for the human iPSC-CM.

“injected current (-9.7 ± 1.5 pA/pF) was administered independently to each well as required.” (Supplementary material, page 12).

Supplementary Figure 4. Automated patch-clamp (APC) of human atrial and ventricular induced pluripotent stem cell-derived cardiomyocytes (hiPSC-CM). **A**, Schematic of human dermal fibroblast harvesting, reprogramming into induced pluripotent stem cells (hiPSC) and subsequent differentiation into ventricular (Vent.) and atrial cardiomyocytes with the latter receiving retinoic acid (RetA) early in differentiation for atrial lineage confirmation. **B**, Representative action potential (AP; left) and L-type calcium current ($I_{Ca,L}$; right) recorded from atrial hiPSC-CM. insets: current protocol for AP acquisition and voltage protocol for $I_{Ca,L}$ acquisition. **C**, Representative AP (left) and $I_{Ca,L}$ (right) recorded from ventricular hiPSC-CM. **D**, AP duration at 90% repolarization (APD_{90} ; left), current threshold which first elicited an AP response (center) and resting membrane potential (RMP; right; ; $n = 108$ atrial vs 35 ventricular). **E**, $I_{Ca,L}$ current density ($n = 9$ atrial vs 26 ventricular; right). **F**, Representative current-voltage ($I-V$) relationship curves for $I_{Ca,L}$ in atrial and ventricular CM. Data are mean \pm SEM. * $P < 0.05$ vs ventricular.

As human iPSC-CM have been thoroughly characterized using APC previously⁴, we have not included a comprehensive analysis of these cells in the current manuscript. We feel this is not particularly new information and our key focus remains on reporting our success with native cardiomyocytes.

8. CMs from PSCs are typically spontaneous. How did this impact on the 2 Hz recordings?

Indeed, spontaneous beating of our hiPSC-CM is typically lower than 2 Hz when in 2 dimensional monolayers. For this reviewer only, we have included an overview of spontaneous beating data in iPSC-CM. As 2 Hz external stimulation will override any inherent pacemaking present, we feel that spontaneous activity will not have a major impact. We must also state again that hiPSC-CM characterization using APC have been previously documented (please see our response to comment 7 from this reviewer) and is not the focus of our present work.

Reviewer Figure 1. Spontaneous beating activity in early (<d50) and late (>d50) ventricular human induced pluripotent stem cell-derived cardiomyocytes (hiPSC-CM). n/N = hiPSC-CM/batch.

Response to comments of Reviewer 3

This manuscript by Seibertz et. al. describes the use of the Nanion Syncropatch 384 for recording currents from cardiomyocytes derived from pig hearts or from human iPSCs. The Syncropatch has been used extensively for automated patch clamp (APC), but there has been limited use for these harder to manipulate cells. The manuscript provides proof-of-concept data that APC can be applied for these cell types, and is novel and impactful in that respect. Successful deployment of this technology in native or native-like cells will be beneficial for high-throughput drug screening for safety (e.g. hERG) or efficacy. Although I am enthusiastic about the manuscript, my main concern is the somewhat hyperbolic claims made, which suggest that this technology is revolutionary and will allow rapid screening of hundreds of thousands of cells, and the far more modest data provided, which suggest a more incremental benefit.

We thank the reviewer for his/her helpful comments and constructive critiques of our original manuscript that led us to substantially improve the revised manuscript. Detailed responses to individual points are provided below.

Firstly, we share the enthusiasm of this reviewer for this method and agree that our language in the original manuscript could be considered hyperbolic. We have critically reviewed the entire manuscript and corrected instances of flowery language with a more realistic and informative alternative.

1. The main concern is with reproducibility of the system. There are often problems with APC systems over repeated use, including plugging of holes, variability in recordings between different batches of cells, degradation of the borosilicate substrate, degradation in the quality of recording over time, etc. There is nothing in the manuscript to address reproducibility or any of these issues. At the very least, the authors should provide measures of reproducibility between different runs on the same day as well as runs on at least 3 different days using several different CM isolations.

We agree with this reviewer that reproducibility needs to be critically assessed. Therefore we have added a substantial amount of data to our existing data pool. We have looked at the variation in series resistance and average capacitance in atrial cells both in different plates on the same day, and over different days. This extensive analysis also includes a calculation of the Z factor, which provides an assessment of high throughput assay reproducibility. These values in totality are all reported in Swine CM and iPSC-CM in the revised Figure 2 shown below.

Automated patch-clamp of native cardiomyocytes
Commun Biol COMMSBIO-21-3514-A

Figure 2. Quality control of automated patch-clamp of native atrial and ventricular cardiomyocytes (Native CM). **A**, Representative 384-well APC chip partially filled with native CM shown as a screenshot of Nanion DataControl 384 software with unused areas shaded out. Green boxes represent successful whole cell configuration during $I_{Ca,L}$ current measurement. Red and Blue rings indicate atrial and ventricular CM partitions respectively. **B**, Cellular density-dependent optical attachment rates of native ventricular cardiomyocytes and human ventricular induced pluripotent stem cell derived cardiomyocytes (hiPSC-CM) onto the patch-clamp aperture. **C**, Z factor analysis of individual runs for assessment of reproducibility ($I_{Ca,L}$ currents). **D**, NPC-384T chip (upper) with a photomicrograph of a single well with locations of interest labelled (lower). **E**, Representative membrane resistance during seal formation (left) and membrane rupture (right) of native CM in the presence of a membrane test pulse (upper). **F**, Absolute success rates of native CM and hiPSC-CM used in this protocol. **G**, Photomicrograph of attached ventricular hiPSC-CM (upper) and ventricular native CM (lower). The patch-clamp aperture is obscured by the cell. **H**, Time course of membrane capacitance (upper) and series resistance (R_{series} ; lower) in hiPSC-CM (3 plates, same day) and native CM (3 plates, same day) over successive $I_{Ca,L}$ experiments. **I**, Cellular capacitance (upper) and R_{series} (lower) averages

from single plates over 3 separate experimental days (D1, D2, D3) and therefore 3 animals. Data are mean±SEM. n= mean of 24 wells (B) or single plates (C, F, H, I).

2. Showing a single exemplar trace for a high-throughput system does not seem very convincing in Fig. 2 or S5. In Fig. 2 are the separate traces even from the same cell? Again, when collecting data from multiple trials, the authors should show full traces from multiple different cells and different plates, to allow an evaluation of reproducibility. Please also quantify how effective the system is for performing all four protocols listed on single cells. I imagine there will be some failure rate or degradation in quality, and it would be useful to quantify or visualize this.

We agree with these helpful comments and have indeed spent a considerable amount of effort to precisely quantify the success of our multi current protocol (now dubbed CAPER). We visualize both the success of CAPER through a new 3 dimensional analysis of cardiomyocyte electrophysiology and assess the extent of failure through numerical quantification. We have developed an entirely new figure in which these aspects are shown. In this figure, we have also included multiple examples of raw data screenshots from the same cell to respond to this reviewers suggestions.

Figure 6. Overview of the multi-current Calcium, Action Potential and inward rectifiER (CAPER) protocol in native atrial and ventricular cardiomyocytes (CM) using automated patch-clamp. A, Schematic of the cardiomyocyte-aperture interface with corresponding

Automated patch-clamp of native cardiomyocytes

Commun Biol COMMSBIO-21-3514-A

external (blue) and internal (orange) solutions. The substitution of which allows for multi-current acquisition from a single cell. B, Shares of total CM numbers that showed successful measurements of any of the three parameters of interest (L-type calcium current ($I_{Ca,L}$), action potential (AP) duration at 90% repolarization (APD_{90}) and basal inward rectifier current (I_{K1}). The central total indicates the successful cohort of atrial and ventricular (Vent.) CM in which all three parameters were successfully acquired. C, A detailed overview of the CAPER protocol including sequential internal and/or external solution exchange during a single experimental run. The acetylcholine activated inward rectifier current ($I_{K,ACh}$) is an optional addition to the protocol prior to inward rectifier block with $BaCl_2$. D, Three-dimensional visualization of $I_{Ca,L}$, APD_{90} and I_{K1} relationships in atrial (left) and ventricular (right) CM. APD_{90} is expressed as a shadow spectrum where darker colours indicate longer AP duration. E, Time course of membrane capacitance (upper) and series resistance (R_{series} ; lower) from a single plate ($n=5$ CM) over a full CAPER run. F, Direct acquisition software screenshots from a single plate (single animal) showing three complete electrophysiological measurements from single cells.

In the Supplemental material, there are similar screenshots of successful CAPER experiments over 3 different days to show that results are comparable between days and indeed, animals.

Automated patch-clamp of native cardiomyocytes
Commun Biol COMMSBIO-21-3514-A

Supplementary Figure 7. Screenshots over multiple days of the multi-current Calcium, Action Potential and inward rectifier (CAPER) protocol. Direct acquisition software screenshots from a single plate showing complete CAPER acquisition from multiple cells over three days (and therefore 3 animals).

3. Please also provide figures depicting stability in seal quality (e.g. series resistance, cell capacitance) across the various solution changes for the protocol in experiment 2. Again, to support the claims made in the paper, it would be necessary to show that these are reproducible across different experiments on the same day, across different plates, and across experiments performed on different days. A common problem with cardiomyocyte patch clamp is loss of seal quality with repeated solution changes. It would be very helpful to show how much better this system is compared to traditional electrophysiology for these changes.

We have dived deep into our recorded data to produce average results per plate of seal quality, defined through series resistance and cell capacitance stability. In our revised Figure 2, variation between experiments using different plates, animals and days is shown for all documented $I_{Ca,L}$ recordings.

Figure 2. Quality control of automated patch-clamp of native atrial and ventricular cardiomyocytes (Native CM). A, Representative 384-well APC chip partially filled with native CM shown as a screenshot of Nanion DataControl 384 software with unused areas shaded out. Green boxes represent successful whole cell configuration during L-type calcium current

measurement. Red and Blue rings indicate atrial and ventricular CM partitions respectively. **B**, Cellular density-dependent optical attachment rates of native ventricular cardiomyocytes and human ventricular induced pluripotent stem cell derived cardiomyocytes (hiPSC-CM) onto the patch-clamp aperture. **C**, Z factor analysis of individual runs for assessment of reproducibility ($I_{Ca,L}$ currents). **D**, NPC-384T chip (upper) with a photomicrograph of a single well with locations of interest labelled (lower). **E**, Representative membrane resistance during seal formation (left) and membrane rupture (right) of native CM in the presence of a membrane test pulse (upper). **F**, Absolute success rates of native CM and hiPSC-CM used in this protocol. **G**, Photomicrograph of attached ventricular hiPSC-CM (upper) and ventricular native CM (lower). The patch-clamp aperture is obscured by the cell. **H**, Time course of membrane capacitance (upper) and series resistance (R_{series} ; lower) in hiPSC-CM (3 plates, same day) and native CM (3 plates, same day) over successive $I_{Ca,L}$ experiments. **I**, Cellular capacitance (upper) and R_{series} (lower) averages from single plates over 3 separate experimental days (D1, D2, D3) and therefore 3 animals. Data are mean \pm SEM. n = mean of 24 wells (**B**) or single plates (**C**, **F**, **H**, **I**).

In addition, as we agree that solution change could have important effects on seal quality, we have examined the time courses of series resistance and cell capacitance during all inward rectifier experiments. By ratioing the different values before and after the 3rd solution addition, we see that seals appear to remain stable over these potentially disruptive events. Representative time courses and average values over 3 days are shown in the revised Figure 5.

Figure 5. Inward rectifier acquisition from native atrial and ventricular cardiomyocytes (CM) using automated patch-clamp. **A**, Representative basal inward rectifier current (I_{K1}) in atrial (left) and ventricular (vent.; middle) with superimposed acetylcholine-activated inward rectifier ($I_{K,ACh}$) current following carbachol (CCh) application during a depolarizing ramp

Automated patch-clamp of native cardiomyocytes

Commun Biol COMMSBIO-21-3514-A

voltage protocol (right). **B**, Time course of a single plate with atrial ($n=10$; left) and ventricular ($n=11$; right) CM inward current at 90 mV during a typical experiment. Red arrow indicates peak $I_{K,ACH}$. **C**, Peak inward I_{K1} density measured at -90 mV. **D**, Peak inward $I_{K,ACH}$ density measured at -90 mV. **E**, Time course of membrane capacitance from a single plate (B, left) over various external solution changes. **F**, Time course of series resistance (R_{series}) from a single plate (B, left) over various external solution changes. **G**, Ratio of mean capacitance changes per plate between solution change 2 (S2) and solution change 3 (S3) over 3 separate experimental days (D1, D2, D3). **H**, Ratio of R_{series} between S2 and S3 over 3 separate experimental days. Data are mean \pm SEM. * $P<0.05$ vs ventricular. n =number of atrial (151) and ventricular (143) CM from 3 animals (C, D).

4. Given the major likely application for this system will be in high-throughput drug screening, it would be useful to provide some data showing reproducible measurement of a dose-response drug effect on cardiomyocytes, perhaps something simple like a calcium or potassium channel blocker.

We agree with this reviewer and have chosen to examine the devices capability to discern the effects of calcium channel blocker Nifedipine on $I_{Ca,L}$. The APC system was able to report sensitive concentration dependent responses to nifedipine which is shown in the revised Figure 3. We feel this adequately demonstrates the capability of APC to increase the throughput of drug screening in primary cardiomyocytes.

Automated patch-clamp of native cardiomyocytes
Commun Biol COMMSBIO-21-3514-A

Figure 3. L-type calcium current ($I_{Ca,L}$) acquisition and analysis from native atrial and ventricular cardiomyocytes (CM) using automated patch-clamp. **A**, Schematic of swine cardiac tissue harvesting and CM isolation. **B**, Representative recordings of $I_{Ca,L}$ in atrial (upper) and ventricular (vent.; lower) CM. Inset: Voltage protocol (upper right). **C**, Peak $I_{Ca,L}$ density measured at +10 mV. **D**, Current-voltage (I - V) relationship curves for $I_{Ca,L}$ in atrial ($n=71$) and ventricular ($n=26$) CM. **E**, Right: Voltage protocols for I - V /activation experiments (upper) and S1/S2 voltage protocols for $I_{Ca,L}$ inactivation (lower). Left: $I_{Ca,L}$ activation, measured as G/G_{max} in atrial ($n=71$) and ventricular ($n=26$) CM, with corresponding inactivation curves (I/I_{max}) in atrial ($n=15$) and ventricular ($n=6$) CM. **F**, Activation kinetics of $I_{Ca,L}$ expressed as time to peak amplitude. **F**, Activation kinetics of $I_{Ca,L}$ expressed as time to peak amplitude. **G**, Concentration dependent response to nifedipine (Nif) application in $I_{Ca,L}$ current (upper), time course of average $I_{Ca,L}$ from a single plate following nifedipine application (lower, $n=10$). **H**, Normalized concentration response curve of $I_{Ca,L}$ following nifedipine application in atrial and ventricular CM with corresponding half maximal effective concentration (EC_{50}). **I**, Biphasic inactivation

kinetics of $I_{Ca,L}$ expressed as fast decay (τ_{fast} ; upper) and slow decay (τ_{slow} , lower) Data are mean \pm SEM. * $P < 0.05$ vs ventricular. n =number of atrial (176) and ventricular (58) CM from 3 animals (B, E, F).

5. There are many experimental details missing:

5.1. Are cardiomyocytes from the same animal used in multiple runs during the day, or does each run require a fresh isolation?

We have clarified this in the methods section of the revised version of the manuscript. One animal was used per day. Multiple runs were undertaken with a single animal.

“Cells were not kept in solution overnight, therefore every day of experimentation was conducted using cellular material from a different animal” (page 6, line 107).

5.2. Are plates reused? If so, how are they treated between runs? - if plates are not reused, how reproducible are recordings across different plates using cardiomyocytes from the same animal? Are there substantial differences in failure rates?

Plates are not reused between runs. We thank the reviewer for this and their previous questions about assessing reproducibility. We have added a substantial amount of experiments and analysis in order to quantify reproducibility between plates. We kindly refer the reader to our response to comment 3 of this reviewer.

5.3. How is successful solution exchange measured, and, since there is no laminar flow microfluidics to control this, how complete and reproducible are automated solution exchanges? This strikes me as a particular concern in using microwells, as the more complete a solution exchange will be, the more likely it will jostle a cardiomyocyte and damage the seal.

Solution exchanges are automated, however the system is designed to 1) always insert pipettes to the deepest area of the reservoir that it is withdrawing solutions from. This ensures that no air bubbles or aberrations are present in the liquid handling pipettes when full. Therefore solution deposits are standardized 2) extracellular solution is never fully exchanged, instead, the pipettes move into the 80 μ l well at a minimum depth to ensure they can remove 40 μ l with minimal cellular disruption. 3) In a similar fashion, when 40 μ l are subsequently added, the pipettes stay as far away from the cells as possible and deposit solution in a soft and standardized manner. These steps ensure perturbations to the cells are minimal. However, of course disruptions can occur, which we have examined in our response to comment 3 of this reviewer. We have clarified this in the revised methods section.

“Solution withdrawal and addition throughout all experiments is precisely calibrated so when removing or adding solution, the pipettes constantly keep the maximum distance possible from

the cells at the bottom of the APC chip. This limits potential cellular disruption and loss of seal integrity during solution exchanges.” (page 7, line 132).

5.4. For the data shown in Fig. 2 and S5, is data from a single experiment/plate/isolation? Across multiple trials?

The data shown in each figure has been substantially changed, updated and clarified in the revised manuscript. Data collected is from multiple plates over 3 experimental days. Plate number and variability are detailed in Figure 2.

Figure 2. Quality control of automated patch-clamp of native atrial and ventricular cardiomyocytes (Native CM). *A*, Representative 384-well APC chip partially filled with native CM shown as a screenshot of Nanion DataControl 384 software with unused areas shaded out. Green boxes represent successful whole cell configuration during L-type calcium current measurement. Red and Blue rings indicate atrial and ventricular CM partitions respectively. *B*, Cellular density-dependent optical attachment rates of native ventricular cardiomyocytes and

Automated patch-clamp of native cardiomyocytes

Commun Biol COMMSBIO-21-3514-A

human ventricular induced pluripotent stem cell derived cardiomyocytes (hiPSC-CM) onto the patch-clamp aperture. C, Z factor analysis of individual runs for assessment of reproducibility ($I_{Ca,L}$ currents). D, NPC-384T chip (upper) with a photomicrograph of a single well with locations of interest labelled (lower). E, Representative membrane resistance during seal formation (left) and membrane rupture (right) of native CM in the presence of a membrane test pulse (upper). F, Absolute success rates of native CM and hiPSC-CM used in this protocol. G, Photomicrograph of attached ventricular hiPSC-CM (upper) and ventricular native CM (lower). The patch-clamp aperture is obscured by the cell. H, Time course of membrane capacitance (upper) and series resistance (R_{series} ; lower) in hiPSC-CM (3 plates, same day) and native CM (3 plates, same day) over successive $I_{Ca,L}$ experiments. I, Cellular capacitance (upper) and R_{series} (lower) averages from single plates over 3 separate experimental days (D1, D2, D3) and therefore 3 animals. Data are mean \pm SEM. n = mean of 24 wells (B) or single plates (C, F, H, I).

6. In general, though the Syncropatch 384 may ultimately prove quite useful for high-throughput voltage-clamping of cardiomyocytes and other primary cells, the data presented in this manuscript is far more modest and preliminary. I would suggest toning down the manuscript in terms of claims made for this system. None of the data displayed actually shows high-throughput data collection, for example (it's more in the moderate-range throughput), nor detailed range of mechanistic studies.

Once again, we thank this reviewer for their kind evaluation of this proof of concept manuscript. The data shown here is indeed moderate range throughput, even though the automated patch clamp system by definition is easily able to scale this up following optimization. We have therefore refined our wording of these claims. We have made substantial additions and omissions in the revised version of the manuscript which we hope will present our data in a more realistic, yet still exciting manner. Please also see our revised limitations sections in the discussion (shown below) and our response to the introductory comment from this reviewer and from reviewer 1.

“As a proof of concept study, several limitations exist in our presently reported methodology. Our output cannot yet be considered truly high throughput, as our conservative partial plate utilization (max. 128 wells) restricts our data point sample size to a lower value than would be possible with a full 384 well plate” (page 18, line 403).

7. A fairly minor issue, but if available, for those of us who are electrophysiologists it would be very neat to see the membrane resistance versus time trace as a cell forms a gigaOhm seal and the automated suction causes cell break-in.

As passionate electrophysiologists ourselves, we agree and thank the reviewer for this small, albeit useful, idea. Traces of membrane resistance during seal formation and cell rupture are now included as a Supplementary Figure 2 of the revised version of the manuscript. This is shown below.

Automated patch-clamp of native cardiomyocytes
Commun Biol COMMSBIO-21-3514-A

Supplementary Figure 2. Representative time course of seal resistance (R_{Seal}) during cell addition, suction application, whole cell configuration and eventual experimentation. Before whole cell configuration is achieved, transiently elevated external Ca^{2+} levels (5 mM max) are thought to interact with fluoride in the internal solution to form a precipitate at the aperture-cell interface, fostering giga seal formation.

References

1. Voigt, N., Zhou, X.-B. & Dobrev, D. Isolation of human atrial myocytes for simultaneous measurements of Ca²⁺ transients and membrane currents. *J. Vis. Exp.* e50235–e50235 (2013). doi:10.3791/50235
2. Voigt, N., Pearman, C. M., Dobrev, D. & Dibb, K. M. Methods for isolating atrial cells from large mammals and humans. *J. Mol. Cell. Cardiol.* **86**, 187–198 (2015).
3. Kameyama, M., Kameyama, A., Takano, E. & Maki, M. Run-down of the cardiac L-type Ca²⁺ channel: partial restoration of channel activity in cell-free patches by calpastatin. *Pflugers Arch.* **435**, 344–349 (1998).
4. Obergrussberger, A. *et al.* The suitability of high throughput automated patch clamp for physiological applications.. *J. Physiol.* **600**, 277–297 (2021).

Reviewers' comments:

Reviewer #1 (Remarks to the Author):

SUMMARY

This manuscript is actually a de novo version of the manuscript that was submitted as a brief communication in December 2021. The main message of the manuscript is that the use of an automated patch clamp apparatus is not limited to recordings from stem cells (or similar cells). If used in combination with a newly developed plate, automated patch clamp recordings can be successfully made from native cardiomyocytes. The latter is the innovative factor of the manuscript. Due its de novo status, I reviewed the manuscript from scratch, in addition to critically reading the rebuttal letter. The authors have carefully addressed my initial comments. However, the large amount of new figures and text makes that I have a lot of (novel) specific comments, both regarding the main manuscript and the supplementary information. These specific comments are listed below.

OVERALL IMPRESSION

The manuscript has improved considerably. This is because the authors now focus on the actual patch clamp data obtained with their Nanion SyncroPatch 384 automated patch clamp apparatus. As a consequence, it reads much more like a scientific paper, and less like a salesperson presentation of Nanion Technologies, than the original manuscript.

SPECIFIC COMMENTS – Main Manuscript

[Line numbers refer to the Word source file]

Introduction, last line. Extend “freshly isolated mammalian cardiomyocytes” to “freshly isolated mammalian atrial and ventricular cardiomyocytes”.

Materials and Methods, lines 108–109: “Myocyte suspension was pipetted into 128 wells of the APC chip with 40 μ l per well.” Please explain (also) here why only 128 wells of the 384 available were used. I realize that this aspect is addressed in the Results section, where it is mentioned that “partial plates were utilized to conserve cellular numbers and allow for efficient tests of reproducibility”.

Materials and Methods, lines 113–114: “All experiments were recorded using the SyncroPatch 384 (Nanion Technologies, Munich, Germany) at room temperature.” From the rebuttal letter, I learned that the room temperature was 21°C. I would therefore write “at room temperature (21°C)”.

Materials and Methods, lines 115–116: “PatchControl 384 (Nanion Technologies) software allowed for the digitization and acquisition of data (Digitization: 10 kHz).” Is this digitization rate dependent on the number of wells in use?

Materials and Methods, lines 119–120: “Success rate was defined as the absolute population of wells able to reach whole cell configuration.” From the Results section, it seems that success rate was defined as the percentage of wells able to reach whole cell configuration rather than the “absolute population”. Furthermore, from the legend to Figure 2, it seems that “whole cell configuration” should be read as “whole cell configuration during L-type calcium current measurement”. Is this correct?

Materials and Methods, lines 122–124: “When starting an experiment, APC chips are loaded with the desired pipette solution and 30 μ l of a divalent-free solution containing (in mM): 10 HEPES, 140 NaCl, 4 KCl, 5 glucose.” Where exactly is this “divalent-free solution” located? Is the volume of 30 μ l for the entire chip or for a single well?

Materials and Methods, lines 137–139: "(...) with a holding potential of 80 mV and a 100 ms ramp pulse to -40 mV. This is held for 20 s before a 100 ms test-pulse to +10 mV." Should '20 s' perhaps read '20 ms'?

Materials and Methods, lines 146–148: "The fast and slow time constants of the biphasic I_{Ca,L} decay were measured by fitting two standard exponential functions to each recording and assessing the tau (τ) for each fit." Were these fits carried out over a range of test potentials or for a single specific test potential? Were the (relative) amplitudes of the fast and slow also analyzed?

Materials and Methods, lines 174–175: "I_{K,ACh} was defined as the CCh-dependent current increase of inward current at -90 mV." Is this I_{K,ACh} the initial current or that obtained after desensitization (see also below)? I realize that Figure 5 suggests that I_{K,ACh} is defined as the initial current.

Materials and Methods, lines 198–200: "The positive control used in this proof of concept study consisted of Chinese hamster ovary (CHO) stably expressing Nav1.5 (Charles River Laboratories) on the same plate." How were these CHO cells used? Was their peak sodium current measured? It should be explained which data of these cells were used as 'positive control' and how these data were acquired.

Materials and Methods, lines 207–209: "Exclusion criteria (quality control) included a seal resistance <100 M Ω , a peak current of <50 pA, and an R_{series} of 200 M Ω at 10 mV." Which 'peak current' are you referring to? Should this current not be 'ratioed to cell capacitance and expressed as current density'?

Materials and Methods, lines 209–210: "AP parameters such as resting membrane potential (RMP) and Action potential duration was analyzed offline using DataControl 384 software." Write 'were' instead of 'was'.

Materials and Methods, lines 210–212: "Using the AP search mode, action potentials that did not display a clear threshold take-off potential following increasing current stimuli were excluded." What is 'AP search mode'?

Materials and Methods, general. It seems that alle recordings (both APs and individual membrane currents) have been made with EGTA in the internal (pipette) solution. Have recordings been made without EGTA? If so, were these successful? If not, what was the rationale for using EGTA in all recordings?

Results, lines 225–228: "The patching success rate, defined as effective seal formation (>100 M Ω) and whole cell configuration, and achieved through gentle negative pressure application, was 13.9 \pm 1.7% in native cells (15 plates over 3 days) and showed no significant changes over successive experimental days." Figure 2A suggests that the success rate with atrial cells (red 'ring') is substantially higher than with ventricular cells (blue 'ring'). Has this been tested? What was the outcome?

Results, lines 235–237: "In addition, Z factor analysis, a marker for high throughput assay reproducibility, consistently showed good to excellent values of assay robustness and reproducibility in multiple plates of primary cells over different days (Figure 2C)." What is referred to as 'good' here, is 'moderately successful' according to the Materials and Methods section. Which is correct?

Results, lines 235–237: "Clear temporal characteristics of I_{Ca,L} activation and biphasic decay are also easily extracted (Figure 3F, I)." Were these data obtained at a test potential of +10 mV? What was the relative amplitude of the fast and slow components of decay? Were these different between atrial and ventricular cells?

Results, lines 264–266: “When normalized to $I_{Ca,L}$ amplitude at full pharmacological block, atrial myocytes showed an EC_{50} of 6.08 ± 1.14 nM and ventricular cardiomyocytes showed 3.41 ± 0.71 nM.” Do these EC_{50} values show a statistically significant difference? Were these EC_{50} values obtained with a Hill equation? If so, what were the associated nH values?

Results, lines 271–274: “Atrial myocytes showed a shorter action potential duration at 50% repolarization (APD₅₀) compared to ventricular (Figure 4C). This typical discrepancy in chamber specific phenotype was also replicated using the same APC method in hiPSC-CM generated using subtype specific differentiation protocols (Supplementary Figure 4).” However, Figure 4 shows action potential duration at 50% repolarization, whereas Supplementary Figure 4 shows action potential duration at 90% repolarization. It would be informative to show APD₅₀ and APD₉₀ in both Figure 4 and Supplementary Figure 4, or at least show either APD₅₀ or APD₉₀ in both Figure 4 and Supplementary Figure 4.

Results, lines 312–314: “In this small cohort, we observed a uniform increase of action potential duration at 90% repolarization (APD₉₀) as $I_{Ca,L}$ amplitude increases in atrial cardiomyocytes.” Looking at Figure 6D, I would not speak of ‘a uniform increase’.

Results, lines 328–329: “Similar stability and current quality was observed in all successful experiments across different days (Figure 6F, Supplementary Figure 7).” How does Figure 6F, which shows data from a single plate, confirm this statement?

Discussion, lines 337–338: “Numerous detailed electrophysiological parameters were able to be reproducibly extracted from separate cohorts of swine native cardiomyocyte tissue, (...).” I would write ‘isolated swine native cardiomyocytes’ instead of ‘swine native cardiomyocyte tissue’.

Discussion, lines 345–348: “After decades of failure to record from primary adult mammalian cardiomyocytes, we now report for the first time a highly versatile method for deep and user unbiased electrophysiological phenotyping of primary cardiac material.” This reads as an overstatement. In the past decades, successful recordings from primary adult mammalian cardiomyocytes have been carried out in numerous laboratories over the world. You should either remove or tone down this statement.

Discussion, lines 381–383: “In contrast, the corresponding pipette solution in a conventional patch-clamp system needs to be maintained throughout a complete experiment.” Successful attempts have been made to replace pipette solutions during an experiment. I would therefore not write that ‘the corresponding pipette solution in a conventional patch-clamp system needs to be maintained throughout a complete experiment’, but, with appropriate references, that exchanging the pipette solution during such experiment is highly complex (or something similar).

Discussion, lines 414–415: “In typical APC experiments, fluoride ions in the internal solution aid in seal formation by reacting with external Ca^{2+} and forming a precipitate around the aperture-cell interface.” Please provide references.

References, #18. ‘Maria G. Rotordam et al.’ should read ‘Rotordam, M. G. et al.’

References, #26 and #27. Will these application notes remain publicly available? Do they have a DOI?

Figure 1, legend. The legend to Figure 1 refers to panels A–D, but the actual figure has only labels A and B. The legend should be adapted to the figure (or vice versa).

Figure 2F and its legend. How is ‘absolute success rate’ defined?

Figure 2H. Membrane capacitance and series resistance remain nicely constant over time when expressed as mean \pm SEM for an entire plate. However, is this also true for individual wells?

Figure 3, legend: "B, Representative recordings of $I_{Ca,L}$ in atrial (upper) and ventricular (vent.; lower) CM." Panel B shows "IM" (for 'membrane current?') rather than $I_{Ca,L}$. This also holds for other figures and panels. Panel B does not show "vent."

Figure 3, legend: "E, Right: Voltage protocols for I-V/activation experiments (upper) and S1/S2 voltage protocols for $I_{Ca,L}$ inactivation (lower). Left: $I_{Ca,L}$ activation, measured as G/G_{max} in atrial ($n=71$) and ventricular ($n=26$) CM, with corresponding inactivation curves (I/I_{max}) in atrial ($n=15$) and ventricular ($n=6$) CM." Voltage protocols are shown left and (in)activation curves right.

Figure 3H and its legend. How is 'normalized response' defined? Is this the percent peak $I_{Ca,L}$ density measured at +10 mV? How large are the numbers of atrial and ventricular myocytes?

Figure 3, legend, last line: " n =number of atrial (176) and ventricular (58) CM from 3 animals (B, E, F)." '(B, E, F)' should probably read '(C, F, I)'.

Figure 4, A and E. Given the APD50 values reported in panel C, I would not speak of 'representative traces' in panels A and E. In particular, the duration of the ventricular action potentials seems much longer than 'representative'. I suggest to replace panels A and E with more representative ones.

Figure 4D. I am seriously concerned by the 'resting membrane potential (RMP) quantification' of Figure 4D. It seems that the resting membrane potential of atrial CMs can be as negative as -110 mV or even more negative. This is at odds with the potassium reversal potential of around -86 mV that can be computed from the 'internal' and 'external' solutions. How reliable are these RMP data? And what about other data? According to Figure 4C, the action potential duration at 50% repolarization (APD50) of both atrial and ventricular swine native cardiomyocytes can be as small as 5-10 ms. Again, how reliable are these data?

Figure 4, legend, last line: " n =number of atrial (127) and ventricular (45) CM from 3 animals (C, D; F)." '(C, D; F)' should probably read '(B, C, D)'.

Figure 5, legend: "B, Time course of a single plate with atrial ($n= 10$; left) and ventricular ($n= 11$; right) CM inward current at 90 mV during a typical experiment." The membrane potential of '90 mV' should probably read '-90 mV'.

Figure 5, legend: "E, Time course of membrane capacitance from a single plate (B, left) over various external solution changes. F, Time course of series resistance (R_{series}) from a single plate (B, left) over various external solution changes." Data are for a single plate. Are these data representative for other plates? How many cells were on this single plate?

Figure 6F. What are the 'numbers' along the vertical axes? Does '400p' (left panels) denote '400 pA' and '1n' (right panels) '1 nA'? And does '100m' (middle panels) denote '100 mV'?

SPECIFIC COMMENTS – Supplementary Information

[Line numbers refer to the pdf version]

Supplementary Information, general: It seems that some text is in dark grey, whereas other text is in pure black. Select one color and use it throughout.

Supplementary Information, line 6: "Rupamanjari Majumder, PhD, Sebastian Clauß, MD." Insert a space between "PhD" and "Sebastian".

Supplementary Information, lines 37-38: "Human induced pluripotent stem cell (hiPSC) line UMGi014-

C clone 14 (isWT1.14) were derived from the dermal fibroblasts of a healthy male donor." Replace "were" with "was".

Supplementary Information, lines 41–43: "hiPSC were cultured on 1:120 Matrigel™ (BD Biosciences) coated plates and maintained with Stem MACS IPS-Brew XF medium (Miltenyi Biotec) daily." Now that 'hiPSC' abbreviates 'human induced pluripotent stem cell' (line 37), write 'hiPSCs' instead of 'hiPSC'. Similarly, write 'hiPSC-CMs' instead of 'hiPSC-CM' at lines 59 and 60 as well as at other instances in the supplementary figures and their legends. Similarly, write 'AP' for action potential and 'APs' for action potentials. This holds for the main manuscript as well.

Supplementary Information, lines 62–66: "Cells were returned to 4 °C for 10 minutes and then exposed to pre-cooled HBSS (Thermo Fisher Scientific) at a volume to provide sufficient cellular density (>100,000 cells per ml; Supplementary Figure 2) and sparingly resuspended with a glass pipette in order to detach cells into isolated bodies." How does this statement relate to Supplementary Figure 2, which (only) shows a "representative time course of seal resistance"?

Supplementary Information, lines 69–70: "IK1 and CAPER experiments were not attempted." What are "IK1 and CAPER experiments"? Explain here.

Supplementary Information, lines 70–71: "For action potential acquisition, injected current (-9.7 ± 1.5 pA/pF) was administered independently to each well as required." Add duration of current pulse.

Supplementary Figure 1. Why would the 1,000 cardiomyocytes studied by a skilled operator yield 1,000 "data points" (ratio 1:1), while the 115,000 cardiomyocytes studied through automated patch clamp yield 350,000 "data points" (ratio 1:3)?

Supplementary Figure 2, legend: "Before whole cell configuration is achieved, transiently elevated external Ca^{2+} levels (5 mM max) are thought to interact with fluoride in the internal solution to form a precipitate at the aperture-cell interface, fostering giga seal formation." Please provide references.

Supplementary Figure 3, legend: "B, Representative L-type calcium current ($\text{I}_{\text{Ca,L}}$) recorded from atrial CM (upper) and ventricular (vent.; lower) CM." Panel B shows "IM" (for 'membrane current'?) rather than $\text{I}_{\text{Ca,L}}$. Panel B does not show "vent." This also applies to Supplementary Figure 4.

Supplementary Figure 3, legend: "C, $\text{I}_{\text{Ca,L}}$ current density (...)." Is " $\text{I}_{\text{Ca,L}}$ current density" defined as peak current density at +10 mV? This also applies to Supplementary Figure 4.

Supplementary Figure 4, legend: "B, Representative action potential (AP; left) and L-type calcium current ($\text{I}_{\text{Ca,L}}$; right) recorded from atrial hiPSC-CM. insets: current protocol for AP acquisition and voltage protocol for $\text{I}_{\text{Ca,L}}$ acquisition." Does panel B shows APs and membrane current from a single hiPSC-CM or from multiple hiPSC-CMs? Printed at 100% size, the insets are barely readable, also because the supplied figure is not sharp enough (both in the pdf and its Word format source file).

Supplementary Figure 4, legend: "G, Representative current-voltage (I-V) relationship curves for $\text{I}_{\text{Ca,L}}$ in atrial and ventricular CM." What are we looking at in panel G? Are these average data (without error bars) or data from one representative atrial hiPSC-CM (not just "CM") and one representative ventricular hiPSC-CM? How were the smooth I-V curves obtained?

Supplementary Figure 4. Panel D suggests that the resting membrane potential (RMP) of both atrial and ventricular hiPSC-CMs can be as negative as -95 mV. This seems at odds with the potassium reversal potential of around -86 mV that can be computed from the 'internal' and 'external' solutions and requires explanation.

Supplementary Figure 5, legend: "A, Representative uncorrected basal inward rectifier current (IK1) in

atrial (left) and ventricular (vent.; right) with superimposed acetylcholine-activated inward rectifier (IK,ACh) current following carbachol (CCh) application, and during full block with BaCl₂ application, all during a depolarizing ramp voltage protocol (lower).” Please clarify which of the three traces show the ‘uncorrected basal inward rectifier current (IK1)’ and the ‘superimposed acetylcholine-activated inward rectifier (IK,ACh) current’. Was the curve upon CCh application obtained after the “eventual desensitization” of panel D? Printed at 100% size, the smaller text is barely readable, also because the supplied figure is not sharp enough (both in the pdf and its Word format source file).

Supplementary Figure 5. Panels D and E suggest that the membrane capacitance of the hiPSC-CMs can be almost zero and the membrane resistance can actually be zero or even below. How reliable are these data? Are there any exclusion criteria with respect to membrane capacitance and/or membrane resistance?

Supplementary Figure 6, legend: “Screenshots of Supplementary Video 1.” The 15 screenshots of Supplementary Video 1 require a more detailed explanation than just these 5 words.

Supplementary Figure 6. The screenshots are so small that the text has become unreadable. This is not only because the supplied figure is not sharp enough, but also because the screenshots are simply far too small.

Supplementary Figure 7, legend: “Direct acquisition software screenshots from a single plate showing complete CAPER acquisition from multiple cells over three days (and therefore 3 animals).” This figure requires better explanation. What exactly are we looking at? Does the three times three panels per day show APs and membrane current recorded from three single hiPSC-CMs from the plate of that specific day? If so, how were these three hiPSC-CMs selected? And are these data from atrial or from ventricular hiPSC-CMs? You should consider the use of ‘Experiment 1’, ‘Experiment 2’, and ‘Experiment 3’ rather than the somewhat confusing ‘Day 1’, ‘Day 2’, and ‘Day 3’ (as if recordings were made from the same cells over three days).

Supplementary Video 1. The screenshots are so small that the text has become barely readable. This is not only because the supplied figure is not sharp enough, but also because the screenshots are simply too small. I would prefer a series of six 2D APD90 vs. I_{Ca,L} plots obtained at IK1 densities of 0, 1, 2, 3, 4, and 5 pA/pF, respectively, instead of, or in addition to, a single 3D surface plot.

Supplementary Video 1, legend: “An in silico surface plot of all three experimental CAPER parameters (L-type calcium current [I_{Ca,L}], action potential duration at 90% repolarization [APD90] and basal inward rectifier current [IK1]) was generated through a randomized array of 10,000 CMs.” It should be explained in more detail how these in silico data were obtained. How was this ‘randomized array of 10,000 CMs’ constructed? Were (only) I_{Ca,L} and IK1 varied? If so, were they independently varied and over which range? Did all 10,000 model cells generate an action potential (and thus an APD90) or did only a selection of the 10,000 model cells generate an action potential? How were the simulations carried out with respect to hardware, software, stimulus current, stimulation frequency, steady-state data, etc.? How are I_{Ca,L} and IK1 along the axes of the 3D surface plot expressed? Are these perhaps expressed as (peak) current at a specific membrane potential?

Supplementary Information, References. Use a consistent format for your references and provide the DOI for all three references (writing doi: 10.1016/j.xpro.2020.100026 instead of doi:https://doi.org/10.1016/j.xpro.2020.100026, etc.). Remove “Find the latest version : Deep phenotyping of human induced pluripotent stem cell – derived atrial and ventricular cardiomyocytes” from Reference #1.

Reviewer #2 (Remarks to the Author):

I believe the revised manuscript is very much improved and I have no further critical comments.

Reviewer #3 (Remarks to the Author):

Seibertz et. al. have provided a substantial amount of new data in their revised manuscript to address my concerns regarding reproducibility and data quality. The new version presents a more nuanced look at the technology and shows both the obvious strengths and potential drawbacks of the technique. I have no further major concerns that need to be addressed. Congratulations on a really nice manuscript!

I noted a couple of very small issues below (line numbers refer to manuscript without tracked changes):

- 1) Fig. 2E, missing scale bars for current traces
- 2) Fig. 3F, Line 595, caption is duplicated
- 3) Line 246, please clarify which numbers are APC and which are conventional patch

Response to comments of Reviewer 1

The manuscript has improved considerably. This is because the authors now focus on the actual patch clamp data obtained with their Nanion SyncroPatch 384 automated patch clamp apparatus. As a consequence, it reads much more like a scientific paper, and less like a salesperson presentation of Nanion Technologies, than the original manuscript.

We thank the reviewer for his/her comprehensive manuscript review, their multiple constructive suggestions and their eye for detail. We feel this is particularly important and has allowed us to further improve the manuscript. A point-by-point response is outlined below.

SPECIFIC COMMENTS

1. Introduction, last line. Extend “freshly isolated mammalian cardiomyocytes” to “freshly isolated mammalian atrial and ventricular cardiomyocytes”.

This has been corrected according to the reviewer’s suggestion.

2. Materials and Methods, lines 108–109: “Myocyte suspension was pipetted into 128 wells of the APC chip with 40 μ l per well.” Please explain (also) here why only 128 wells of the 384 available were used. I realize that this aspect is addressed in the Results section, where it is mentioned that “partial plates were utilized to conserve cellular numbers and allow for efficient tests of reproducibility”

The following text has also been added to the methods section.

“Myocyte suspension was pipetted into 128 wells of the APC chip with 40 μ l per well. This partial plate utilization concentrates cellular density over a smaller chip area, effectively increasing the chances of a successful experiment. Atrial and ventricular cardiomyocytes were investigated simultaneously on the same APC chip.” (page 6, line 108).

3. Materials and Methods, lines 113–114: “All experiments were recorded using the SyncroPatch 384 (Nanion Technologies, Munich, Germany) at room temperature.” From the rebuttal letter, I learned that the room temperature was 21°C. I would therefore write ‘at room temperature (21°C)’.

This has been corrected according to the reviewer’s suggestion.

4. Materials and Methods, lines 115–116: “PatchControl 384 (Nanion Technologies) software allowed for the digitization and acquisition of data (Digitization: 10 kHz).” Is this digitization rate dependent on the number of wells in use?

This is not rate dependent and is the temporal resolution offered by the 384 channel amplifier of the system. The following sentence has been added to the revised version of the manuscript.

"[...] software allowed for the digitization and acquisition of data (System digitization rate: 10 kHz)." (page 6, line 117).

5. Materials and Methods, lines 119–120: "Success rate was defined as the absolute population of wells able to reach whole cell configuration." From the Results section, it seems that success rate was defined as the percentage of wells able to reach whole cell configuration rather than the 'absolute population'. Furthermore, from the legend to Figure 2, it seems that 'whole cell configuration' should be read as 'whole cell configuration during L-type calcium current measurement'. Is this correct?

Both observations of the reviewer are correct. For the first point, we have updated the revised manuscript to read:

"Success rate was defined as the percentage of wells able to reach whole cell configuration" (page 6, line 121).

For the second observation, we have changed the figure legend according to the reviewer's suggestion.

"F, Percentage success rates for whole-cell configuration from single plates of native CMs and hiPSC-CMs used in this protocol during $I_{Ca,L}$ acquisition." (Figure 2 Legend: page 27, line 601).

6. Materials and Methods, lines 122–124: "When starting an experiment, APC chips are loaded with the desired pipette solution and 30 μ l of a divalent-free solution containing (in mM): 10 HEPES, 140 NaCl, 4 KCl, 5 glucose." Where exactly is this 'divalent-free solution' located? Is the volume of 30 μ l for the entire chip or for a single well?

We thank the reviewer for identifying areas where clarification is necessary. To more clearly address the reviewer's comments, we have updated the text to read:

"When starting an experiment, the APC chip is loaded with the desired pipette solution and separately, 30 μ l of a divalent-free solution is automatically pipetted into each well. This initial extracellular solution contains (in mM): 10 HEPES, 140 NaCl, 4 KCl, 5 glucose." (page 6, line 125).

7. Materials and Methods, lines 137–139: "(...) with a holding potential of 80 mV and a 100 ms ramp pulse to -40 mV. This is held for 20 s before a 100 ms test-pulse to +10 mV." Should '20 s' perhaps read '20 ms'?

We thank the reviewer for noticing this mistake. 20 ms indeed. This has been corrected according to the reviewer's suggestion.

8. Materials and Methods, lines 146–148: “The fast and slow time constants of the biphasic $I_{Ca,L}$ decay were measured by fitting two standard exponential functions to each recording and assessing the tau (τ) for each fit.” Were these fits carried out over a range of test potentials or for a single specific test potential? Were the (relative) amplitudes of the fast and slow also analyzed

Data for the biphasic decay and time to peak parameters were extracted from $I_{Ca,L}$ signals at +10 mV. This has now been clarified in the revised methods section and the legend of **Figure 3**. Both are shown below.

“The fast and slow time constants of the biphasic $I_{Ca,L}$ decay at peak current (+10 mV) were measured by fitting two standard exponential functions to each recording and assessing the tau (τ) for each fit.” (page 7, line 152).

“F, Activation kinetics of $I_{Ca,L}$ at +10 mV expressed as time to peak amplitude. [...] I, Biphasic inactivation kinetics of $I_{Ca,L}$ at +10 mV expressed as fast decay (τ_{fast} ; upper) and slow decay (τ_{slow} , lower).” (Figure 3 Legend page 29, line 621).

This data was analysed using the proprietary Nanion DataControl 384 software. Presently, the algorithm of this batch-based analysis software does not have the capability to assess the amplitude/relative fraction of the biphasic decay functions of the L-type Calcium current – a current limitation of the technology and method. As the atrial and ventricular cardiomyocytes showed no significant differences in neither the fast nor slow components, nor indeed the time to peak, we do not expect to observe differences in the biphasic decay amplitude. However, in response to this reviewer’s question, we have submitted a request to the relevant programming department at Nanion Technologies to include this useful feature in future iterations of the software. This will likely take many months for a new release, which is unfortunately outside our timeframe of revision.

9. Materials and Methods, lines 174–175: “ $I_{K,ACh}$ was defined as the CCh-dependent current increase of inward current at -90 mV.” Is this $I_{K,ACh}$ the initial current or that obtained after desensitization (see also below)? I realize that Figure 5 suggests that $I_{K,ACh}$ is defined as the initial current.

$I_{K,ACh}$ is obtained as the initial current as described in **Figure 5** and **Supplementary Figure 5**. We have revised this part of the methods to make this clear:

“ $I_{K,ACh}$ was defined as the initial (peak) CCh-dependent current increase of inward current which is measured at -90 mV.” (page 9, line 181).

10. Materials and Methods, lines 198–200: “The positive control used in this proof of concept study consisted of Chinese hamster ovary (CHO) stably expressing Nav1.5 (Charles River Laboratories) on the same plate.” How were these CHO cells used? Was their peak sodium current measured? It should be explained which data of these cells were used as ‘positive control’ and how these data were acquired.

These CHO cells were always included on the same plate as the swine CM. As they should not exhibit Ca^{2+} currents, their response to an L-type Ca^{2+} current-inducing protocol was quantified and used as control to compare with the experimental recordings. We refer the interested reader to another publication also utilising this test¹.

In response to this reviewers point, we have modified our description of the Z factor test to more clearly explain our method.

“Where s = standard deviation, \bar{x} = mean, $_1$ = experimental group expressing maximal current (native CM), $_2$ = control group expressing minimal current. The control group used to assess minimal current consisted of Chinese hamster ovary (CHO) cells stably expressing Nav1.5 (Charles River Laboratories). These were measured in the same plate as the native cells for each experiment. The $I_{\text{Ca,L}}$ protocol was used to assess Z' .” (page 10, line 207).

11. Materials and Methods, lines 207–209: “Exclusion criteria (quality control) included a seal resistance <100 M Ω , a peak current of <50 pA, and an Rseries of 200 M Ω at 10 mV).” Which ‘peak current’ are you referring to? Should this current not be ‘ratioed to cell capacitance and expressed as current density’?

A definition of peak current has been clarified in the revised methods section of the manuscript. This is shown below.

“Results were only included if they showed a seal resistance of >100 M Ω , a peak current ($I_{\text{Ca,L}}$: +10 mV, $I_{\text{K1}}/I_{\text{K,ACh}}$: -90 mV) of >50 pA, and an R_{series} of <200 M Ω (at 10 mV)” (page 10, line 215).

A cut off for peak current such as this is typically applied as technical step to ensure we are measuring above the noise level of the system. This is why we have opted to not ratio to cellular capacitance to assess current density as an exclusion criteria.

12. Materials and Methods, lines 209–210: “AP parameters such as resting membrane potential (RMP) and Action potential duration was analyzed offline using DataControl 384 software.” Write ‘were’ instead of ‘was’.

This has been corrected according to the reviewer’s suggestion.

13. Materials and Methods, lines 210–212: “Using the AP search mode, action potentials that did not display a clear threshold take-off potential following increasing current stimuli were excluded.” What is ‘AP search mode’?

AP search mode is a function of the proprietary DataControl 384 software from Nanion Technologies. We have updated the text to make this clear. The updated sentence proceeds as follows.

“Using the AP search feature of the software, action potentials that did not display a clear threshold take-off potential following increasing current stimuli were excluded.” (page 10, line 219).

14. Materials and Methods, general. It seems that alle recordings (both APs and individual membrane currents) have been made with EGTA in the internal (pipette) solution. Have recordings been made without EGTA? If so, were these successful? If not, what was the rationale for using EGTA in all recordings?

No recordings were undertaken without EGTA in the pipette solution. EGTA in the internal solution is a standard in automated patch clamp¹ and is used as a measure of recording quality control. It allows for the chelation of any divalent ions within the pipette (intracellular) solution which could otherwise influence the quality and specificity of the recordings of specific ionic membrane currents.

15. Results, lines 225–228: “The patching success rate, defined as effective seal formation (>100 MΩ) and whole cell configuration, and achieved through gentle negative pressure application, was 13.9±1.7% in native cells (15 plates over 3 days) and showed no significant changes over successive experimental days.” Figure 2A suggests that the success rate with atrial cells (red ‘ring’) is substantially higher than with ventricular cells (blue ‘ring’). Has this been tested? What was the outcome?

Ventricular cardiomyocytes did yield, on average, a smaller patching success rate as indicated by the screenshot in **Figure 2A**. This due to the smaller populations of ventricular cells available after isolation and the high variability in ventricular isolation success rates. “[...] 8790±1610 viable atrial and 7200±3903 viable ventricular cardiomyocytes.” (page 11, line 228).

We attribute this phenomenon of smaller ventricular cell populations to the isolation procedure. Our isolation protocol implemented here (and elsewhere^{2,3}) is specifically designed, particularly in its designated collagenase constituents, for cellular isolation of human atrial myocardium. It therefore follows that these protocols would not be perfectly optimised for isolating ventricular cardiomyocytes from a swine sample.

We have added the following sentence into the revised version of the results section to illustrate the difference between atrial and ventricular patching success rates.

“Out of successful measurements in native cardiomyocytes, 29.3±5.9% (P<0.05) more recordings were obtained from atrial cells compared to ventricular cells” (page 11, line 239).

16. Results, lines 235–237: “In addition, Z factor analysis, a marker for high throughput assay reproducibility, consistently showed good to excellent values of assay robustness and reproducibility in multiple plates of primary cells over different days (Figure 2C).” What is referred to as ‘good’ here, is ‘moderately successful’ according to the Materials and Methods section. Which is correct?

We have removed the words ‘moderately successful’ from the methods section and remained consistent in our definitions throughout the manuscript as per **Figure 2**.

17. Results, lines 235–237: “Clear temporal characteristics of $I_{Ca,L}$ activation and biphasic decay are also easily extracted (Figure 3F, I).” Were these data obtained at a test potential of +10 mV? What was the relative amplitude of the fast and slow components of decay? Were these different between atrial and ventricular cells?

Yes, these data were extracted at +10 mV using the proprietary Nanion DataControl 384 software. We have clarified that these measurements were taken at +10 mV peak current value.

“Clear temporal characteristics of $I_{Ca,L}$ activation and biphasic decay were also extracted at +10 mV (Figure 3F, I)” (page 12, line 267).

Regarding the amplitudes of the biphasic decay functions of the L-type Calcium current, please see our response to comment 8 from this reviewer.

18. Results, lines 264–266: “When normalized to $I_{Ca,L}$ amplitude at full pharmacological block, atrial myocytes showed an EC50 of 6.08 ± 1.14 nM and ventricular cardiomyocytes showed 3.41 ± 0.71 nM.” Do these EC50 values show a statistically significant difference? Were these EC50 values obtained with a Hill equation? If so, what were the associated nH values?

These curves were generated using a standard sigmoidal dose response function using a least squares regression fit within Graphpad Prism and an assumed Hill coefficient of 1. EC50 values were significantly different between atrial and ventricular cohorts. We have updated **Figure 3** in the revised version of the manuscript to reflect this. This figure is shown below.

Figure 3. L-type calcium current ($I_{Ca,L}$) acquisition and analysis from native atrial and ventricular cardiomyocytes (CM) using automated patch-clamp. **A**, Schematic of swine cardiac tissue harvesting and CM isolation. **B**, Representative recordings of membrane current (I_M) showing $I_{Ca,L}$ in atrial (upper) and ventricular (vent.; lower) CMs. Inset: Voltage protocol (upper right). **C**, Peak $I_{Ca,L}$ density measured at +10 mV. **D**, Current-voltage (I - V) relationship curves for $I_{Ca,L}$ in atrial ($n=71$) and ventricular ($n=26$) CMs. **E**, Left: Voltage protocols for I - V /activation experiments (upper) and S1/S2 voltage protocols for $I_{Ca,L}$ inactivation (lower). Right: $I_{Ca,L}$ activation, measured as G/G_{max} in atrial ($n=71$) and ventricular ($n=26$) CMs, with corresponding inactivation curves (I/I_{max}) in atrial ($n=15$) and ventricular ($n=6$) CMs. **F**, Activation kinetics of $I_{Ca,L}$ at +10 mV expressed as time to peak amplitude. **G**, Concentration dependent response to nifedipine (Nif) application in $I_{Ca,L}$ current (upper), time course of average $I_{Ca,L}$ from a single plate following nifedipine application (lower, $n=10$). **H**, Concentration response curve of $I_{Ca,L}$ (normalized to current amplitude at full block with $5 \mu\text{M}$ Nif) following nifedipine application in atrial and ventricular CMs with corresponding half

maximal effective concentration (EC50). **I**, Biphasic inactivation kinetics of $I_{Ca,L}$ at +10 mV expressed as fast decay (τ_{fast} ; upper) and slow decay (τ_{slow} , lower). Data are mean \pm SEM from 3 animals. * $P < 0.05$ vs ventricular. n =number of atrial (176) and ventricular (58) CMs from 3 animals (**C**, **F**, **I**) or atrial (94) and Ventricular (39) (**H**).

19. Results, lines 271–274: “Atrial myocytes showed a shorter action potential duration at 50% repolarization (APD50) compared to ventricular (Figure 4C). This typical discrepancy in chamber specific phenotype was also replicated using the same APC method in hiPSC-CM generated using subtype specific differentiation protocols (Supplementary Figure 4).” However, Figure 4 shows action potential duration at 50% repolarization, whereas Supplementary Figure 4 shows action potential duration at 90% repolarization. It would be informative to show APD50 and APD90 in both Figure 4 and Supplementary Figure 4, or at least show either APD50 or APD90 in both Figure 4 and Supplementary Figure 4.

Thank you for pointing out this discrepancy. We have updated **Supplementary Figure 4** to more consistently reflect the data parameters, particularly APD₅₀, in **Figure 4**. The updated **Supplementary Figure 4** is shown below.

Supplementary Figure 4. Automated patch-clamp (APC) of human atrial and ventricular induced pluripotent stem cell-derived cardiomyocytes (hiPSC-CM). **A**, Schematic of human dermal fibroblast harvesting, reprogramming into induced pluripotent stem cells (hiPSC) and subsequent differentiation into ventricular (Vent.) and atrial cardiomyocytes with the latter receiving retinoic acid (RetA) early in differentiation for atrial lineage confirmation. **B**, Representative action potential (AP; left) and L-type calcium current ($I_{Ca,L}$; right) recorded from separate atrial hiPSC-CMs. insets: current protocol for AP acquisition and voltage protocol for $I_{Ca,L}$ acquisition. **C**, Representative AP (left) and $I_{Ca,L}$ (right) recorded from ventricular hiPSC-CMs. **D**, AP duration at 50% repolarization (APD₅₀; left), current threshold which first elicited an AP response (center) and resting membrane potential (RMP; right; $n = 108$ atrial vs 35 ventricular). **E**, APD₅₀ (ms) for atrial (red) and ventricular (cyan) cells. **F**, Current threshold (pA) for atrial (red) and ventricular (cyan) cells. **G**, RMP (mV) for atrial (red) and ventricular (cyan) cells. **H**, $I_{Ca,L}$ (pA/pF) for atrial (red) and ventricular (cyan) cells. **I**, $I_{Ca,L}$ - V_M relationship for atrial (red) and ventricular (cyan) cells.

ventricular). **F**, $I_{Ca,L}$ current density measured at +10 mV ($n = 9$ atrial vs 26 ventricular; right). **G**, A single representative current-voltage (I - V) relationship plot for $I_{Ca,L}$ in an individual atrial and ventricular hiPSC-CM. Data are mean \pm SEM. * $P < 0.05$ vs ventricular.

20. Results, lines 312–314: “In this small cohort, we observed a uniform increase of action potential duration at 90% repolarization (APD90) as $I_{Ca,L}$ amplitude increases in atrial cardiomyocytes.” Looking at Figure 6D, I would not speak of ‘a uniform increase’

We have removed the word “uniform” from this sentence.

21. Results, lines 328–329: “Similar stability and current quality was observed in all successful experiments across different days (Figure 6F, Supplementary Figure 7).” How does Figure 6F, which shows data from a single plate, confirm this statement?

We have updated the text to read:

“Similar stability and current quality was observed in all successful experiments across different days. Representative screen shots of raw sequential CAPER measurements on single cells are shown in Figure 6F and Supplementary Figure 7.” (page 15, line 341).

22. Discussion, lines 337–338: “Numerous detailed electrophysiological parameters were able to be reproducibly extracted from separate cohorts of swine native cardiomyocyte tissue, (...)” I would write ‘isolated swine native cardiomyocytes’ instead of ‘swine native cardiomyocyte tissue’.

This has been corrected according to the reviewer’s suggestion.

23. Discussion, lines 345–348: “After decades of failure to record from primary adult mammalian cardiomyocytes, we now report for the first time a highly versatile method for deep and user unbiased electrophysiological phenotyping of primary cardiac material.” This reads as an overstatement. In the past decades, successful recordings from primary adult mammalian cardiomyocytes have been carried out in numerous laboratories over the world. You should either remove or tone down this statement.

We have adjusted this sentence to clarify that there have indeed been decades of failure to obtain recordings from native cardiomyocytes using **automated patch clamp**. The sentence now reads:

“After decades of failure to record from primary adult mammalian cardiomyocytes with APC, we now report for the first time a highly versatile method for deep and user unbiased electrophysiological phenotyping of primary cardiac material” (page 16, line 359).

24. Discussion, lines 381–383: “In contrast, the corresponding pipette solution in a conventional patch-clamp system needs to be maintained throughout a complete experiment.” Successful attempt have been made to replace pipette solutions during an experiment. I would therefore not write that ‘the corresponding pipette solution in a conventional patch-clamp system needs to be maintained throughout a complete experiment’, but, with appropriate references, that exchanging the pipette solution during such experiment is highly complex (or something similar).

This has been corrected according to the reviewer’s suggestion. The sentence now reads:

“In contrast, the corresponding pipette solution in a conventional patch-clamp system usually needs to be maintained throughout a complete experiment and is extremely difficult to exchange during an active recording.”^[4] (page 17, line 395).

25. Discussion, lines 414–415: “In typical APC experiments, fluoride ions in the internal solution aid in seal formation by reacting with external Ca²⁺ and forming a precipitate around the aperture-cell interface.” Please provide references.

As this phenomenon is not yet well described using this new technology, we have redacted the majority of this sentence here and in the supplementary material (please see comment 49 from this reviewer). The sentence now reads:

“In typical APC experiments, fluoride ions in the internal solution seem to foster giga seal formation and help to increase electrical resistance and seal stability.” (page 19, line 431).

26. References, #18. ‘Maria G. Rotordam et al.’ should read ‘Rotordam, M. G. et al.’

This has been corrected according to the reviewer’s suggestion.

27. References, #26 and #27. Will these application notes remain publicly available? Do they have a DOI?

These application notes will always remain publicly available on the companies’ respective websites. We have removed one of these citations and added the respective URL to the remaining citation within the reference list. The owners do not have a DOI for this content.

In addition, we have also included a recently submitted manuscript to our citations which highlights the advent of fluoride-free recording with this device⁵. This new manuscript is under review elsewhere. The confidential document will be supplied with our submission as a related manuscript file for the reviewers use only.

28. Figure 1, legend. The legend to Figure 1 refers to panels A–D, but the actual figure has only labels A and B. The legend should be adapted to the figure (or vice versa).

We thank the reviewer for spotting this remnant of our original manuscript. This has now been corrected according to the reviewer’s suggestion.

29. Figure 2F and its legend. How is 'absolute success rate' defined?

Please see our response to comment 5 from this reviewer. We have now removed the word absolute, made sure we refer constantly to the percentage success rate, and clarified this within the figure legend of **Figure 2**. The updated legend is shown below.

"F, Percentage success rates for whole-cell configuration from single plates of native CMs and hiPSC-CMs used in this protocol during $I_{Ca,L}$ acquisition." Figure 2 Legend: (page 27, line 601).

30. Figure 2H. Membrane capacitance and series resistance remain nicely constant over time when expressed as mean \pm SEM for an entire plate. However, is this also true for individual wells?

This is also true for individual wells. To demonstrate this, we have provided a figure for this reviewer below, which shows the capacitance measurements and series resistance from single cells of a plate shown in **Figure 2**.

Reviewer Figure 1. Single-well stability. Time course of a single plate (15 cells) showing series resistance (R_{series} ; upper) and membrane capacitance (lower) in native CMs (3 plates, same day) during $I_{\text{Ca,L}}$ experiments. Shading is only for optical differentiation of each line.

31. Figure 3, legend: “B, Representative recordings of $I_{\text{Ca,L}}$ in atrial (upper) and ventricular (vent.; lower) CM.” Panel B shows “IM” (for ‘membrane current’?) rather than $I_{\text{Ca,L}}$. This also holds for other figures and panels. Panel B does not show “vent.”

I_{M} refers to membrane current throughout the manuscript because this is what we are measuring in all instances. We have updated all relevant figure legends to provide this definition.

Panel B shows a ventricular example of an L-type calcium current recording. Throughout the figure, the word ventricular is regularly shortened to Vent. This shorthand is defined in the figure legend, as is any other definition, following the first instance of the full word. This is consistent throughout all figures of this manuscript.

32. Figure 3, legend: “E, Right: Voltage protocols for I-V/activation experiments (upper) and S1/S2 voltage protocols for I_{Ca,L} inactivation (lower). Left: I_{Ca,L} activation, measured as G/G_{max} in atrial (n=71) and ventricular (n=26) CM, with corresponding inactivation curves (I/I_{max}) in atrial (n=15) and ventricular (n=6) CM.” Voltage protocols are shown left and (in)activation curves right.

Thank you for noticing this mistake. This has been corrected according to the reviewer's suggestion.

33. Figure 3H and its legend. How is ‘normalized response’ defined? Is this the percent peak I_{Ca,L} density measured at +10 mV? How large are the numbers of atrial and ventricular myocytes?

These points have been added in the revised Figure legend of **Figure 3**. The sections of interest are shown below:

*“H, Concentration response curve of I_{Ca,L} (normalized to current amplitude at full block with 5 μM Nif) following nifedipine application in atrial and ventricular CMs with corresponding half maximal effective concentration (EC₅₀). I, Biphasic inactivation kinetics of I_{Ca,L} expressed as fast decay (τ_{fast} ; upper) and slow decay (τ_{slow} , lower) Data are mean±SEM from 3 animals. *P<0.05 vs ventricular. n=number of atrial (176) and ventricular (58) CMs from 3 animals (C, F, I) or atrial (94) and Ventricular (39) (H).” (Figure 3 Legend: page 29, line 624).*

34. Figure 3, legend, last line: “n=number of atrial (176) and ventricular (58) CM from 3 animals (B, E, F).” ‘(B, E, F)’ should probably read ‘(C, F, I)’.

Thank you for pointing out this error. This has been corrected according to the reviewer's suggestion.

35. Figure 4, A and E. Given the APD₅₀ values reported in panel C, I would not speak of ‘representative traces’ in panels A and E. In particular, the duration of the ventricular action potentials seems much longer than ‘representative’. I suggest to replace panels A and E with more representative ones.

This is a fair comment. We have updated the representative traces to examples that we hope more accurately reflect our data. The new **Figure 4** is shown below.

Figure 4. Action potential acquisition from native atrial and ventricular cardiomyocytes (CM) using automated patch-clamp. **A**, Representative traces membrane voltage (V_M) showing atrial (left) and ventricular (vent.; right) triggered action potentials during successive increases in pulse current injection. Insets: current protocol. **B**, Current at which action potential take off was first observed. **C**, Action potential duration at 50% repolarization (APD_{50}). **D**, Resting membrane potential (RMP) quantification. **E**, Representative traces of AP trains at 1 Hz showing stable RMP between pulses. Data are mean \pm SEM. * $P < 0.05$ vs ventricular. n =number of atrial (127) and ventricular (45) CMs from 3 animals (**B**, **C**, **D**).

36. Figure 4D. I am seriously concerned by the 'resting membrane potential (RMP) quantification' of Figure 4D. It seems that the resting membrane potential of atrial CMs can be as negative as -110 mV or even more negative. This is at odds with the potassium reversal potential of around -86 mV that can be computed from the 'internal' and 'external' solutions. How reliable are these RMP data? And what about other data? According to Figure 4C, the action potential duration at 50% repolarization (APD_{50}) of both atrial and ventricular swine native cardiomyocytes can be as small as 5–10 ms. Again, how reliable are these data?

Indeed, the resting membrane potential (RMP) is variable in these recordings. This is due to the presence of iterative current injections that clamp the cell to a set hyperpolarised RMP value (-90 mV). This approach reveals the individual cellular dependence and variability upon current injections, expressed through minuscule changes in access properties over the time course of an experiment. These could contribute to the observed variation in RMP/APD₅₀. It is unclear why that is more pronounced/different in atrial over ventricular cells, However, it could be related to differences in expression levels of depolarizing/hyperpolarizing ion channels, such as variations in IK1 (as suggested by Figure 5C).

Importantly, typical amplifiers, including those within the SyncroPatch system, are mainly designed for current measurements in voltage-clamp mode. In their current-clamp configuration, they are characterized by relatively low input resistance resulting in current absorption and thereby causing a voltage drop over the cell membrane and necessitating the use of extra injected holding current.⁶ This is an inherent disadvantage of the current clamp technique, as the majority of papers (including ours) report artificial RMP values due to the presence of these holding currents or indeed due to insufficient seal quality, introducing systematic error and possibly mis-reporting apparent RMP.⁷ For measurement of true resting membrane potential, sharp microelectrode techniques using 'voltage follower' amplifiers are recommended, along with pipettes of very high resistances (>50 MΩ). Automated Patch clamp technology has not yet reached the capability to offer this technique.

In a scientific context, reliability refers to the extent to which the results of an experiment are consistent when repeated under identical conditions. We have provided single point data plots across the entire manuscript for the reader to assess for themselves the reliability of our findings. Considering the variability inherent in biological systems and the novelty of this approach, we feel that the variability that we report is consistent enough to be reliable. However, of course, further optimisation of this method will serve to increase the success rate and indeed the reliability of our findings. To address this, we have added the following sentence to the limitations section of the discussion:

“Further optimization is required in order increase the CAPER success rate, for example through increasing working temperature to more physiological temperatures while maintaining cell viability or selectively targeting the composition of extracellular and intracellular solutions to enhance seal quality. Such modifications should also be targeted towards reducing the heterogeneity of the presently reported experimental data within the cellular subtype cohorts.” (page 19, line 423).

37. Figure 4, legend, last line: “n=number of atrial (127) and ventricular (45) CM from 3 animals (C, D; F).” ‘(C, D; F)’ should probably read ‘(B, C, D)’.

Again, we thank the reviewer for noticing this error. This has been corrected according to the reviewer’s suggestion.

38. Figure 5, legend: “B, Time course of a single plate with atrial (n= 10; left) and ventricular (n= 11; right) CM inward current at 90 mV during a typical experiment.” The membrane potential of ‘90 mV’ should probably read ‘-90 mV’

We thank the reviewer for noticing this sloppy mistake. This has been corrected according to the reviewer's suggestion.

39. Figure 5, legend: "E, Time course of membrane capacitance from a single plate (B, left) over various external solution changes. F, Time course of series resistance (R_{series}) from a single plate (B, left) over various external solution changes." Data are for a single plate. Are these data representative for other plates? How many cells were on this single plate?

Yes these data are representative for all plates. Our stable data shown in **Figure 2** details comparable cell capacitance and series resistance over multiple plates. This was replicated after solution exchanges and the differences on a per plate basis during solution exchange are shown in **Figure 5G, H**. We have modified the following sentence in the results section.

"Figures 5E, F show representative changes in both values respectively from one plate. The ratio of parameter change between the second solution addition (S2) and the third solution addition (S3) shows no significant difference from multiple plates over multiple days of experimentation (Figure 5G, H) indicating consistent seal quality during solution changes across all plates that were used." (page 14, line 312).

The plate in **Figure 5E, F** is the same plate as that shown in **Figure 5B**, left, as stated in the original figure legend. In order to clarify this further, we have repeated the cell number in the figure legend for E and F. This is shown below.

"E, Time course of membrane capacitance from a single plate (B, left [n=10]) over various external solution changes. F, Time course of series resistance (R_{series}) from a single plate (B, left [n=10]) over various external solution changes." (Figure 5 Legend: page 32, line 655).

40. Figure 6F. What are the 'numbers' along the vertical axes? Does '400p' (left panels) denote '400 pA' and '1n' (right panels) '1 nA'? And does '100m' (middle panels) denote '100 mV'?

These assumptions are all correct. As these are screenshots of the software, we apologise for the confusion. The legend has been updated with the following sentence to make these aspects clear.

"Y axis indicates the membrane current expressed as picoamperes (p) or nanoamperes in the screenshots of $I_{Ca,L}$ (left) and I_{K1} (right), or membrane voltage expressed as millivolts (m; center)." (page 33, line 676).

41. Supplementary Information, general: It seems that some text is in dark grey, whereas other text is in pure black. Select one color and use it throughout.

Again, we thank the reviewer for noticing this sloppy mistake, and those below. This has been corrected according to the reviewer's suggestion.

42. Supplementary Information, line 6: “Rupamanjari Majumder, PhD, Sebastian Clauß, MD.” Insert a space between “PhD” and “Sebastian”.

This has been corrected according to the reviewer’s suggestion.

43. Supplementary Information, lines 37–38: “Human induced pluripotent stem cell (hiPSC) line UMGi014-C clone 14 (isWT1.14) were derived from the dermal fibroblasts of a healthy male donor.” Replace “were” with “was”.

This has been corrected according to the reviewer’s suggestion.

44. Supplementary Information, lines 41–43: “hiPSC were cultured on 1:120 Matrigel™ (BD Biosciences) coated plates and maintained with Stem MACS IPS-Brew XF medium (Miltenyi Biotec) daily.” Now that ‘hiPSC’ abbreviates ‘human induced pluripotent stem cell’ (line 37), write ‘hiPSCs’ instead of ‘hiPSC’. Similarly, write ‘hiPSC-CMs’ instead of ‘hiPSC-CM’ at lines 59 and 60 as well as at other instances in the supplementary figures and their legends. Similarly, write ‘AP’ for action potential and ‘APs’ for action potentials. This holds for the main manuscript as well.

This has been corrected throughout the supplements and manuscript according to the reviewer’s suggestions.

45. Supplementary Information, lines 62–66: “Cells were returned to 4 °C for 10 minutes and then exposed to pre-cooled HBSS (Thermo Fisher Scientific) at a volume to provide sufficient cellular density (>100,000 cells per ml; Supplementary Figure 2) and sparingly resuspended with a glass pipette in order to detach cells into isolated bodies.” How does this statement relate to Supplementary Figure 2, which (only) shows a “representative time course of seal resistance”?

Indeed, this observation is correct. This mistake has been amended.

46. Supplementary Information, lines 69–70: “IK1 and CAPER experiments were not attempted.” What are “IK1 and CAPER experiments”? Explain here.

These aspects have now been defined in the supplementary material.

“hiPSC-CM experiments were conducted under the same conditions as native cardiomyocytes outlined in the manuscript. Inward rectifier (I_{K1}) measurements and multi parameter protocols such as the Calcium-Action Potential- Inward Rectifier (CAPER) protocol were not attempted.” (Supplementary Information: page 4).

47. Supplementary Information, lines 70–71: “For action potential acquisition, injected current (-9.7 ± 1.5 pA/pF) was administered independently to each well as required.” Add duration of current pulse.

This is referring to holding current which clamps the cell at low resting membrane potentials. We have updated the text as follows.

“For action potential acquisition, injected holding current (-9.7 ± 1.5 pA/pF) was administered independently to each well as required.” (Supplementary Information: page 4).

48. Supplementary Figure 1. Why would the 1,000 cardiomyocytes studied by a skilled operator yield 1,000 “data points” (ratio 1:1), while the 115,000 cardiomyocytes studied through automated patch clamp yield 350,000 “data points” (ratio 1:3)?

This is a fair observation. This 1:3 ratio was to reflect the potential of multi parameter studies such as the CAPER protocol which has the capacity to measure 3 or more electrophysiological parameters in a single cell. We have clarified this in the revised version of the figure legend for **Supplementary Figure 1**.

“For the purposes of this figure: ‘Data points’ are defined as currents/action potentials measured per cell e.g $I_{Ca,L}$, AP, and I_{K1} (3 parameters) as seen in the CAPER protocol. They do not include the numerous recordings possible using pharmacological screening of a single current/AP.” (Supplementary Figure 1, Legend).

49. Supplementary Figure 2, legend: “Before whole cell configuration is achieved, transiently elevated external Ca^{2+} levels (5 mM max) are thought to interact with fluoride in the internal solution to form a precipitate at the aperture-cell interface, fostering giga seal formation.” Please provide references.

As this phenomenon is not yet well described using this new technology, we have redacted the majority of this sentence here and in the main manuscript (please also see comment 25 from this reviewer). The sentence now reads:

“Before whole cell configuration is achieved, transiently elevated external Ca^{2+} levels (5 mM max) help to foster giga seal formation.” (Supplementary Figure 2, Legend).

50. Supplementary Figure 3, legend: “B, Representative L-type calcium current ($I_{Ca,L}$) recorded from atrial CM (upper) and ventricular (vent.; lower) CM.” Panel B shows “IM” (for ‘membrane current’?) rather than $I_{Ca,L}$. Panel B does not show “vent.” This also applies to Supplementary Figure 4.

Please kindly refer to our response to comment 31 from this reviewer. The necessary modifications have been made throughout the revised version of the manuscript.

51. Supplementary Figure 3, legend: “C, $I_{Ca,L}$ current density (...).” Is “ $I_{Ca,L}$ current density” defined as peak current density at +10 mV? This also applies to Supplementary Figure 4.

To clarify this, we have amended the figure legends of both figures in question to make it clear that these current densities were acquired at +10 mV.

“C, $I_{Ca,L}$ current density measured at +10 mV ($n = 17$ atrial vs 19 ventricular from 3 animals).” (Supplementary Figure 3, Legend).

“F, $I_{Ca,L}$ current density measured at +10 mV ($n = 9$ atrial vs 26 ventricular; right).” (Supplementary Figure 4, Legend).

52. Supplementary Figure 4, legend: “B, Representative action potential (AP; left) and L-type calcium current (I_{Ca,L}; right) recorded from atrial hiPSC-CM. insets: current protocol for AP acquisition and voltage protocol for I_{Ca,L} acquisition.” Does panel B shows APs and membrane current from a single hiPSC-CM or from multiple hiPSC-CMs? Printed at 100% size, the insets are barely readable, also because the supplied figure is not sharp enough (both in the pdf and its Word format source file).

The action potentials are from a single cell receiving increasing current injections until AP take off is reached. The membrane current recordings are from a different cell. We have clarified this in the respective figure legend and increased the size of the protocol insets to make this clear. The updated figure legend is shown below.

“B, Representative action potential (AP; left) and L-type calcium current (I_{Ca,L}; right) recorded from separate atrial hiPSC-CMs” (Supplementary Figure 4, Legend).

53. Supplementary Figure 4, legend: “G, Representative current-voltage (I-V) relationship curves for I_{Ca,L} in atrial and ventricular CM.” What are we looking at in panel G? Are these average data (without error bars) or data from one representative atrial hiPSC-CM (not just “CM”) and one representative ventricular hiPSC-CM? How were the smooth I-V curves obtained?

Panel G shows an IV relationship for a single atrial hiPSC-CM and a single ventricular hiPSC-CM. To make this clear, we have updated the figure legend as shown below.

“G, A single representative current-voltage (I-V) relationship plot for I_{Ca,L} in an individual atrial and ventricular hiPSC-CM.” (Supplementary Figure 4, Legend).

The lines of best fit were obtained using a modified Boltzmann equation. We thank the reviewer for this question which has allowed us to double check that our methods are accurate and that we have followed the previous methods of other publications correctly^{8,9}. In addition, we have supplied the formula how these fitted lines were calculated in the revised methods section of the main manuscript. This is shown below.

“The I-V curves were fitted with a modified Boltzmann equation:

$$I_{Ca,L} = \frac{G_{max}(V - E_{rev})}{1 + \exp\left(\frac{[V_{50} - V]}{k}\right)}$$

where G_{max} = maximal conductance, E_{rev} =reversal potential, V₅₀ = half-activation potential, and k = slope factor.” (page 7, line 145).

54. Supplementary Figure 4. Panel D suggests that the resting membrane potential (RMP) of both atrial and ventricular hiPSC-CMs can be as negative as -95 mV. This seems at odds with the potassium reversal potential of around -86 mV that can be computed from the ‘internal’ and ‘external’ solutions and requires explanation.

Please see our response to comment 36 from this reviewer. Indeed, the measured RMP is lower than the reversal potential according to the Nernst equation due to the presence of extreme injected holding currents as well as the possible impact of cellular parameters such as seal quality and access resistance. All holding currents are reported in the manuscript and Supplementary Information. This is an inherent disadvantage of the current clamp method, and naturally should be considered when assessing data acquired through current clamp.

55. Supplementary Figure 5, legend: “A, Representative uncorrected basal inward rectifier current (I_{K1}) in atrial (left) and ventricular (vent.; right) with superimposed acetylcholine-activated inward rectifier ($I_{K,ACh}$) current following carbachol (CCh) application, and during full block with $BaCl_2$ application, all during a depolarizing ramp voltage protocol (lower).” Please clarify which of the three traces show the ‘uncorrected basal inward rectifier current (I_{K1})’ and the ‘superimposed acetylcholine-activated inward rectifier ($I_{K,ACh}$) current’. Was the curve upon CCh application obtained after the “eventual desensitization” of panel D? Printed at 100% size, the smaller text is barely readable, also because the supplied figure is not sharp enough (both in the pdf and its Word format source file).

Thank you for pointing out this issue of clarity. We have labelled the traces as they are defined in the figure legend. The curve represents the peak $I_{K,ACh}$, not the quasi steady state. We have updated the figure legend to reflect this. The updated figure and legend are shown below. We have also increased the text size to make the figure text more readable.

Supplementary Figure 5. Features of inward rectifier currents measured in native atrial and ventricular cardiomyocytes (CM) using automated patch-clamp. A, Representative uncorrected basal inward rectifier current (I_{K1}) in atrial (left) and ventricular (vent.; right) CMs with superimposed uncorrected peak (initial) acetylcholine-activated inward rectifier ($I_{K,ACh}$) current following carbachol (CCh) application, and during full block with $BaCl_2$ application, all during a depolarizing ramp voltage protocol (lower). B, Cell capacitance. C, Membrane

resistance, calculated through dividing the driving force of K^+ (30 mV) by the absolute Ba^{2+} sensitive current at -90 mV. **D**, Detailed representative timecourse of CCh application and the eventual desensitization to a quasi steady-state (QSS). **E**, QSS current. Data are mean \pm SEM. * $P < 0.05$ vs ventricular. n =number of atrial (151) and ventricular (143) CMs from 3 animals (**B**, **C**; **E**).

56. Supplementary Figure 5. Panels D and E suggest that the membrane capacitance of the hiPSC-CMs can be almost zero and the membrane resistance can actually be zero or even below. How reliable are these data? Are there any exclusion criteria with respect to membrane capacitance and/or membrane resistance?

Our exclusion criteria are listed in the main manuscript. We exclude any recordings of seal resistance of $< 100 \text{ M}\Omega$, a peak current of $< 50 \text{ pA}$, or an R_{series} of $> 200 \text{ M}\Omega$ (at 10 mV). Exclusion was not applied to cell capacitance due to the expected variability in size between cardiomyocytes as they undergo digestion, isolation and are subsequently subjected to the machine. This variability includes cells with low cellular capacitance. There are no cells included with a membrane resistance and capacitance values of 0 or below. Please note that the y axis is in units of $\text{G}\Omega$, and that the size of the diameter of a single data point on this particular plot is in the realm of $100 \text{ M}\Omega$ or more.

With regards to reliability, please refer to our response to comment 36 from this reviewer. To address this disadvantage of high heterogeneity between our data, we have added a new sentence in the limitations section of the discussion. This is shown below.

Further optimization is required in order increase the CAPER success rate, for example through increasing working temperature to more physiological temperatures while maintaining cell viability or selectively targeting the composition of extracellular and intracellular solutions to enhance seal quality. Such modifications should also be targeted towards reducing the heterogeneity of the presently reported experimental data within the cellular subtype cohorts.” (page 19, line 423).

57. Supplementary Figure 6, legend: “Screenshots of Supplementary Video 1.” The 15 screenshots of Supplementary Video 1 require a more detailed explanation than just these 5 words.

This figure was only intended to supplement the video for those online viewers who cannot access or cannot play the required media. We have taken both this and comment 58 from this reviewer into account and removed this superfluous figure from the supplemental material.

58. Supplementary Figure 6. The screenshots are so small that the text has become unreadable. This is not only because the supplied figure is not sharp enough, but also because the screenshots are simply far too small.

Please see our response to comment 57 from this reviewer above. We agree and have removed this supplementary figure.

59. Supplementary Figure 7, legend: “Direct acquisition software screenshots from a single plate showing complete CAPER acquisition from multiple cells over three days (and therefore 3 animals).” This figure requires better explanation. What exactly are we looking at? Does the three times three panels per day show APs and membrane current recorded from three single hiPSC-CMs from the plate of that specific day? If so, how were these three hiPSC-CMs selected? And are these data from atrial or from ventricular hiPSC-CMs? You should consider the use of ‘Experiment 1’, ‘Experiment 2’, and ‘Experiment 3’ rather than the somewhat confusing ‘Day 1’, ‘Day 2’, and ‘Day 3’ (as if recordings were made from the same cells over three days).

In order to demonstrate the extent of reproducibility of the CAPER protocol, this figure shows 3 representative CAPER experiments from native atrial CMs from 3 different animals. We thank the reviewer for pointing out this issue, and following their guidance, we have altered (now called) **Supplementary Figure 6** and its legend in such a way where it is more understandable. Both the updated figure and the legend are shown below.

Automated patch-clamp of native cardiomyocytes
 Commun Biol COMMSBIO-21-3514B

Animal 1

Animal 2

Animal 3

Supplementary Figure 6. Representative raw data of atrial cardiomyocytes undergoing the multi-current Cal_{Ca}, Action Potential and inward rectifiER (CAPER) protocol. Direct screenshots from the Nanion DataControl 384 software during recording analysis. One row indicates one cell. A single animal was measured per day. Three representative cells are shown for each animal. The Y axis of each recording indicates the membrane current expressed as picoamperes (p) or nanoamperes (n) in the screenshots of $I_{Ca,L}$ (left side) and I_{K1} (right side), or membrane voltage expressed as millivolts (m; center).

60. Supplementary Video 1. The screenshots are so small that the text has become barely readable. This is not only because the supplied figure is not sharp enough, but also because the screenshots are simply too small. I would prefer a series of six 2D APD90 vs. I_{Ca,L} plots obtained at I_{K1} densities of 0, 1, 2, 3, 4, and 5 pA/pF, respectively, instead of, or in addition to, a single 3D surface plot.

Firstly, as described in our response to comment 57 and 58 from this reviewer, we have removed this unnecessary and indeed, difficult to read, supplemental figure with small screenshots.

Secondly, the reason that our brief exploration of *in silico* modeling is relegated to the supplements is because it is not the focus of the current study. We do not speculate or draw inferences based on the results of this comparison. It is included to provide a brief glimpse into what is possible when the potential power of automated patch clamp of native cardiomyocyte material is combined with mathematical modelling. The '3D' data was derived from a mathematical model that was recently published¹⁰. In order to mimic the variability in the natural electrophysiology of different cells, a random noise was introduced into the parameter space of the model. Please see our detailed responses to reviewer comments 60 to 64 for more information. Because this model is already published, and indeed not the focus of this study, we have kept Supplementary Video 1 included in the supplements and but have not elaborated further on its detailed comparison with our experimental data as we feel this is beyond the scope of the present work.

61. Supplementary Video 1, legend: "An in silico surface plot of all three experimental CAPER parameters (L-type calcium current [I_{Ca,L}], action potential duration at 90% repolarization [APD90] and basal inward rectifier current [I_{K1}]) was generated through a randomized array of 10,000 CMs." It should be explained in more detail how these in silico data were obtained. How was this 'randomized array of 10,000 CMs' constructed?

We have provided a point by point response to the model related questions from this reviewer below. This surface plot is based on a custom written code to measure the AP characteristics and currents from a single cardiomyocyte whose parameters (ion channel conductances, time constants and rate constants for the reaction kinetics) could be varied within a pre-set spread. Each parameter varied from 50% to 150% of the average value from the established model. This code was run 10 000 times. Based on this question and the following relevant questions, we have updated the legend of Supplementary Video 1 to read:

“Supplementary Video 1. (Single screen shot above) Multi-current Calcium, Action Potential and inward rectifier (CAPER) results from atrial cardiomyocytes in three dimensions. This video shows a comparison of our experimental results overlaid across an established mathematical in silico model of swine atrial physiology¹⁰. Though combining our experimental CAPER data (L-type calcium current [$I_{Ca,L}$], action potential duration at 90% repolarization [APD_{90}] and basal inward rectifier current [I_{K1}]) from atrial cardiomyocytes, we are able to see a glimpse of the reliability of our multi-dimensional experimental results. The three dimensional surface plot was generated by recording AP characteristics and currents from 10 000 individual in silico swine atrial cardiomyocytes whose parameters (ion channel conductances, time constants and rate constants for reaction kinetics) were randomly varied within a preset spread (50% -150%) of the average model value¹⁰. After reaching steady state (5 s), the in silico cells were stimulated with a 4 ms pulse of 20 pA. APD_{90} was acquired from all simulations able to generate an action potential (~40%) and the corresponding peak $I_{Ca,L}$ and I_{K1} were obtained from each successful cell at +10 mV and -90 mV, respectively, as was used experimentally. Note: our experimental recording temperature (21°C) differs from that of the model (37°C). Despite imperfect data complementation, this represents an exciting first step into high throughput corroboration of computational models and the capabilities of automated patch clamp in the investigation of cardiac electrophysiology.”(Supplementary Video 1, Legend).

62. Were (only) $I_{Ca,L}$ and I_{K1} varied? If so, were they independently varied and over which range?

These currents were not varied. The idea was to study the natural variation in these currents between cells. Thus, we did not vary $I_{Ca,L}$ or I_{K1} in particular. Instead we varied all parameters to produce a set of completely non-identical cells and then measured the peak values of these currents, resulting from the applied changes.

63. Did all 10,000 model cells generate an action potential (and thus an APD90) or did only a selection of the 10,000 model cells generate an action potential?

This is a very relevant question. Not all 10,000 model cells were able to produce an action potential. Out of 10,000 cells, approximately 4000 cells were able to do so, whereas the remaining ~6000 cells either remained silent or went into states of complete depolarisation, or turned out to be unstable combinations of the parameters (meaning physiologically irrelevant).

64. How were the simulations carried out with respect to hardware, software, stimulus current, stimulation frequency, steady-state data, etc.?

We performed the simulations on a local Desktop with 32GB RAM, using our own developed C code (serial programming). The model cells were stimulated once with currents of strength: 20pA and duration: 4 ms. In each case, the cell model was simulated for 5 s of real time to allow the model parameters to reach steady state, before the external stimulus was applied to invoke the action potential.

65. How are $I_{Ca,L}$ and I_{K1} along the axes of the 3D surface plot expressed? Are these perhaps expressed as (peak) current at a specific membrane potential?

To make sure the modelling data are comparable with the experimental results, the reported currents are indeed peak currents recorded at specific membrane potentials. For $I_{Ca,L}$, this was +10 mV And for I_{K1} , the current was measured at -90 mV.

66. Supplementary Information, References. Use a consistent format for your references and provide the DOI for all three references (writing doi: 10.1016/j.xpro.2020.100026 instead of doi: <https://doi.org/10.1016/j.xpro.2020.100026>, etc.). Remove “Find the latest version : Deep phenotyping of human induced pluripotent stem cell – derived atrial and ventricular cardiomyocytes” from Reference #1.

This has been corrected according to the reviewer’s suggestion.

Response to comments of Reviewer 2

I believe the revised manuscript is very much improved and I have no further critical comments.

We are very grateful for the guidance from this reviewer which has helped us to substantially improve this manuscript into a more complete and useful paper. We thank the reviewer for their time.

Response to comments of Reviewer 3

Seibertz et. al. have provided a substantial amount of new data in their revised manuscript to address my concerns regarding reproducibility and data quality. The new version presents a more nuanced look at the technology and shows both the obvious strengths and potential drawbacks of the technique. I have no further major concerns that need to be addressed. Congratulations on a really nice manuscript! I noted a couple of very small issues below (line numbers refer to manuscript without tracked changes).

We thank this reviewer for their kind words and guidance in formulating a manuscript that we agree is now a much more balanced exploration of the potential power of this method. A point-by-point overview of the corrections from this reviewer are detailed below.

1. Fig. 2E, missing scale bars for current traces

This mistake has been rectified. The updated **Figure 2E** is shown below.

Automated patch-clamp of native cardiomyocytes
Commun Biol COMMSBIO-21-3514B

Figure 2. Quality control of automated patch-clamp of native atrial and ventricular cardiomyocytes (Native CM). **A**, Representative 384-well APC chip partially filled with native CMs shown as a screenshot of Nanion DataControl 384 software with unused areas shaded out. Green boxes represent successful whole cell configuration during L-type calcium current ($I_{Ca,L}$) measurement. Red and blue rings indicate atrial and ventricular CM partitions respectively. **B**, Cellular density-dependent optical attachment rates of native ventricular cardiomyocytes and human ventricular induced pluripotent stem cell derived cardiomyocytes (hiPSC-CM) onto the patch-clamp aperture. **C**, Z factor analysis of individual runs for assessment of reproducibility ($I_{Ca,L}$ currents). **D**, NPC-384T chip (upper) with a photomicrograph of a single well with locations of interest labelled (lower). **E**, Representative membrane resistance during seal formation (left) and membrane rupture (right) of a native CM in the presence of a membrane test pulse (upper). **F**, Percentage success rates for whole-cell configuration from single plates of native CMs and hiPSC-CMs used in this protocol during $I_{Ca,L}$ acquisition. **G**, Photomicrograph of an attached ventricular hiPSC-CM (upper) and ventricular native CM (lower). The patch-clamp aperture is obscured by the cell. Scale bar denotes 50 μm . **H**, Time course of membrane capacitance (upper) and series resistance (R_{series} ; lower) in hiPSC-CMs (3 plates, same day) and native CMs (3 plates, same day) over successive $I_{Ca,L}$ experiments. **I**, Cellular capacitance (upper) and R_{series} (lower) averages from single plates over 3 separate

experimental days (D1, D2, D3) and therefore 3 animals. Data are mean±SEM. n= mean of 24 wells (B) or single plates (C, F, H, I).

2. Fig. 3F, Line 595, caption is duplicated

Indeed! We apologise for this mistake and have removed the duplicate caption.

3. Line 246, please clarify which numbers are APC and which are conventional patch

We thank the reviewer for this suggestion and have updated the text accordingly. The text in question now reads:

“(Atrial: -4.29 ± 0.17 [APC] vs -4.17 ± 1.74 [traditional patch-clamp] pA/pF; Ventricular: -8.65 ± 1.2 [APC] vs -5.5 ± 1.11 [traditional patch-clamp] pA/pF; Figure 3C, Supplementary Figure 3)” (page 12, line 256).

References

1. Obergrussberger, A. *et al.* The suitability of high throughput automated patch clamp for physiological applications. *J. Physiol.* **600**, 277–297 (2021).
2. Voigt, N., Zhou, X.-B. & Dobrev, D. Isolation of human atrial myocytes for simultaneous measurements of Ca²⁺ transients and membrane currents. *J. Vis. Exp.* e50235–e50235 (2013). doi:10.3791/50235
3. Voigt, N., Pearman, C. M., Dobrev, D. & Dibb, K. M. Methods for isolating atrial cells from large mammals and humans. *J. Mol. Cell. Cardiol.* **86**, 187–198 (2015).
4. Tang, J. M., Wang, J., Quandt, F. N. & Eisenberg, R. S. Perfusing pipettes. *Pflugers Arch.* **416**, 347–350 (1990).
5. Rapedius, M. *et al.* There is no F in APC: using physiological fluoride-free solutions for high throughput automated patch clamp experiments. *Front. Mol. Neurosci.* in review (2022).
6. Magistretti, J., Mantegazza, M., Guatteo, E. & Wanke, E. Action potentials recorded with patch-clamp amplifiers: are they genuine? *Trends Neurosci.* **19**, 530–534 (1996).
7. Horváth, A. *et al.* Low Resting Membrane Potential and Low Inward Rectifier Potassium Currents Are Not Inherent Features of hiPSC-Derived Cardiomyocytes. *Stem Cell Reports* **10**, 822–833 (2018).
8. Ochi, R., Chettimada, S., Kizub, I. & Gupte, S. A. Dehydroepiandrosterone inhibits I(Ca,L) and its window current in voltage-dependent and -independent mechanisms in arterial smooth muscle cells. *Am. J. Physiol. Heart Circ. Physiol.* **315**, H1602–H1613 (2018).
9. Senatore, A. *et al.* Mapping of dihydropyridine binding residues in a less sensitive invertebrate L-type calcium channel (LCa v 1). *Channels (Austin)*. **5**, 173–187 (2011).
10. Peris-Yagüe, V. *et al.* A Mathematical Model for Electrical Activity in Pig Atrial Tissue. *Front. Physiol.* **13**, 812535 (2022).

REVIEWERS' COMMENTS:

Reviewer #1 (Remarks to the Author):

The authors have carefully responded to my numerous comments, some of which were highly specific. Their time and efforts to do this are greatly appreciated. I agree with Reviewer 3 that the authors have created a very nice manuscript – congratulations!